# FRACTIONAL-ORDER SPIKING NEURAL NETWORK

**Chengjie Ge**[1]*, **Yufeng Peng**[1]*, **Zihao Li**[1], **Qiyu Kang**[1]†, **Xueyang Fu**[1], **Xuhao Li**[2], **Qixin Zhang**[3], **Junhao Ren**[3], **Zheng-Jun Zha**[1]

[1]University of Science and Technology of China    [2]Anhui University
[3]Nanyang Technological University
https://github.com/PhysAGI/spikeDE

## ABSTRACT

Spiking Neural Networks (SNNs) draw inspiration from biological neurons to enable brain-like computation, demonstrating effectiveness in processing temporal information with energy efficiency and biological realism. Most existing SNNs are based on neural dynamics such as the (leaky) integrate-and-fire (IF/LIF) models, which are described by *first-order* ordinary differential equations (ODEs) with Markovian characteristics. This means the potential state at any time depends solely on its immediate past value, potentially limiting network expressiveness. Empirical studies of real neurons, however, reveal long-range correlations and fractal dendritic structures, suggesting non-Markovian behavior better modeled by *fractional-order* ODEs. Motivated by this, we propose a *fractional-order* spiking neural network ($f$-SNN) framework that strictly generalizes integer-order SNNs and captures long-term dependencies in membrane potential and spike trains via fractional dynamics, enabling richer temporal patterns. We further release an open-source toolbox, **spikeDE**, to support the $f$-SNN framework across diverse architectures and real-world tasks. Experimentally, fractional adaptations of established SNNs into the $f$-SNN framework achieve superior accuracy, comparable energy efficiency, and improved robustness to noise, underscoring the promise of $f$-SNNs as an effective extension of traditional SNNs.

## 1 INTRODUCTION

Neural networks have evolved substantially as researchers continuously explore models that better reflect biological neural systems while maintaining strong performance. Traditional artificial neural networks (ANNs) excel across many tasks (Krizhevsky et al., 2012; LeCun et al., 2015; Vaswani et al., 2017) but differ from real biological mechanisms, and modern models require far more compute than the human brain (Dhar, 2020). This gap has motivated Spiking Neural Networks (SNNs) (Maass, 1997; Ghosh-Dastidar & Adeli, 2009; Lee et al., 2016; Wu et al., 2018; Zheng et al., 2021; Zhou et al., 2022), which model neural activity more realistically by communicating through discrete spikes rather than continuous values. Their event-driven computation paradigm allows for significant energy savings, particularly when implemented on neuromorphic hardware (Roy et al., 2019; Pei et al., 2019). Additionally, SNNs naturally handle time as part of their processing, making them well-suited for tasks with time-series data or real-time interactions in changing environments (Yao et al., 2023b; Luo et al., 2024; Yao et al., 2021). These features make SNNs strong candidates for applications that need both energy efficiency and good temporal processing.

Despite these advantages, existing SNN models predominantly describe spiking neuronal membrane-potential dynamics using the widely adopted Integrate-and-Fire (IF) and Leaky Integrate-and-Fire (LIF) neurons (Stein, 1967), along with variants including nonlinear spike initiation (Ermentrout, 1996; Fourcaud-Trocmé et al., 2003), ternary spikes (Guo et al., 2024), adaptive membrane time constants (Koch et al., 1996; Zhang et al., 2025), and threshold adaptation or learning (Bellec et al., 2018; Benda, 2021). These models discretize *first-order* ordinary differential equations (ODEs) which contains only $\mathrm{d}/\mathrm{d}t$ terms (Hodgkin & Huxley, 1952; Maass, 1997; Ghosh-Dastidar & Adeli, 2009; Eshraghian et al., 2023b) and assume a Markovian property in which the current state depends

---

*First two authors contributed equally.
†Correspondence to: Qiyu Kang <qiyukang@ustc.edu.cn>.

mainly on the immediate previous state (see (9)). While this simplification enables computational tractability, it fundamentally limits the expressiveness of these networks. Neurophysiological research has demonstrated that real neurons display far more complex behaviors influenced by long-term correlations (Gilboa et al., 2005), fractal dendritic structures (Coop et al., 2010; Kirch & Gollo, 2020), and the interaction of multiple active membrane conductances (La Camera et al., 2006; Miller & Troyer, 2002). These dynamics cannot be adequately captured by integer-order models (Ulanovsky et al., 2004; La Camera et al., 2006; Miller & Troyer, 2002; Spain et al., 1991) and suggest that non-Markovian dynamics play a significant role in biological neural computation. Fractional calculus instead offers mathematical tools for modeling such dynamics better than standard first-order ODEs (Diethelm, 2010; Baleanu et al., 2012). In contrast to integer-order calculus, the *fractional-order* derivative $\mathrm{d}^\alpha/\mathrm{d}t^\alpha$, with non-integer $\alpha$ values, considers the entire history of a function, weighted by a power-law kernel. The fractional leaky integrate-and-fire ($f$-LIF) neuron dynamic, introduced and studied in (Teka et al., 2014; Deng et al., 2022), serves as an example of applying these concepts. This model can effectively explain spiking frequency adaptations observed in most biological neurons (Ha & Cheong, 2017) and has been shown to generate more reliable spike patterns than integer-order models when subjected to noisy input (Teka et al., 2014). Despite these promising findings, the integration of SNNs and fractional neurons remains a largely unexplored area (Lee & Monahan).

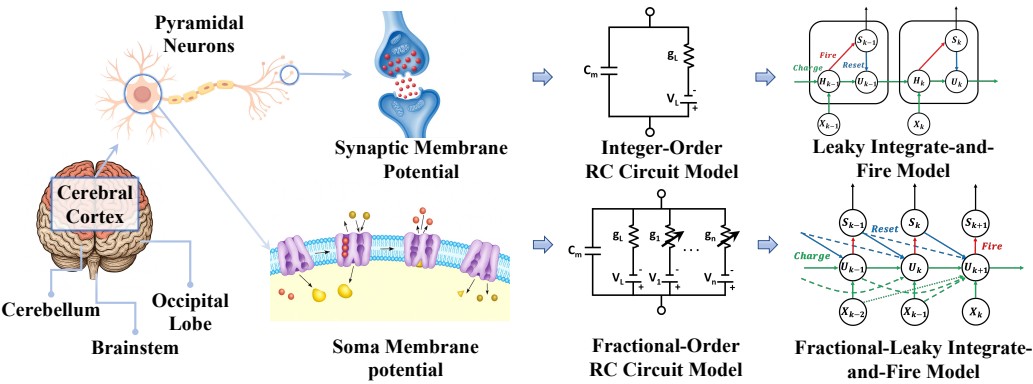

Figure 1: Comparison of traditional SNN and $f$-SNN framework.

In this paper, we introduce a generalized *fractional-order* SNN ($f$-SNN) framework, which incorporates fractional-order dynamics into the neuronal membrane potential charging. By replacing the first-order ODE neurons traditionally used in SNNs with fractional-order ODEs ($f$-ODEs), $f$-SNN naturally captures long-term dependencies that are beyond the capability of standard SNN models, leading to improved performance on tasks that require complex temporal processing. We highlight that our framework is a more general framework which subsumes many traditional SNNs as special instances by setting $\alpha = 1$. We evaluate $f$-SNN models on multiple benchmark datasets spanning neuromorphic event-driven vision, graph domains, and static vision fields. Experimental results show that $f$-SNN models consistently outperform conventional SNN models across various evaluation metrics. Moreover, $f$-ODEs are robust to perturbations (Sabatier et al., 2015; Kang et al., 2024c); in particular, neural $f$-ODEs admit tighter input–output perturbation bounds than integer-order models (Kang et al., 2024c). Building on this, an additional advantage of our proposed $f$-SNN framework is its superior robustness under input perturbations. These findings underscore the practical advantages of integrating fractional-order dynamics into SNNs and point to the broader applicability of our $f$-SNN in real-world scenarios.

**Main contributions.** Our objective in this paper is to formulate a generalized fractional-order SNN framework. Our key contributions are summarized as follows:

• We propose an $f$-SNN framework that integrates $f$-ODEs into SNNs to naturally capture long-term dependencies using the fractional-order operator $\mathrm{d}^\alpha/\mathrm{d}t^\alpha$. This framework generalizes the traditional class of integer-order SNNs that use IF, LIF neuron dynamics, and their variants, subsuming them as a special case by setting $\alpha = 1$.

- We establish fundamental theoretical distinctions between $f$-SNNs and traditional SNNs, proving that fractional-order dynamics confer three key advantages: persistent memory through power-law relaxation, irreducibility to finite classical ensembles, and enhanced robustness to perturbations.
- We underscore the compatibility of $f$-SNN, emphasizing its ability to be seamlessly integrated to augment the performance of many existing SNNs by using non-integer $\alpha$ with various neural network architectures like convolutional neural networks (CNN), Transformer, ResNet, and multilayer perceptron (MLP) (Vaswani et al., 2017; LeCun et al., 1989; He et al., 2016; Zhou et al., 2022). We provide the community with an open-source, out-of-the-box toolbox to support the $f$-SNN framework (see supplementary code and Section E). We conduct extensive experiments on multiple datasets, demonstrating that $f$-SNN consistently improves traditional SNNs, achieving superior accuracy, comparable energy efficiency, and enhanced robustness.

## 2 PRELIMINARIES

This section reviews essential concepts. We introduce fractional calculus, which generalizes derivatives to non-integer orders and naturally models systems with memory or non-local dependencies. We then outline conventional SNN approaches based on discretizing integer-order neuron dynamics.

### 2.1 FRACTIONAL CALCULUS

When examining a function $y(t)$ with respect to (w.r.t.) time $t$, we traditionally define the first-order derivative as the instantaneous rate of change: $\frac{\mathrm{d}y(t)}{\mathrm{d}t} := \lim_{\Delta t \to 0} \frac{y(t+\Delta t)-y(t)}{\Delta t}$. The literature offers various definitions of fractional derivatives (Tarasov, 2011). We focus on the Caputo fractional derivative $D^\alpha$ for the formal definition of $\mathrm{d}^\alpha/\mathrm{d}t^\alpha$ (Diethelm, 2010), which has the notable advantage of allowing initial conditions to be specified in the same manner as integer-order differential equations.

**Definition 1** (Caputo Fractional Derivative). *For a function $y(t)$ defined over an interval $[0, T]$, its Caputo fractional derivative of order $\alpha \in (0, 1]$ is given by (Diethelm, 2010):*

$$D^\alpha y(t) := \frac{1}{\Gamma(1-\alpha)} \int_0^t (t-\tau)^{-\alpha} y'(\tau) \, \mathrm{d}\tau, \tag{1}$$

*where $y'(\tau)$ denotes the first-order derivative of $y(\tau)$.*

**Remark 1.** (1) *reveals that the fractional derivative incorporates the historical states of the function through a power-weighted integral term when $\alpha \in (0,1)$, highlighting its memory dependence. As $\alpha \to 1$, the Caputo derivative $D^\alpha$ converges to the standard first-order derivative $\frac{\mathrm{d}}{\mathrm{d}t}$. Indeed, letting $F(s) = \mathcal{L}\{f(t)\}$ be Laplace transform of $f(t)$, we have $\mathcal{L}\{D_t^\alpha f(t)\} = s^\alpha F(s) - s^{\alpha-1} f(0)$ (Diethelm, 2010)[Theorem 7.1]. As $\alpha \to 1$, the Laplace transform of the Caputo fractional derivative converges to that of the traditional first-order derivative $sF(s) - f(0)$. Consequently, for $\alpha = 1$, $D^1 y = y'$, uniquely determined via the inverse Laplace transform (Cohen, 2007).*

A first-order ODE and its fractional extension with Caputo derivative can be written as

$$\text{integer-order ODE: } \frac{\mathrm{d}y(t)}{\mathrm{d}t} = f(t, y(t)); \tag{2}$$

$$\text{fractional-order ODE: } D^\alpha y(t) = f(t, y(t)), \tag{3}$$

where $f$ defines the system dynamics and initial condition $y(0) = y_0$ is specified in both cases.

### 2.2 INTEGER-ORDER SPIKING NEURON AND SNN

Existing SNN models predominantly describe spiking neuronal membrane-potential dynamics using discretized first-order ODEs with derivative $\mathrm{d}/\mathrm{d}t$, including the widely adopted IF and LIF dynamics (Stein, 1967) and variants with adaptive membrane time constants or threshold adaptation/learning (Koch et al., 1996; Zhang et al., 2025; Bellec et al., 2018; Benda, 2021). We present only standard IF and LIF in the main paper as a showcase; however, SNNs based on other neuron variants can be encapsulated and extended within our $f$-SNN framework.

**IF and LIF neurons.** Let $U(t)$ denote the membrane potential, $I_{\mathrm{in}}(t)$ the input current, $R > 0$ the membrane resistance, and $\tau > 0$ the membrane time constant. The standard subthreshold dynamics

of IF and LIF are described by the following first-order ODEs:

$$\text{IF neuron dynamics: } \tau \, \frac{dU(t)}{dt} = R \, I_{\text{in}}(t), \tag{4}$$

$$\text{LIF neuron dynamics: } \tau \, \frac{dU(t)}{dt} = -U(t) + R \, I_{\text{in}}(t). \tag{5}$$

A spike $S(t)$ is emitted when $U(t^-)$ crosses the threshold $\theta$, i.e., $S(t) = H\left(U\left(t^-\right) - \theta\right)$, where $H(\cdot)$ denotes the Heaviside step function. Upon spiking, one uses either a *soft reset* or a *hard reset*:

$$\text{1) soft reset: } U(t^+) \leftarrow U(t^-) - \theta; \quad \text{or} \quad \text{2) hard reset: } U(t^+) \leftarrow U_{\text{reset}}. \tag{6}$$

**Traditional SNN based on standard IF and LIF neuron dynamics.** Many SNNs are based on the neuron dynamics described in (4) and (5). In the simplest case, the forward Euler method is employed to solve a first-order ODE (2). Let $h > 0$ be the discretization step size, $t_k = kh$, $N = T/h$, and let $y_k$ denote the numerical approximation of $y(t_k)$. We have

$$y_{k+1} = y_k + h \, f(t_k, y_k), \quad k = 0, 1, \dots, N-1. \tag{7}$$

To make this time-varying solution compatible with sequence-based neural network models, we discretize time and treat $k$ as the sequence index. Correspondingly, applying (7) to (4) and (5) yields

$$\text{IF (discrete): } U_{k+1} = U_k + \frac{hR}{\tau} \, I_{\text{in},k}, \quad \text{LIF (discrete): } U_{k+1} = \left(1 - \frac{h}{\tau}\right) U_k + \frac{hR}{\tau} \, I_{\text{in},k}. \tag{8}$$

In practice, the factor is often absorbed into learnable synaptic weights, and the input current is represented as $X_k^{(\Phi)}$, where $X_k^{(\Phi)}$ the presynaptic spike vector or feature map (e.g., produced by Convolution, MLP, ResNet, or Transformer) with $\Phi$ denoting the learnable synaptic weights of those layers. For simplicity, in the following we omit $\Phi$ and denote it simply by $X_k$. For computational efficiency, we adopt the common simplifications $h = 1$ and $R = 1$, and define $\beta := 1 - \frac{1}{\tau}$. Together with spiking and reset mechanisms, we have the following iterations:

$$\begin{aligned}
\text{IF charge: } & U_k = U_{k-1} + X_k, \\
\text{or LIF charge: } & U_k = \beta \, U_{k-1} + X_k. \\
\text{spike: } & S_k = H(U_k - \theta), \\
\text{reset: (soft) } & U_k \leftarrow U_k - \theta \, S_k \;\; \text{or} \;\; \text{(hard) } U_k \leftarrow (1 - S_k) \, U_k + S_k \, U_{\text{reset}}.
\end{aligned} \tag{9}$$

Spikes are discrete and non-differentiable, which complicates SNN training. The surrogate-gradient method (Wu et al., 2018) keeps the hard spike $H(U - \theta)$ in the forward pass but uses a smooth surrogate for its derivative in backpropagation. A common choice is a threshold-shifted sigmoid, $H(U - \theta) \approx \sigma(U) = \frac{1}{1 + e^{\theta - U}}$ which preserves discrete firing while enabling gradient flow.

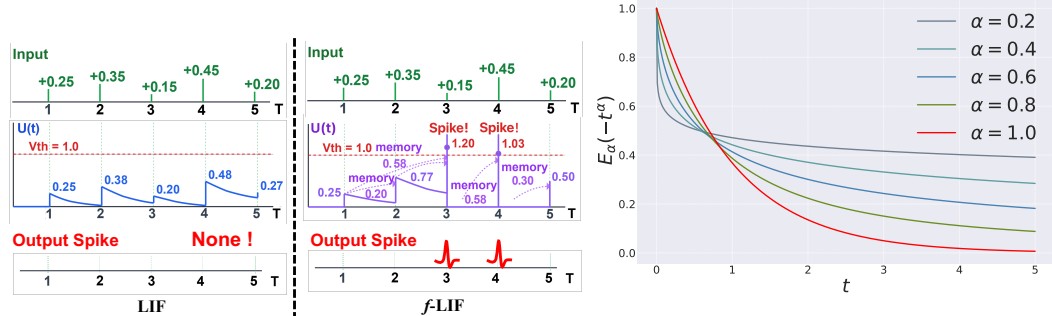

Figure 2: SNN vs $f$-SNN dynamics. In $f$-SNNs, past membrane potentials influence the current state via a power-law memory kernel; traditional integer-order SNNs lack this.

Figure 3: Mittag–Leffler function $E_\alpha(-t^\alpha)$. For $\alpha = 1$, LIF shows fast exponential decay $(E_1(-t) = e^{-t})$; for $0 < \alpha < 1$, $f$-LIF exhibits slow algebraic decay, reflecting memory.

## 3 $f$-SNN FRAMEWORK

We present the $f$-SNN framework in this section using fractional spiking neuronal dynamics based on $f$-ODEs, which generalize integer-order neuron dynamics such as standard IF and LIF neurons (4)

and (5). To make the time-varying solution compatible with neural network models, we follow the procedure in Section 2.2 to discretize time and enable iterations. Section 3.2 reveals the fundamental distinctions between $f$-SNNs and traditional SNNs through analysis of their long-time behavior, demonstrating how $f$-SNNs provide persistent memory via power-law relaxation, irreducibility to finite classical ensembles, and enhanced robustness.

## 3.1 FRAMEWORK

Traditional integer-order SNNs, as discussed in Section 2.2, model subthreshold spiking neuronal dynamics with first-order ODEs; (4) and (5) are representative examples. In our general $f$-SNN framework, we replace the first-order derivative $\mathrm{d}/\mathrm{d}t$ with the generalized Caputo fractional derivative $D^\alpha$ of order $\alpha \in (0, 1]$. Since IF and LIF are the dominant neuron models used in traditional SNNs, we present only their fractional extensions in the main paper as a showcase; however, many other neuron variants can likewise be encapsulated and extended within our $f$-SNN framework. We begin with the presentation of $f$-IF and $f$-LIF neurons.

**$f$-IF and $f$-LIF neurons.** The fractional dynamics of IF and LIF are described by the $f$-ODEs:

$$f\text{-IF neuron dynamics: } \tau\, D^\alpha U(t) = R\, I_{\mathrm{in}}(t), \tag{10}$$

$$f\text{-LIF neuron dynamics: } \tau\, D^\alpha U(t) = -U(t)\ +\ R\, I_{\mathrm{in}}(t). \tag{11}$$

Spike generation and reset follow the same rules as in the integer-order case: $S(t) = H\left(U\left(t^-\right) - \theta\right)$. These dynamics naturally introduce a memory effect: the current membrane potential depends on the entire history of the potential because, by definition (see (1)), the Caputo derivative includes an integral over past states. Biologically, such modeling is consistent with observed spike-frequency adaptation and long-memory behaviors (Teka et al., 2014; Ha & Cheong, 2017). The order $\alpha$ controls the degree of adaptation—$\alpha = 1$ recovers the standard IF/LIF models, while $\alpha < 1$ induces power-law memory and increased temporal correlations in the potential trace. Fractional neuron models are also observed to produce reliable spike patterns under noisy input (Baker et al., 2024).

**$f$-SNN based on $f$-IF and $f$-LIF neuron dynamics:** In Section 2.2, we apply the forward Euler method to discretize standard IF/LIF dynamics and obtain integer-order SNNs. Here, the $f$-IF (10) and $f$-LIF (11) neurons exhibit fractional dynamics that belong to the $f$-ODE class (3). We instead use the fractional Adams–Bashforth–Moulton (ABM) predictor discretization (Diethelm et al., 2004) to achieve this goal. Using the same time grid as above, $t_k = kh$ with $N = T/h$ and step size $h > 0$, and letting $y_k$ denote the numerical approximation of $y\left(t_k\right)$, we obtain

$$y_k = y_0 + \frac{1}{\Gamma(\alpha)} \sum_{j=0}^{k-1} \mu_{j,k}\, f(t_j, y_j), \quad k = 0, 1, \dots, N-1. \tag{12}$$

where the weight coefficients are $\mu_{j,k} = \frac{h^\alpha}{\alpha}[(k-j)^\alpha - (k-1-j)^\alpha]$. This formulation makes the memory effect explicit by incorporating weighted contributions from all past function evaluations, reflecting the nonlocal nature of $D^\alpha$. When $\alpha = 1$, (12) reduces exactly to the Euler method (7), further highlighting the compatibility between the $f$-SNN framework and traditional SNNs.

Applying (12) to (10) and (11) yields the fractional discrete updates:

$$f\text{-IF (discrete): } U_k = U_0 + \frac{R}{\tau\,\Gamma(\alpha)} \sum_{j=0}^{k-1} \mu_{j,k}\, I_{\mathrm{in},j},$$

$$f\text{-LIF (discrete): } U_k = U_0 + \frac{1}{\tau\,\Gamma(\alpha)} \sum_{j=0}^{k-1} \mu_{j,k}\left(-U_j + R\, I_{\mathrm{in},j}\right).$$

Similar to Section 2.2, we denote the general input as $X_k$, where $X_k$ is the presynaptic spike vector or feature map produced by various architectures (convolution, MLP, ResNet, Transformer, etc.). For simplicity, we set $h = 1$ and $R = 1$. Note that $\beta = 1 - \frac{1}{\tau}$ in IF/LIF neurons does not apply to the fractional cases. Instead, one obtains a history-convolution with a stationary power-law kernel. Define

$c_m^{(\alpha)} = \frac{1}{\tau^\alpha \, \alpha \Gamma(\alpha)} \left[ (m+1)^\alpha - m^\alpha \right]$. Then the fractional iterations (charge-spike-reset) are as follows:

$$f\text{-IF charge: } U_k = U_0 + \sum_{m=0}^{k-1} c_m^{(\alpha)} X_{k-m},$$

$$\text{or } f\text{-LIF charge: } U_k = U_0 + \sum_{m=0}^{k-1} c_m^{(\alpha)} \left( -U_{k-1-m} + X_{k-m} \right). \tag{13}$$

$$\text{spike: } S_k = H(U_k - \theta),$$

$$\text{reset: (soft) } U_k \leftarrow U_k - \theta \, S_k \quad \text{or} \quad \text{(hard) } U_k \leftarrow (1 - S_k)U_k + S_k U_{\text{reset}}.$$

Here, spiking and reset are applied at each step as usual. We follow the literature to use the surrogate-gradient method (Wu et al., 2018) to train $f$-SNN that keeps the hard spike $H(U - \theta)$ in the forward pass but uses a smooth surrogate for its derivative in backpropagation.

**Remark 2.** *Note that $c_m^{(\alpha)}$ is causal and decays as a power law, explicitly encoding memory. The profile of $c_m^{(\alpha)}$ is visualized in Fig. 8, highlighting the algebraic decay characteristic of fractional order systems. When $\alpha \to 1$, we have $c_m^{(1)} = 1/\tau$ for all $m$ (a constant kernel), and taking first differences of (13) recovers the Euler recursions (9). For efficiency, we may leverage the short-memory approximation principle (Deng, 2007; Podlubny, 1999) and truncate the sum in (13) to $\sum_{m=\max(0,\,k-M)}^{k-1}$, i.e., a sliding memory window of fixed width $M$. With fast (FFT-based) convolution, the full-memory case can be computed in $O(N \log N)$ time (Mathieu et al., 2013), while the truncated window yields $O(NM)$. The full model complexity is summarized in Section C.5.*

## 3.2 THEORETICAL ANALYSIS

In this section, we theoretically distinguish the $f$-SNN framework from traditional SNNs. We begin by proving that $f$-SNNs exhibit a persistent memory effect characterized by genuine long-range temporal dependence. We then demonstrate that the dynamics of $f$-SNNs generally cannot be exactly realized by any finite-dimensional linear system of integer-order modes, thereby establishing that fractional-order systems strictly exceed the expressive capacity of integer-order models. Finally, we prove that $f$-SNNs demonstrate superior robustness to input perturbations.

We first analyze membrane-potential relaxation under constant input, showing how distant past inputs keep influencing the present. For intuition, we focus on the LIF and $f$-LIF neurons and use the continuous formulations (5) and (11):

**Proposition 1** (Long-memory Behavior). *Under a constant current input $I_{\text{in}}(t) \equiv I_c$, assume the input is small enough that no spiking occurs over the interval considered (subthreshold regime). Then the solutions to the LIF (5) and the $f$-LIF dynamics (11) are*

$$U^{\text{LIF}}(t) = RI_c + \left[ U_0 - RI_c \right] e^{-t/\tau}, \tag{14}$$

$$U^{f\text{-LIF}}(t) = RI_c + \left( U_0 - RI_c \right) E_\alpha(-t^\alpha/\tau), \tag{15}$$

*respectively, where $E_\alpha(z) = \sum_{k=0}^{\infty} \frac{z^k}{\Gamma(\alpha k + 1)}$ is the Mittag–Leffler function Diethelm (2010). Key properties include:*

- *When $\alpha = 1$, $E_1(-t/\tau) = e^{-t/\tau}$, which is the classical exponential relaxation (Eshraghian et al., 2023a)[Eq(2)].*
- *For $0 < \alpha < 1$, $E_\alpha$ exhibits (i) initial stretched–exponential decay and (ii) a power-law tail for large $t$:*

$$E_\alpha \left( -\frac{t^\alpha}{\tau^\alpha} \right) \sim \frac{\tau^\alpha}{\Gamma(1 - \alpha) \, t^\alpha} \quad \text{as } t \to \infty,$$

*These behaviors are visualized in Fig. 3.*

**Remark 3.** *While both LIF and $f$-LIF converge to the same steady state, Proposition 1 highlights fundamentally different relaxation behaviors. The LIF uses an integer-order derivative (Markovian dynamics; future evolution depends only on the current state) and shows exponential relaxation $e^{-t/\tau}$, characteristic of memoryless processes. In contrast, the $f$-LIF employs the fractional derivative*

$D^\alpha$, which is inherently non-Markovian, incorporating a power-weighted integral over the entire past history. This is reflected in the Mittag–Leffler relaxation $E_\alpha(-t^\alpha/\tau)$: for $0 < \alpha < 1$, its power-law tail ($\sim t^{-\alpha}$) indicates that past inputs decay algebraically slow rather than exponentially fast, creating a persistent memory influence. This slow decay means that inputs from the distant past continue to influence the current membrane potential, enabling the $f$-LIF to naturally capture long-term temporal correlations.

The $f$-SNN framework demonstrates superior robustness compared to traditional SNNs. Empirical studies show that $f$-LIF neurons maintain reliable spike patterns under noisy inputs (Teka et al., 2014). Here, we provide theoretical robustness guarantees.

**Theorem 1** (Robustness of $f$-SNN). *Consider a fractional $f$-IF neuron governed by the dynamics $\tau D^\alpha U(t) = RI_{\text{in}}(t)$ with fractional order $0 < \alpha < 1$ and initial condition $U_0 = 0$. Under a constant input current $I_c$ subject to an additive perturbation $\epsilon$ (where $|\epsilon| \ll I_c$), the system exhibits the following robustness properties relative to the classical integer-order model ($\alpha = 1$):*

- *Membrane Potential Robustness:* *The membrane potential deviation due to perturbation evolves as:*

$$\Delta U^{f\text{-IF}}(t) = \frac{R\epsilon}{\tau\Gamma(\alpha+1)}t^\alpha \quad \text{(sub-linear growth)} \tag{16}$$

$$\Delta U^{\text{IF}}(t) = \frac{R\epsilon}{\tau}t \quad \text{(linear growth)} \tag{17}$$

  *For $0 < \alpha < 1$, the fractional-order dynamics suppress long-term perturbation accumulation through sub-linear temporal scaling.*

- *Spike Timing Sensitivity:* *For small perturbations $\epsilon \ll I_c$, the spike time shift magnitude scales as:*

$$|\Delta t_s^{f\text{-IF}}| \propto \epsilon \cdot I_c^{-(1+1/\alpha)} \tag{18}$$

$$|\Delta t_s^{\text{IF}}| \propto \epsilon \cdot I_c^{-2} \tag{19}$$

  *Since $(1 + 1/\alpha) > 2$ for $0 < \alpha < 1$, the fractional-order model exhibits enhanced spike timing robustness for high input currents.*

**Remark 4.** *The fractional-order dynamics yield distinct robustness advantages. The sub-linear perturbation growth $t^\alpha$ ($\alpha < 1$) significantly suppresses long-term accumulation compared to linear growth in classical models. Additionally, the enhanced spike timing stability becomes crucial for precise temporal coding applications (Bohte et al., 2002; Booij & tat Nguyen, 2005; Rathi et al., 2019). These properties make f-SNNs particularly suited for tasks requiring sustained accuracy and temporal precision under varying input conditions, as confirmed by our experiments in Section 4.*

We now establish that $f$-SNNs possess computational capabilities that fundamentally exceed those of finite integer-order systems:

**Theorem 2** (Irreducibility of $f$-IF Dynamics to Finite Classic LIF Ensembles). *Let $U^{f\text{-IF}}$ denote the trajectory of a $f$-IF neuron with order $\alpha \in (0, 1)$. There exist no finite integer $W$, weights $\{\phi_i\}_{i=1}^W$, and leak factors $\{\beta_i\}_{i=1}^W$ such that the following holds:*

$$\hat{U}_k = \sum_{i=1}^W \phi_i U_k^{\text{LIF}(\beta_i)} \equiv U_k^{f\text{-IF}} \quad \forall k.$$

*for general input $X_k$. The impulse response error of the approximation is $O(k^{\alpha-1})$, decaying algebraically slowly. The $f$-IF neuron is mathematically equivalent to an aggregate of integer-order LIF neurons if and only if $W \to \infty$, specifically as an integral over a continuum of leak factors.*

**Remark 5** (Implications for Expressive Power). *A single $f$-IF neuron represents a continuum of timescales that would require infinitely many integer-order SNN units for exact equivalence. The slow $O(k^{\alpha-1})$ error decay confirms that such long-range dependencies are inaccessible to finite-order models. Moreover, this expressivity advantage is not "washed out" by the spiking nonlinearity: in Corollary 1, we show that $f$-IF spike trains encode temporal information that no finite LIF ensemble can reproduce, even under arbitrary Boolean combinations.*

## 4 EXPERIMENTS

In this section, we evaluate $f$-SNNs on benchmarks spanning neuromorphic event-driven vision and graph domains. Additional experiments on static datasets, including CIFAR10, CIFAR100, and ImageNet, are detailed in Section D.2.3. Across metrics, $f$-SNNs consistently outperform conventional SNNs. In particular, fractional adaptations of established SNN architectures within the $f$-SNN framework achieve higher accuracy, comparable energy efficiency, and improved robustness to noise, supporting $f$-SNNs as an effective extension of traditional SNNs. *Importantly, our primary aim is not to achieve state-of-the-art (SOTA) results, but to demonstrate that the generalized $f$-SNN framework can improve existing integer-order SNNs.* To our knowledge, SOTA performance on very large datasets typically requires substantial computational resources, even within the energy-efficient SNN community. We focus on fair comparisons by replacing the integer-order IF/LIF modules (9) in traditional SNNs with the $f$-IF/$f$-LIF modules (13) from our $f$-SNN framework.

### 4.1 NEUROMORPHIC DATA CLASSIFICATION TASKS

Neuromorphic data are event-driven and exhibit strong spatiotemporal correlations. SNNs, with their natural adaptability to spatiotemporal data (e.g., dynamic event processing and sparse coding), efficiently model these correlations. Therefore, we conducted a series of experiments on neuromorphic datasets. Our experiments primarily focus on the following key evaluation aspects: (1) **Classification performance**, and (2) **Robustness** of the proposed $f$-SNN model. More ablation studies and experimental details will be presented in the Section D.

Table 1: Neuromorphic data classification results in terms of classification accuracy (%) on the multiple datasets. The best results are **boldfaced**, while the runner-ups are underlined.

| Datasets/Configs | Architecture | Timesteps | LIF (SpikingJelly) | LIF (snnTorch) | $f$-LIF ($f$-SNN) |
|---|---|---|---|---|---|
| **N-MNIST** | CNN-based | 16 | 0.9927 | 0.9908 | **0.9948** |
| **DVS-Lip** | CNN-based | 16 | 0.4241 | 0.3271 | **0.4342** |
| **DVS128Gesture** | CNN-based | 16 | 0.9340 | 0.8899 | **0.9480** |
| | Transformer-based | 16 | 0.9514 | 0.8715 | **0.9583** |
| **N-Caltech101** | CNN-based | 16 | 0.6682 | 0.6521 | **0.7026** |
| | Transformer-based | 16 | 0.7263 | 0.6567 | **0.7627** |
| **HarDVS** | CNN-based | 8 | 0.4610 | 0.4626 | **0.4766** |
| | Transformer-based | 8 | 0.4520 | 0.4614 | **0.4723** |

**Dataset & Baselines.** We conduct comprehensive evaluations of the proposed $f$-SNN framework on neuromorphic datasets including N-MNIST (Orchard et al., 2015), DVS128Gesture (Amir et al., 2017), N-Caltech101 (Orchard et al., 2015), DVS-Lip (Tan et al., 2022), and the large-scale dataset HarDVS (Wang et al., 2024). The dataset details and experiment setting details are provided in Appendix Section D.

**Experimental Setup.** For the N-MNIST dataset, we set the batch size to 512, the number of time steps $T$ to 16, and train for 100 epochs using the Adam optimizer. For other neuromorphic datasets, we follow the standard preprocessing pipeline of the SpikingJelly framework to convert event data into frame representations. For time step configuration, DVS128Gesture, N-Caltech101, and DVS-Lip are set to 16 time steps, while HarDVS is set to 8 time steps due to the large data size. N-Caltech101 is split into training and test sets with an 8:2 ratio. These datasets use a batch size of 16, with input dimensions uniformly adjusted to 128×128 pixels. We train for 200 epochs using the Adam optimizer.

**Classification Performance.** In $f$-SNN, we have a hyperparameter $\alpha$ indicating the fractional order, which gives the model an additional degree of freedom to capture richer temporal patterns. During experiments, the optimal $\alpha$ is obtained via hyperparameter tuning. The experimental results on neuromorphic datasets are shown in Table 1. We conduct comprehensive comparisons between $f$-SNN and baseline models. The experimental results demonstrate that under the same network configurations, regardless of whether CNN or Transformer architectures are employed, $f$-SNN significantly and consistently outperforms baseline networks implemented based on SpikingJelly and snnTorch frameworks. These results validate the effectiveness and superiority of the proposed $f$-SNN method in terms of classification performance. This is because $f$-SNN captures long-term

dependencies in membrane potential via fractional dynamics, enabling richer temporal patterns than traditional models.

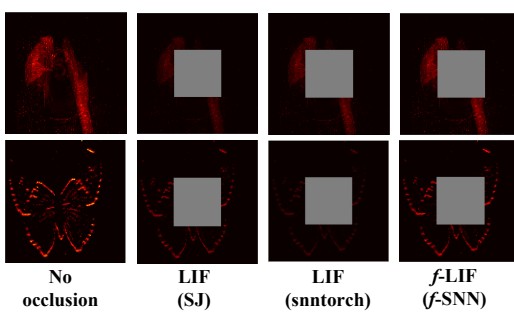

No occlusion    LIF (SJ)    LIF (snntorch)    *f*-LIF (*f*-SNN)

Figure 5: Feature map visualizations of LIF and f-LIF in occluded block scenarios.

**Robustness Analysis.** We further validate the robustness advantages of *f*-SNN. We comprehensively test the model's stability from five dimensions: noise injection, occlude block, temporal truncate, temporal jitter, and discard frame. Detailed experimental settings are provided in Section D.1.3. The experimental results are shown in Fig. 4, where *f*-SNN significantly outperforms baseline methods across all five robustness testing dimensions. Particularly under high-intensity noise injection and occlude block interference conditions, our method demonstrates exceptional anti-interference capability and stability. To more intuitively validate our viewpoint, we visualize the shallow feature maps with occlude blocks, with results shown in Fig. 5. Our

*f*-SNN model can better capture object features under occlusion conditions compared to the other two models. This advantage is primarily attributed to the inherent characteristics of the *f*-LIF neuron module, which can generate more stable and reliable spike patterns, thereby effectively enhancing the noise suppression capability and robustness performance of the entire network. We refer the readers to the discussions in Section 3.1. Evaluation details are provided in the appendix.

## 4.2 GRAPH LEARNING TASKS.

For graph learning tasks, our experiments focus on the following key aspects of evaluation: (1) **Node Classification** performance; (2) **Energy Efficiency**; and (3) the **Robustness** of the proposed *f*-SNN framework.

**Dataset & Baselines.** We conduct experiments on two mainstream GNN methods: SGCN (Zhu et al., 2022), and DRSGNN (Zhao et al., 2024), using several commonly used graph learning datasets. Specifically, Node classification is performed with SGCN and DRSGNN on Cora (McCallum et al., 2000), Citeseer (Sen et al., 2008), Pubmed (Wang et al., 2019), Photo, Computers, and ogbn-arxiv (Hu et al., 2020). To ensure fairness, we only replace the integer-order neuron modules in the baseline models with our proposed modules, i.e., the LIF neuron (9) in SGCN and DRSGNN are changed to our *f*-LIF iterations (13). This ensures our fractional adaptations have the same trainable parameters as the baselines; only the charging phases (9) and (13) differ. Dataset details are provided in Section D.

**Experiment Setup.** For node classification tasks based on SGCN and DRSGNN, we use Poisson spike encoding. The number of timesteps $N$ is set to 100, and the batch size to 32. Datasets are split into

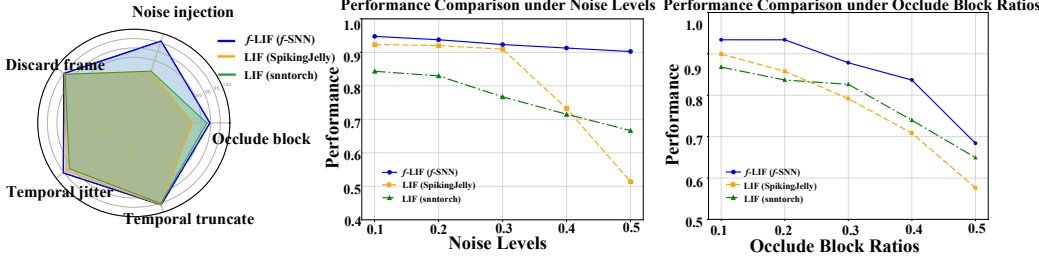

Figure 4: Robustness comparison between the proposed *f*-SNN and two integer-order baselines (LIF in SpikingJelly and LIF in snnTorch). Left: Radar chart aggregating five corruption types (larger is better): Gaussian noise injection, center occlude block, temporal truncate, temporal jitter, and discard frame. Middle: Performance vs. noise level (x-axis: Gaussian noise std). Right: Performance vs. occlusion ratio (x-axis: area ratio of the center block). The *f*-LIF (*f*-SNN) shows consistently higher performance and slower degradation under all corruption types.

training/validation/test with ratios $0.7/0.2/0.1$. For DRSGNN experiments, the positional-encoding dimension is 32, using Laplacian (LSPE) (Dwivedi et al., 2023) or random-walk (RWPE) (Dwivedi et al., 2021) encodings. All experiments are run independently 20 times; we report the mean and standard deviation. Other experimental details are included in Section D.

Table 2: Node classification results in terms of classification accuracy (%) on multiple datasets. The best results are **boldfaced**, while the runner-ups are underlined. Standard deviations are provided as subscripts. The choice of $f$-SNNs' parameter $\alpha$ will be shown in Table 4.

| Methods | Cora | Citeseer | Pubmed | Photo | Computers | ogbn-arxiv |
|---|---|---|---|---|---|---|
| SGCN (SJ) | $81.81_{\pm 0.69}$ | $71.83_{\pm 0.23}$ | $86.79_{\pm 0.32}$ | $87.72_{\pm 0.25}$ | $70.86_{\pm 0.24}$ | $50.26_{\pm 0.11}$ |
| SGCN (snnTorch) | $83.12_{\pm 1.41}$ | $71.68_{\pm 0.95}$ | $59.82_{\pm 1.07}$ | $83.34_{\pm 0.89}$ | $74.88_{\pm 0.87}$ | $21.55_{\pm 0.13}$ |
| SGCN ($f$-SNN) | $\mathbf{88.08}_{\pm 0.58}$ | $\mathbf{73.80}_{\pm 0.51}$ | $87.17_{\pm 0.28}$ | $\mathbf{92.49}_{\pm 0.32}$ | $\mathbf{89.12}_{\pm 0.21}$ | $\mathbf{51.10}_{\pm 0.14}$ |
| DRSGNN (SJ) | $83.30_{\pm 0.64}$ | $72.72_{\pm 0.24}$ | $87.13_{\pm 0.34}$ | $88.31_{\pm 0.15}$ | $76.55_{\pm 0.17}$ | $50.13_{\pm 0.14}$ |
| DRSGNN (snnTorch) | $80.98_{\pm 1.71}$ | $68.00_{\pm 0.69}$ | $59.56_{\pm 1.05}$ | $82.28_{\pm 0.93}$ | $76.78_{\pm 0.81}$ | $28.46_{\pm 0.25}$ |
| DRSGNN ($f$-SNN) | $\mathbf{88.51}_{\pm 0.62}$ | $\mathbf{75.11}_{\pm 0.45}$ | $87.29_{\pm 0.32}$ | $91.93_{\pm 0.20}$ | $\mathbf{88.77}_{\pm 0.20}$ | $\mathbf{53.13}_{\pm 0.13}$ |

**Node Classification Performance.** The experimental results based on SGCN and DRSGNN are shown in Table 2. Our fractional extension of SGCN and DRSGNN outperforms the original versions implemented with traditional *integer-order* SNN toolboxes (SpikingJelly or snnTorch) in terms of accuracy. These results highlight the clear advantage of our method in improving model accuracy.

**Energy Consumption Analysis.** Following (Yao et al., 2023a; 2024), we compare the energy consumption of $f$-SNN and the integer-order method (SpikingJelly). Fig. 6a shows that $f$-SNN achieves higher accuracy and significantly lower energy consumption across datasets, demonstrating its superior energy efficiency. Details will be discussed in the Section D.3.

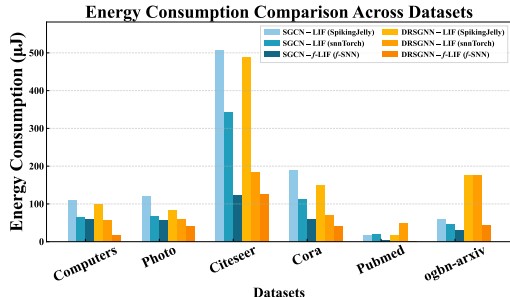

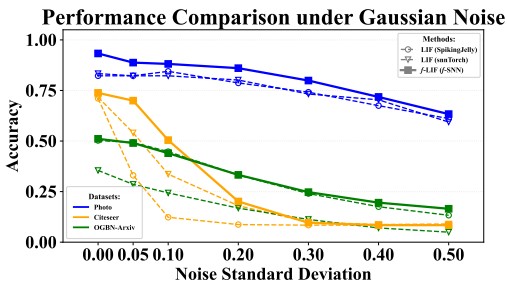

(a) Comparison of energy consumption between integer-order SpikingJelly and snnTorch baselines and our $f$-SNN framework.

(b) Robustness test for SGCN.

Figure 6: Energy consumption and robustness evaluation. Best zoomed on screen.

**Robustness Test.** We further validate the robustness advantage of $f$-SNN. Specifically, we randomly add Gaussian noise (Hall, 1994) of varying intensities to the spike signals input to the network to evaluate the robustness of spiking graph neural networks under different noise conditions. The experimental results are shown in Fig. 6b.

## 5 CONCLUSION

In this work, we introduced a new $f$-SNN framework, which extends traditional SNNs by replacing first-order ODEs with fractional-order ODEs to capture the non-Markovian characteristics and long-term dependencies observed in biological neurons. Our experiments demonstrate that $f$-SNNs consistently outperform integer-order SNNs across neuromorphic vision and graph benchmarks, achieving higher accuracy, comparable energy efficiency, and improved noise robustness. The accompanying open-source toolbox facilitates adoption of the $f$-SNN framework across diverse architectures and applications. These results establish $f$-SNNs as a promising extension of traditional SNNs, offering a mathematically rigorous and biologically plausible approach to enhancing neuromorphic computing capabilities.

## ACKNOWLEDGMENTS

This work is supported by the National Natural Science Foundation of China under Grants 62225207, 62436008, 62576326, and 12301491. To improve the readability, parts of this paper have been grammatically revised using ChatGPT OpenAI (2022).

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

# A  RELATED WORK

## A.1  SPIKING NEURAL NETWORKS

Traditional artificial neural networks (ANNs) have achieved remarkable success across a wide range of tasks (Krizhevsky et al., 2012; LeCun et al., 2015; Vaswani et al., 2017). However, these models differ significantly from biological neural networks, and their computational requirements far exceed those of the human brain (Dhar, 2020). This discrepancy has motivated the development of Spiking Neural Networks (SNNs) (Maass, 1997; Ghosh-Dastidar & Adeli, 2009; Lee et al., 2016; Wu et al., 2018; Zheng et al., 2021; Zhou et al., 2022), which offer a more biologically plausible model by communicating through discrete spikes instead of continuous signals. Their event-driven computation enables significant energy savings, particularly on neuromorphic hardware (Roy et al., 2019; Pei et al., 2019). Furthermore, SNNs treat time as an intrinsic component, making them well-suited for applications in time-series prediction and real-time interactive systems (Yao et al., 2023b; Luo et al., 2024; Yao et al., 2021). Early biophysical models such as the Hodgkin-Huxley model (Hodgkin & Huxley, 1952) offer an accurate description of action-potential generation but are computationally expensive, particularly for large-scale learning tasks. Consequently, simplified neuron models, such as the Integrate-and-Fire (IF) and Leaky Integrate-and-Fire (LIF) models, have become widely adopted in SNN research (Abbott, 1999; Stein, 1965; 1967). Beyond the basic IF/LIF models, a variety of extensions have been proposed to address different modeling challenges. Adaptive Leaky Integrate-and-Fire (ALIF) SNN neuron incorporates neural adaptation mechanisms, such as adaptive thresholds, enhancing the temporal dependency modeling and working memory capacity of SNNs (Bellec et al., 2018; Benda, 2021). The Generalized LIF (GLIF) introduces a more physiologically motivated framework, enabling accurate spike detection and unsupervised differentiation of cortical cell types (Teeter et al., 2018). The Complementary LIF (CLIF) model further enhances temporal gradient propagation and long-term dependency learning by incorporating a complementary membrane potential state (Huang et al., 2024). The Parallel Spiking Neuron (PSN) model eliminates the need for reset mechanisms, facilitating fully connected temporal modeling that allows for time-step parallelism, which accelerates both training and inference (Fang et al., 2023b). Additional neuron models, such as ternary spikes (Guo et al., 2024) and adaptive membrane time constants (Koch et al., 1996; Zhang et al., 2025), further extend the capabilities of SNNs. Despite these advances, most existing SNNs discretize only first-order ordinary differential equations (ODEs), which describe dynamics governed by $d/dt$ terms, and assume a Markovian property, where the state at any time depends only on its immediate past (see (9)) (Maass, 1997; Ghosh-Dastidar & Adeli, 2009; Eshraghian et al., 2023b). While this simplification aids tractability, it imposes limitations on expressiveness. Neurophysiological evidence indicates that real neurons exhibit long-range temporal correlations (Gilboa et al., 2005), fractal dendritic morphologies (Coop et al., 2010; Kirch & Gollo, 2020), and interactions among multiple active membrane conductances (La Camera et al., 2006; Miller & Troyer, 2002), leading to dynamics that integer-order, Markovian models capture only imperfectly (Ulanovsky et al., 2004; La Camera et al., 2006; Miller & Troyer, 2002; Spain et al., 1991).

**Distinction from Prior SNN Families.** The models discussed above all belong to the *integer-order* family, where the subthreshold dynamics can be expressed as a first-order ordinary differential equation (ODE) of the form

$$\text{First-order SNN neuron dynamics:} \quad \frac{dU(t)}{dt} = \text{Dynamic}\big(U(t), I_{\text{in}}(t)\big), \tag{20}$$

where $\text{Dynamic}(\cdot)$ denotes the specific membrane-potential update rule used by models such as IF, LIF (cf. (10) and (11)), ALIF, GLIF, CLIF, and other related variants. These approaches mainly explore different choices for the function $\text{Dynamic}(\cdot)$, all within the same first-order, Markovian framework.

In contrast, our work introduces a *fractional-order* SNN framework, which is based on

$$\text{Fractional-order SNN neuron dynamics:} \quad D_t^\alpha U(t) = \text{Dynamic}\big(U(t), I_{\text{in}}(t)\big), \ 0 < \alpha \le 1, \tag{21}$$

where $D_t^\alpha$ represents a fractional (nonlocal-in-time) derivative. This formulation generalizes the integer-order case, which is recovered when $\alpha = 1$, and incorporates long-term memory in the membrane potential and spike trains through fractional dynamics. In the main text, we instantiate this framework with fractional IF and LIF neurons, but the same approach can naturally extend to

more complex neuron models, such as ALIF, GLIF, and CLIF. Our `spikeDE` toolbox offers modular implementations that facilitate the realization of these fractional variants. We believe this contribution significantly advances the field by introducing nonlocal-in-time discrete dynamics to SNN modeling.

## A.2 FRACTIONAL NEURAL DYNAMICS AND LEARNING

We first review the prior application of fractional calculus into SNNs/ANNs and then position our contribution accordingly.

**Fractional biological neuron modelling and shallow fractional Hopfield spiking network**. At the neuron level, the $f$-LIF modelling (Teka et al., 2014; Deng et al., 2022) of biological neurons explains spike-frequency adaptation in pyramidal neurons (Ha & Cheong, 2017) and yields more reliable spike patterns under noise (Teka et al., 2014). The work (Rombouts & Bohte, 2010) proposes that a neuron's spike-train can be interpreted as a fractional derivative of its input signal. They show encoding/decoding efficiency and link fractional dynamics to predictive coding. At the network level, related efforts investigate shallow fractional Hopfield-type spiking networks (Zhang et al., 2026). These studies primarily focus on dynamical system properties, proving the coexistence of multiple equilibrium points, solution boundedness, and global attractivity—rather than learning representations for complex tasks.

*Distinction: Crucially, prior work is restricted to biological modeling, signal-approximation, or dynamical analysis of fixed-weight, shallow networks, neglecting the learning problem. We bridge this gap by formulating the first generalizable $f$-SNNs framework for end-to-end training. This advances $f$-SNNs from theoretical constructs to a trainable computational paradigm compatible with modern deep architectures (e.g., Transformers), strictly generalizing integer-order SNNs.*

**Fractional deep learning and fractional differential equation neural solvers**. In the continuous ANN domain, fractional calculus has been integrated into deep learning frameworks to enhance expressivity. For instance (Kang et al., 2024a;b) leverage fractional calculus to improve graph neural network performance and robustness, while Nobis et al. (2024) utilizes fractional diffusion processes to improve diversity in generative modeling. Separately, in the domain of scientific computing, Physics-Informed Neural Networks (PINNs) have been extended to solve fractional partial differential equations ($f$-PINNs) (Pang et al., 2019). Subsequent developments have focused on scalability, such as gradient-enhanced variants for convergence (Yu et al., 2022b), and optimized training via operator-matrix methods for high-dimensional problems (Ma et al., 2023; Taheri et al., 2024)

*Distinction: Our $f$-SNN model fundamentally differs from these approaches. First, unlike $f$-PINNs, which serve as function approximators to solve a given fractional equation, we embed fractional dynamics inside the neuron model as a computational engine. Second, unlike fractional ANNs that operate on continuous signals, $f$-SNNs function in the discrete, event-driven domain.*

**Fractional-order gradients for training NNs**. A complementary line of research applies fractional derivatives to define gradient operators and learning dynamics for training SNNs/ANNs. For example, fractional gradient descent algorithms (Khan et al., 2018; Shin et al., 2023) replace the standard integer-order gradient update with a fractional counterpart. These methods smooth the optimization landscape, enabling faster convergence and better escape from local minima compared to standard stochastic gradient descent (SGD). In the spiking domain, Gyöngyössy et al. (2022); Yang et al. (2025; 2023) have applied fractional gradients for training SNNs.

*Distinction: These approaches use fractional calculus as an optimization tool to adjust the weight update trajectory, whereas our work embeds fractional dynamics within the neurons themselves. This is analogous to the difference between designing a network optimizer (Adam vs. SGD) versus changing the network architecture (CNN vs. Transformer).*

## A.3 EVENT CAMERA

Event cameras, as novel bio-inspired sensors, capture pixel-level brightness changes through an asynchronous triggering mechanism (Gallego et al., 2020). With microsecond-level temporal resolution (equivalent to 10,000 fps) and a high dynamic range (140 dB) (Rebecq et al., 2019; Wang et al., 2025), they provide a groundbreaking solution for perception in high-speed dynamic scenes. Unlike traditional frame-based vision sensors, event cameras only record pixels undergoing changes in the

scene, generating sparse event streams. This data structure not only significantly reduces redundant information but also enables robust perception under rapid motion and challenging lighting conditions. In the field of event stream processing, the asynchronous and sparse nature of event data poses challenges for conventional frame-based CNN algorithms. To address these challenges, researchers have proposed various encoding and processing methods tailored to the unique characteristics of event data. For instance, (Neftci et al., 2019) introduced a surrogate gradient-based method to transform event streams into pulse sequences compatible with neural network processing. (Xu et al., 2025) proposed the Motion-Encoded Time-Surface (METS), which dynamically encodes pixel-level decay rates in time surfaces to capture the spatiotemporal dynamics reflected by events. This approach successfully addresses the challenge of pose tracking in high-speed scenarios using event cameras. Significant progress has also been made in network architecture optimization. (Yu et al., 2022a) developed the STSC-SNN model, which introduces synaptic connections with spatiotemporal dependencies, significantly enhancing the ability of spiking neural networks to process temporal information. Furthermore, event cameras, with their low latency and high dynamic range, have demonstrated broad application potential in fields such as robotic control, autonomous driving, and object tracking. For example, (Cuadrado et al., 2023) proposed a 3D convolution-based spatiotemporal feature encoding method, utilizing a hierarchical separable convolution architecture to greatly improve the accuracy and efficiency of optical flow estimation in driving scenarios using event cameras. On the hardware optimization front, researchers have actively developed systems tailored for efficient event stream processing. (Isik et al., 2024) constructed a neuromorphic vision system based on the Intel Loihi 2 chip, achieving significantly lower power consumption compared to traditional GPU solutions, thus providing critical support for the efficient deployment of event cameras.

## B  MORE TECHNICAL DETAILS

### B.1  MORE ABOUT FRACTIONAL CALCULUS

In Section 2.1 of the main paper, we presented the (left) Caputo fractional derivative and discussed numerical schemes for solving fractional-order ODEs. Here we provide additional background on fractional calculus that underpins our approach. For clarity, we note that throughout the main paper, all references to the Caputo fractional derivative specifically denote the left Caputo fractional derivative $D^\alpha$.

#### B.1.1  CLASSICAL DERIVATIVES AND INTEGRALS

For a scalar function $y(t)$, the ordinary first-order derivative captures its instantaneous rate of change:

$$\frac{\mathrm{d}y(t)}{\mathrm{d}t} = \dot{y}(t) := \lim_{\Delta t \to 0} \frac{y(t + \Delta t) - y(t)}{\Delta t}. \tag{22}$$

Let $J$ denote the integral operator that assigns to each function $y(t)$, which we assume to be Riemann integrable on the closed interval $[0, T]$, its antiderivative starting from $a$:

$$Jy(t) := \int_a^t y(\tau)\mathrm{d}\tau \quad \text{for } t \in [0, T]. \tag{23}$$

When considering positive integers $n \in \mathbb{N}^+$, we write $J^n$ to indicate the $n$-fold composition of $J$, where $J^1 = J$ and $J^n := J \circ J^{n-1}$ for $n \geq 2$. Through repeated integration by parts, one can show that (Diethelm, 2010)[Lemma 1.1.]:

$$J^n y(t) = \frac{1}{(n-1)!} \int_a^t (t - \tau)^{n-1} y(\tau)\mathrm{d}\tau \quad \text{for } n \in \mathbb{N}^+. \tag{24}$$

#### B.1.2  EXTENDING TO NON-INTEGER ORDERS: FRACTIONAL INTEGRALS AND DERIVATIVES

**Fractional Integrals Operators:** Fractional integrals extend classical integration theory by allowing non-integer orders of integration. Among various formulations, the Riemann-Liouville fractional integrals are particularly fundamental (Tarasov, 2011)[page 4]. For a positive real parameter $\alpha \in \mathbb{R}^+$, we define the left-sided and right-sided Riemann-Liouville fractional integral operators, denoted $J_{\text{left}}^\alpha$

and $J_{\text{right}}^{\alpha}$, as follows:

$$
J_{\text{left}}^{\alpha} y(t) := \frac{1}{\Gamma(\alpha)} \int_a^t (t - \tau)^{\alpha-1} y(\tau) \, \mathrm{d}\tau,
$$

$$
J_{\text{right}}^{\alpha} y(t) := \frac{1}{\Gamma(\alpha)} \int_t^b (\tau - t)^{\alpha-1} y(\tau) \, \mathrm{d}\tau,
$$
(25)

where $\Gamma(\alpha)$ represents the gamma function, which provides a continuous extension of the factorial operation to real and complex domains. The key distinction from classical repeated integration lies in the flexibility of the order parameter: while traditional calculus restricts the order $n$ in (24) to positive integers, the fractional order $\alpha$ in (25) spans the entire positive real line, enabling a continuous spectrum of integration orders.

**Fractional Derivative Operators:** Parallel to fractional integration, the concept of a fractional derivative extends the operation of differentiation to arbitrary non-integer orders. This allows for a more nuanced understanding of rates of change in complex systems.

One common formulation is the Riemann-Liouville fractional derivative. The left-sided ($^{\text{RL}}D^{\alpha}$) and right-sided ($^{\text{RL}}D_{b-}^{\alpha}$) versions are formally defined by first applying a fractional integral and then an integer-order derivative (Tarasov, 2011):

$$
^{\text{RL}}D^{\alpha} y(t) := \frac{\mathrm{d}^m}{\mathrm{d}t^m} J_{\text{left}}^{m-\alpha} y(t) = \frac{1}{\Gamma(m-\alpha)} \frac{\mathrm{d}^m}{\mathrm{d}t^m} \int_0^t \frac{y(\tau) d\tau}{(t-\tau)^{\alpha-m+1}}
$$

$$
^{\text{RL}}D_{b-}^{\alpha} y(t) := (-1)^m \frac{\mathrm{d}^m}{\mathrm{d}t^m} J_{\text{right}}^{m-\alpha} y(t) = \frac{(-1)^m}{\Gamma(m-\alpha)} \frac{\mathrm{d}^m}{\mathrm{d}t^m} \int_t^T \frac{y(\tau) d\tau}{(\tau-t)^{\alpha-m+1}},
$$
(26)

Here, $m$ is the smallest integer such that $m - 1 < \alpha \le m$.

Another widely used definition is the Caputo fractional derivative. The left-sided ($D^{\alpha}$) and right-sided ($D_{b-}^{\alpha}$) Caputo derivatives are distinct from the Riemann-Liouville formulation in the order of operations: they involve first taking an integer-order derivative and then applying a fractional integral (Tarasov, 2011). They are defined as follows:

$$
D^{\alpha} y(t) := J_{\text{left}}^{m-\alpha} \frac{\mathrm{d}^m}{\mathrm{d}t^m} y(t) = \frac{1}{\Gamma(m-\alpha)} \int_a^t \frac{\frac{\mathrm{d}^m}{\mathrm{d}\tau^m} y(\tau) \, \mathrm{d}\tau}{(t-\tau)^{\alpha-m+1}},
$$

$$
D_{b-}^{\alpha} y(t) := (-1)^m J_{\text{right}}^{m-\alpha} \frac{\mathrm{d}^m}{\mathrm{d}t^m} y(t) = \frac{(-1)^m}{\Gamma(m-\alpha)} \int_t^b \frac{\frac{\mathrm{d}^m}{\mathrm{d}\tau^m} y(\tau) \, \mathrm{d}\tau}{(\tau-t)^{\alpha-m+1}}.
$$
(27)

The Caputo formulation offers several advantages: it produces zero when applied to constant functions (matching classical derivatives), and it accommodates standard initial conditions in differential equations, making it particularly suitable for modeling physical systems. We therefore choose the Caputo formulation.

A fundamental characteristic distinguishing fractional derivatives from their integer-order counterparts is their inherent non-locality. The integral representations in (26) and (27) reveal that fractional derivatives incorporate weighted contributions from the function's entire history on the interval $[a, t]$ (for left-sided) or $[t, b]$ (for right-sided). This memory effect contrasts sharply with classical derivatives, which depend only on infinitesimal neighborhoods around the evaluation point. The weighting kernel $(t - \tau)^{\alpha-m+1}$ determines how past states influence the present derivative value, with the fractional order $\alpha$ controlling the decay rate of this historical influence.

This memory-dependent nature makes fractional derivatives particularly valuable for modeling systems with hereditary properties, long-range interactions, or anomalous diffusion phenomena. In the limiting case where $\alpha$ approaches an integer value, these fractional operators smoothly transition to their classical counterparts (Diethelm, 2010), establishing fractional calculus as a genuine generalization of traditional calculus. For instance, when $\alpha = n \in \mathbb{N}$, both Riemann-Liouville and Caputo derivatives reduce to the standard $n$-th order derivative, ensuring theoretical consistency and practical applicability across the entire spectrum of differentiation orders. When dealing with vector-valued functions, the fractional operators act independently on each component, in direct analogy to the multivariate extension of ordinary differentiation and integration.

## B.2 SURROGATE FUNCTIONS IN $f$-SNN

Training SNNs presents a fundamental challenge: the spiking function (Heaviside step function) is non-differentiable, making standard backpropagation impossible. To address this, similar to other works (Wu et al., 2018), we employ surrogate gradient methods that replace the undefined derivative of the step function with smooth approximations during the backward pass. In the main paper, we present the threshold-shifted sigmoid function. Our toolbox also implements other commonly used surrogate functions. We present them in this section.

For all surrogate functions, the forward pass computes the standard Heaviside step function:

$$H(x) = \begin{cases} 1, & \text{if } x \geq 0 \\ 0, & \text{if } x < 0 \end{cases} \tag{28}$$

The backward pass, however, replaces $H'(x)$ with a surrogate gradient $s(x)$. Below, we detail each surrogate function implemented in our toolbox `spikeDE`, which can be chosen freely by users.

### B.2.1 SIGMOID SURROGATE

The sigmoid surrogate uses the derivative of the scaled sigmoid function:

$$s_{\text{sigmoid}}(x) = \kappa \cdot \sigma(\kappa x) \cdot (1 - \sigma(\kappa x)) \tag{29}$$

where $\sigma(x) = \frac{1}{1+e^{-x}}$ is the sigmoid function and $\kappa$ is a scaling parameter (default: $\kappa = 5.0$). This surrogate provides smooth gradients centered around the threshold, with the scale parameter controlling the sharpness of the approximation.

### B.2.2 ARCTANGENT SURROGATE

The arctangent surrogate employs the derivative of the arctangent function:

$$s_{\text{arctan}}(x) = \frac{\kappa}{1 + (\kappa x)^2} \tag{30}$$

where $\kappa$ is the scale parameter (default: $\kappa = 2.0$). This function provides a bell-shaped gradient profile with heavier tails compared to the sigmoid surrogate, potentially allowing gradient flow for neurons further from the threshold.

### B.2.3 PIECEWISE LINEAR SURROGATE

The piecewise linear surrogate defines a simple triangular approximation:

$$s_{\text{linear}}(x) = \begin{cases} \frac{1}{2\gamma}, & \text{if } -\gamma \leq x \leq \gamma \\ 0, & \text{otherwise} \end{cases} \tag{31}$$

where $\gamma$ defines the width of the linear region (default: $\gamma = 1.0$). This surrogate provides constant gradients within a fixed window around the threshold, offering computational efficiency at the cost of gradient smoothness.

### B.2.4 GAUSSIAN SURROGATE

The Gaussian surrogate uses a normalized Gaussian function:

$$s_{\text{gaussian}}(x) = \frac{1}{\sigma\sqrt{2\pi}} \exp\left(-\frac{x^2}{2\sigma^2}\right) \tag{32}$$

where $\sigma$ is the standard deviation parameter (default: $\sigma = 1.0$). This surrogate provides the smoothest gradient profile with exponential decay away from the threshold.

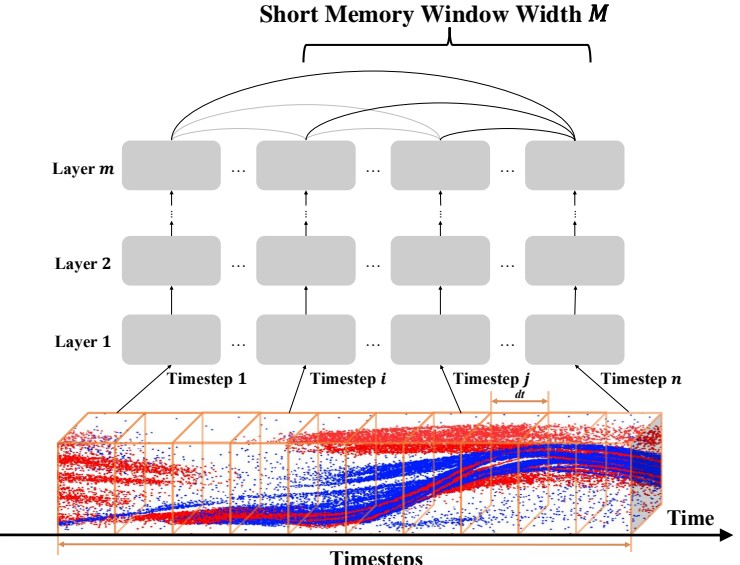

Figure 7: $f$-SNN illustration. Within each layer, the current update aggregates all past states of that layer. For efficiency, the short-memory principle approximates the fractional dynamics by retaining only the last $M$ timesteps.

### B.2.5 SURROGATE GRADIENT IMPLEMENTATION

In practice, during backpropagation through a spiking layer, the gradient of the loss $\mathcal{L}$ with respect to the membrane potential $u$ is computed as:

$$\frac{\partial \mathcal{L}}{\partial u} = \frac{\partial \mathcal{L}}{\partial s} \cdot s(u) \tag{33}$$

where $s$ represents the spike output and $s(u)$ is the chosen surrogate gradient function. The choice of surrogate function and its hyperparameters impacts training dynamics, with sharper surrogates (larger scale parameters) providing more precise threshold behavior but potentially suffering from vanishing gradients.

### B.3 MORE ABOUT FRAMEWORK AND ITS VISUALIZATION

In the main paper, we illustrate the distinct information flow characteristics of $f$-SNN compared to traditional SNNs in Fig. 1. In conventional SNNs, the iterative nature results in skip connections. In contrast, $f$-SNN utilizes dense connections, which arise from the weighted summation within the ABM predictor, as described in (13). The diagram can be seen in Fig. 7. Within each layer, the current update aggregates all past states of that layer. For efficiency, the short-memory principle approximates the fractional dynamics by retaining only the last $M$ timesteps.

The coefficients $c_m^{(\alpha)} = \frac{1}{\tau^\alpha \alpha \Gamma(\alpha)} \left[ (m+1)^\alpha - m^\alpha \right]$ define a causal memory kernel that decays according to a power-law, capturing the historical influence of the system. As depicted in Fig. 8, this decay is "heavy-tailed", meaning that although recent states dominate the influence, the contributions from past events remain significant over time, in contrast to the rapid decay for integer-order system.

For large $m$, the decay of input influence follows the relationship:

$$c_m^{(\alpha)} \propto (m+1)^\alpha - m^\alpha$$

Expanding this expression, we get:

$$c_m^{(\alpha)} \approx m^\alpha \left( \left(1 + \frac{1}{m}\right)^\alpha - 1 \right) \propto m^{\alpha-1},$$

for large $m$. This confirms that the kernel exhibits algebraic decay.

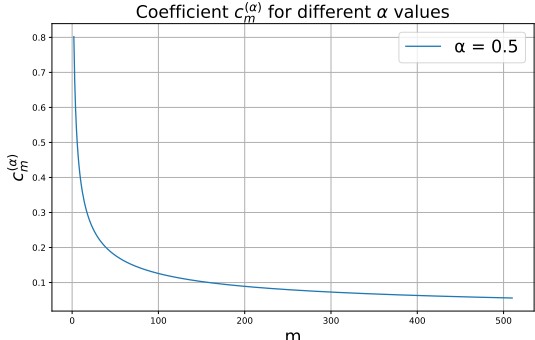

Figure 8: $f$-SNN coefficient $c_m^{(\alpha)}$ visuilization. $c_m^{(\alpha)}$ is causal and decays as a power law, explicitly encoding memory. This decay profile is "heavy-tailed," meaning that while recent states exert the strongest influence, distant past events retain a non-negligible impact compared to exponential decay.

## C  THEORETICAL RESULTS AND COMPLEXITY ANALYSIS

### C.1  PROOF OF PROPOSITION 1

*Proof.* For the first-order case, we multiply both sides of (5) by $e^{t/\tau}$, yielding

$$e^{t/\tau}\frac{dU}{dt} + \frac{1}{\tau}e^{t/\tau}U = \frac{R}{\tau}I_c e^{t/\tau}.$$

This simplifies to

$$\frac{d}{dt}\left(Ue^{t/\tau}\right) = \frac{R}{\tau}I_c e^{t/\tau}.$$

Integrating both sides with respect to $t$, we obtain

$$U(t)e^{t/\tau} = \int \frac{R}{\tau}I_c e^{t/\tau}\,dt + C = RI_c e^{t/\tau} + C,$$

where $C$ is the constant of integration. Solving for $U(t)$, we find

$$U(t) = RI_c + Ce^{-t/\tau}.$$

Applying the initial condition $U(0) = U_0$, we get $U_0 = RI_c + C$. So $C = U_0 - RI_c$. Therefore, the solution is

$$U(t) = RI_c + (U_0 - RI_c)\,e^{-t/\tau}.$$

For the fractional case, we take a more general Laplace transform approach. Applying the Laplace transform to both sides of (11), and using the property

$$\mathcal{L}\left\{{}^C D_t^\alpha U(t)\right\} = s^\alpha U(s) - s^{\alpha-1}U_0,$$

we obtain the transformed equation:

$$s^\alpha U(s) - s^{\alpha-1}U_0 + \frac{1}{\tau}U(s) = \frac{RI_c}{\tau}\cdot\frac{1}{s}.$$

Rearranging terms and solving for $U(s)$, we get:

$$U(s)\left(s^\alpha + \frac{1}{\tau}\right) = s^{\alpha-1}U_0 + \frac{RI_c}{\tau s}.$$

So we have

$$U(s) = \frac{s^{\alpha-1}U_0}{s^\alpha + \frac{1}{\tau}} + \frac{RI_c}{\tau}\cdot\frac{1}{s\left(s^\alpha + \frac{1}{\tau}\right)}.$$

To simplify the second term, observe the partial fraction identity:

$$\frac{1}{s\left(s^\alpha + \frac{1}{\tau}\right)} = \frac{\tau}{s} - \frac{\tau s^{\alpha-1}}{s^\alpha + \frac{1}{\tau}}.$$

Substituting this expression, we obtain:

$$U(s) = \frac{s^{\alpha-1}U_0}{s^\alpha + \frac{1}{\tau}} + RI_c\left(\frac{1}{s} - \frac{s^{\alpha-1}}{s^\alpha + \frac{1}{\tau}}\right) = \frac{RI_c}{s} + \frac{s^{\alpha-1}(U_0 - RI_c)}{s^\alpha + \frac{1}{\tau}}.$$

Taking the inverse Laplace transform and using the identity

$$\mathcal{L}^{-1}\left\{\frac{s^{\alpha-1}}{s^\alpha + a}\right\} = E_\alpha(-at^\alpha),$$

where $E_\alpha(\cdot)$ is the Mittag-Leffler function, we arrive at the time-domain solution:

$$U(t) = RI_c + (U_0 - RI_c)\, E_\alpha\left(-\frac{t^\alpha}{\tau}\right).$$

The power-law tail $E_\alpha\left(-t^\alpha/\tau_\alpha\right) \sim \frac{\tau_\alpha}{\Gamma(1-\alpha)t^\alpha}$ for large $t$ follows from (Diethelm, 2010)[Theorem 4.3.]. $\qquad\square$

## C.2 Proof of Theorem 1

In this section, we analyze the robustness of our $f$-SNN framework under input perturbations. We examine two critical aspects: (i) the temporal evolution of membrane potential perturbations, and (ii) the sensitivity of spike timing to input variations. Our analysis reveals that fractional-order dynamics exhibit distinct robustness properties compared to classical integer-order models.

To facilitate the analysis in this section, we choose the $F$-IF neuron dynamics with the continuous formulation (10):

$$\tau\, D^\alpha U(t) = R\, I_{\text{in}}(t).$$

From Diethelm, 2010[Lemma 6.2.], the above fractional-order ODE can be equivalently written as the following Volterra integral equation:

$$U(t) = U_0 + \frac{1}{\Gamma(\alpha)}\int_0^t (t-u)^{\alpha-1}\frac{R}{\tau}I_{\text{in}}(u)\mathrm{d}u \tag{34}$$

In the following, we consider a constant current input $I_{\text{in}}(t) \equiv I_c$ perturbed by a small deviation $\epsilon$, yielding $I_{\text{perb}}(t) = I_c + \epsilon$. Without loss of generality, we assume $U_0 = 0$.

• **Membrane Potential Robustness**

We denote by $U_{\text{clean}}(t)$ and $U_{\text{perb}}(t)$ the membrane potential under the clean input $I_c$ and perturbed input $I_c + \epsilon$, respectively. From (34), we obtain:

$$U_{\text{clean}}(t) = \frac{RI_c}{\tau\Gamma(\alpha)}\int_0^t (t-u)^{\alpha-1}\mathrm{d}u = \frac{RI_c}{\tau\Gamma(\alpha)}\left[\frac{t^\alpha}{\alpha}\right] = \frac{RI_c}{\tau\Gamma(\alpha+1)}t^\alpha \tag{35}$$

Similarly, we have:

$$U_{\text{perb}}(t) = \frac{R(I_c + \epsilon)}{\tau\Gamma(\alpha)}\int_0^t (t-u)^{\alpha-1}\mathrm{d}u = \frac{R(I_c + \epsilon)}{\tau\Gamma(\alpha+1)}t^\alpha$$

The difference between perturbed and unperturbed membrane potentials evolves as:

$$\Delta U^{f\text{-IF}}(t) = U_{\text{perb}}(t) - U_{\text{clean}}(t) = \frac{R\epsilon}{\tau\Gamma(\alpha+1)}t^\alpha$$

In contrast, for the classical IF model (limit of $\alpha \to 1$), we have:

$$\Delta U^{\text{IF}}(t) = \frac{R\epsilon}{\tau}t$$

Since $0 < \alpha < 1$, the perturbation growth follows $t^\alpha$ (sub-linear) for the fractional model versus $t$ (linear) for the IF model. This sub-linear accumulation demonstrates that fractional-order dynamics inherently suppress long-term accumulation of perturbation through memory effects encoded in the fractional derivative.

• **Spike Timing Sensitivity**

The first spike time occurs when the membrane potential reaches threshold $\theta$. We emphasize the necessity of robust spike timing, noting that various learning algorithms rely explicitly on these temporal values (Bohte et al., 2002; Booij & tat Nguyen, 2005; Rathi et al., 2019).

For the $f$-IF system, from (35), we have:

$$t_s^{\text{clean}} = \left( \frac{\theta \tau \Gamma(\alpha + 1)}{R I_c} \right)^{1/\alpha}$$

Under perturbation $I_c \to I_c + \epsilon$, the spike time changes to:

$$t_s^{\text{perb}} = \left( \frac{\theta \tau \Gamma(\alpha + 1)}{R(I_c + \epsilon)} \right)^{1/\alpha}$$

We define the spike time shift magnitude as $\Delta t_s = |t_s^{\text{clean}} - t_s^{\text{perb}}|$

$$\Delta t_s^{f\text{-IF}} = \left( \frac{\theta \tau \Gamma(\alpha + 1)}{R I_c} \right)^{1/\alpha} \left[ 1 - \left( \frac{I_c}{I_c + \epsilon} \right)^{1/\alpha} \right] \tag{36}$$

For small perturbations $\epsilon \ll I_c$, using Taylor expansion:

$$\left( \frac{I_c}{I_c + \epsilon} \right)^{1/\alpha} = \left( 1 + \frac{\epsilon}{I_c} \right)^{-1/\alpha}$$
$$= 1 + \left( -\frac{1}{\alpha} \right) \left( \frac{\epsilon}{I_c} \right) + \mathcal{O}(\epsilon^2)$$

The first-order approximation is therefore:

$$|\Delta t_s^{f\text{-IF}}| \approx \left( \frac{\theta \tau \Gamma(\alpha + 1)}{R I_c} \right)^{1/\alpha} \cdot \frac{\epsilon}{\alpha I_c}$$

For the classical IF model, following the same procedure, we obtain:

$$|\Delta t_s^{\text{IF}}| \approx \frac{\theta \tau}{R I_c} \cdot \frac{\epsilon}{I_c}$$

Examining the dependence on input current $I_c$, we observe that for the fractional model, the sensitivity decays more rapidly with increasing $I_c$, as the dependence is:

$$\Delta t_s^{f\text{-IF}} \propto \epsilon \cdot I_c^{-(1+1/\alpha)}$$

For the IF model, it decays as:

$$\left| \Delta t_s^{\text{IF}} \right| \propto \epsilon \cdot I_c^{-2}$$

Since $1 + 1/\alpha > 2$ for $0 < \alpha < 1$, this shows that the fractional model is more robust in terms of spike timing for high input currents.

These findings demonstrate that fractional-order dynamics introduce a nuanced robustness profile, with distinct advantages in specific operational regimes rather than universal superiority.

## C.3 PROOF OF THEOREM 2

In this section, we analyze the representational capacity of the $f$-SNN neuron compared to finite banks of standard integer-order SNN neurons. We demonstrate that fractional dynamics induce an infinite-memory structure that cannot be exactly realized by any finite linear combination of integer-order SNN neurons with arbitrary weights. This structure can only be represented as an infinite continuum of integer-order modes.

Let the discrete time index be $k \in \mathbb{Z}_{\geq 0}$. In the main paper, we use a fractional ABM predictor to discretize the $f$-SNN with Caputo derivative $D^\alpha$ for presentation. The literature offers various definitions of fractional derivatives. To facilitate the theoretical analysis in this section, we instead utilize the Grünwald-Letnikov (GL) definition (Diethelm, 2010) here. We emphasize that our $f$-SNN toolbox **spikeDE** supports multiple fractional definitions, including Caputo, GL, and others. The analysis here does not restrict the implementation.

**Definition 2** (Grünwald-Letnikov Fractional Derivative). *For a function $y(t)$ defined over an interval $[0, T]$, its Grünwald-Letnikov fractional derivative of order $\alpha \in (0, 1]$ is given by (Diethelm, 2010):*

$$D_{\mathrm{GL}}^\alpha y(t) := \lim_{h \to 0} \frac{1}{h^\alpha} \sum_{j=0}^{\lfloor \frac{t}{h} \rfloor} (-1)^j \binom{\alpha}{j} y(t - jh). \tag{37}$$

The GL weights are defined as:

$$c_j^{(\alpha)} = (-1)^j \binom{\alpha}{j} \tag{38}$$

Recall the fractional-order ODE from (3): $D_{\mathrm{GL}}^\alpha y(t) = f(t, y(t))$. Approximating $D_{\mathrm{GL}}^\alpha y(t_k)$ by the finite GL sum gives:

$$\sum_{j=0}^k c_j^{(\alpha)} y_{k-j} = h^\alpha f(t_{k-1}, y_{k-1}).$$

Without loss of generality, for the neurons, we set $\tau = 1$, $h = 1$, and $R = 1$ to simplify the analysis (the constants here just rescale things and can be absorbed into input $X$). For simplicity, we primarily consider the $f$-IF neuron to demonstrate that any finite linear combination of integer-order LIF neurons with arbitrary weights cannot exactly realize it. Noting that $c_0^{(\alpha)} = 1$, the discrete-time membrane potential update for the $f$-IF, based on the GL definition, is given by:

$$\sum_{j=0}^k c_j^{(\alpha)} U_{k-j}^{f\text{-IF}} = X_k. \tag{39}$$

where $U_{k-j}^{f\text{-IF}}$ is the membrane potential of $f$-IF neuron.

Recall a standard discrete-time LIF neuron with a leak factor $\beta \in (0, 1)$ is governed by the recurrence:

$$U_k^{\mathrm{LIF}(\beta)} = \beta U_{k-1}^{\mathrm{LIF}(\beta)} + X_k, \quad U_0^{\mathrm{LIF}(\beta)} = 0$$

By unrolling the recurrence, we obtain the solution

$$U_k^{\mathrm{LIF}(\beta)} = \sum_{m=0}^{k-1} \beta^m X_{k-m} = \left( h^{(\beta)} * X \right)_k$$

where the impulse response is the geometric sequence

$$h_m^{(\beta)} = \beta^m, \quad m \geq 0$$

We now consider a finite bank of integer-order neurons with leak factors $\{\beta_i\}_{i=1}^W \subset (0, 1]$ and readout weights $\{\phi_i\}_{i=1}^W \subset \mathbb{R}$. The aggregate output is denoted as:

$$\hat{U}_k = \sum_{i=1}^W \phi_i U_k^{\mathrm{LIF}(\beta_i)}$$

**Theorem 3** (Playback of Theorem 2). *Let $U^{f\text{-IF}}$ denote the trajectory of a $f$-IF neuron with order $\alpha \in (0, 1)$. There exist no finite integer $W$, weights $\{\phi_i\}_{i=1}^W$, and leak factors $\{\beta_i\}_{i=1}^W$ such that the following holds:*

$$\hat{U}_k = \sum_{i=1}^W \phi_i U_k^{\mathrm{LIF}(\beta_i)} \equiv U_k^{f\text{-IF}} \quad \forall k.$$

*for general input $X_k$. The impulse response error of the approximation is $O(k^{\alpha-1})$, decaying algebraically slowly. The $f$-IF neuron is mathematically equivalent to an aggregate of integer-order LIF neurons if and only if $W \to \infty$, specifically as an integral over a continuum of leak factors.*

**Remark 6.** *The result above emphasizes the infinite-dimensional nature of the f-IF neuron compared to any finite-dimensional approximation via integer-order neurons. Intuitively, the impulse response of a standard integer-order neuron decays exponentially ($\beta^k = e^{-\lambda k}$), characterizing a "short-memory" process. In contrast, the fractional neuron exhibits an impulse response that decays according to a power law ($k^{\alpha-1}$), characterizing a "long-memory" process. A finite sum of exponentials can never exactly match a power law tail.*

*Proof.* Recall from (39) that:

$$\sum_{j=0}^{k} c_j^{(\alpha)} U_{k-j}^{f\text{-IF}} = X_k,$$

which represents a discrete convolution $(c^{(\alpha)} * U^{f\text{-IF}})_k = X_k$. Applying the $\mathcal{Z}$-transform converts this convolution into the algebraic product:

$$C(z)U(z) = X(z),$$

where $C(z) = \sum_{j=0}^{\infty} c_j^{(\alpha)} z^{-j}$ is the $\mathcal{Z}$-transform of the kernel, and $U(z)$ and $X(z)$ are the $\mathcal{Z}$-transforms of $U_k^{f\text{-IF}}$ and $X_k$, respectively.

Using the generalized binomial theorem, we identify the series expansion of the kernel as:

$$C(z) = \sum_{j=0}^{\infty} \binom{\alpha}{j}(-z^{-1})^j = (1 - z^{-1})^\alpha.$$

Thus, the transfer function $H_\alpha^{f\text{-IF}}(z) = \frac{U(z)}{X(z)} = \frac{1}{C(z)}$ is the reciprocal of the kernel:

$$H_\alpha^{f\text{-IF}}(z) = (1 - z^{-1})^{-\alpha}.$$

Since $(1 - z^{-1})^{-\alpha}$ has an algebraic branch point at $z = 1$ for $\alpha \notin \mathbb{Z}$, it is non-rational.

By the linearity of the $\mathcal{Z}$-transform, the aggregate transfer function $\widehat{H}(z)$ is the weighted sum of the individual transfer functions. Since the impulse response of the $i$-th LIF neuron is the geometric sequence $h_m^{(\beta_i)} = \beta_i^m$, its $\mathcal{Z}$-transform is the standard geometric series:

$$H_i(z) = \sum_{m=0}^{\infty} \beta_i^m z^{-m} = \frac{1}{1 - \beta_i z^{-1}}.$$

Therefore, the aggregate transfer function is:

$$\widehat{H}(z) = \sum_{i=1}^{W} \phi_i H_i(z) = \sum_{i=1}^{W} \frac{\phi_i}{1 - \beta_i z^{-1}}.$$

This is strictly a rational function of $z$ with degree at most $W$. A discrete-time linear time-invariant (LTI) system has a finite-dimensional state-space realization if and only if its transfer function is rational. Since $H_\alpha^{f\text{-IF}}(z)$ is irrational, it implies that the system possesses an infinite-dimensional state space and cannot be realized by any finite $W$.

The rigorous link between the the transfer functions is established through the asymptotic decay rates of the impulse responses. The impulse response of the finite bank, $h_k^{\text{finite}}$, is the inverse $\mathcal{Z}$-transform of the rational function $\widehat{H}(z)$. This yields a linear combination of geometric sequences:

$$h_k^{\text{finite}} = \mathcal{Z}^{-1}\left\{ \sum_{i=1}^{W} \frac{\phi_i}{1 - \beta_i z^{-1}} \right\} = \sum_{i=1}^{W} \phi_i \beta_i^k.$$

Let $\beta_{\max} = \max_i |\beta_i|$. Since stability requires $|\beta_i| < 1$, the decay is bounded exponentially:

$$|\hat{h}_k| \leq \left( \sum_{i=1}^{W} |\phi_i| \right) \beta_{\max}^k. \tag{40}$$

In contrast, the impulse response of the fractional neuron corresponds to the coefficients of the irrational function $(1 - z^{-1})^{-\alpha}$. By the generalized binomial theorem, for $|z^{-1}| < 1$,

$$\left(1 - z^{-1}\right)^{-\alpha} = \sum_{k=0}^{\infty} (-1)^k \binom{-\alpha}{k} z^{-k} = \sum_{k=0}^{\infty} \frac{\Gamma(k+\alpha)}{\Gamma(\alpha)\Gamma(k+1)} z^{-k}.$$

Thus the coefficient of $z^{-k}$ is

$$h_k^{f\text{-IF}} = \frac{\Gamma(k+\alpha)}{\Gamma(\alpha)\Gamma(k+1)}$$

Using the asymptotic property of the ratio of Gamma functions, this response follows a power-law decay:

$$h_k^{f\text{-IF}} \sim \frac{1}{\Gamma(\alpha)} k^{\alpha-1} \quad \text{as } k \to \infty. \tag{41}$$

Comparing the exponential decay of the finite bank and the power-law decay of the fractional neuron, we observe that the exponential decay is strictly faster than the algebraic decay. Hence, for any fixed $W$, there exists a time step $T^*$ such that for all $k > T^*$, the finite approximation becomes negligible relative to the fractional signal:

$$\lim_{k \to \infty} \left| \frac{h_k^{\text{finite}}}{h_k^{f\text{-IF}}} \right| = 0.$$

This implies that for sufficiently large $k$, the finite approximation $\hat{h}_k$ becomes negligible relative to $h_k^{f\text{-IF}}$. Specifically, there exists a time $T^*$ such that for all $k > T^*$, $|\hat{h}_k| < \frac{1}{2}|h_k^{f\text{-IF}}|$. Consequently, the pointwise error at the tail is lower-bounded:

$$|\hat{h}_k - h_k^{f\text{-IF}}| \geq |h_k^{f\text{-IF}}| - |\hat{h}_k| > \frac{1}{2}|h_k^{f\text{-IF}}| \sim \frac{1}{2\Gamma(\alpha)} k^{\alpha-1}.$$

Thus, the "heavy-tailed" memory induced by the irrational transfer function cannot be captured by the "light-tailed" exponential memory of any finite rational system.

Although no finite realization exists, we show that an infinite representation is valid. Use the Beta-function identity

$$B(m+\alpha, 1-\alpha) = \int_0^1 t^{m+\alpha-1}(1-t)^{-\alpha}dt = \frac{\Gamma(m+\alpha)\Gamma(1-\alpha)}{\Gamma(m+1)}.$$

Divide both sides by $\Gamma(\alpha)\Gamma(1-\alpha) = B(\alpha, 1-\alpha)$ to obtain

$$\frac{\Gamma(m+\alpha)}{\Gamma(\alpha)\Gamma(m+1)} = \int_0^1 t^m \frac{t^{\alpha-1}(1-t)^{-\alpha}}{B(\alpha, 1-\alpha)}dt.$$

Hence, with

$$d\nu_\alpha(\beta) := \frac{\beta^{\alpha-1}(1-\beta)^{-\alpha}}{B(\alpha, 1-\alpha)}d\beta$$

we have

$$h_m^{f\text{-IF}} = \int_0^1 \beta^m d\nu_\alpha(\beta).$$

Therefore, there exists a spectral measure $\nu_\alpha$ supported on $[0, 1]$ such that:

$$U_k^{f\text{-IF}} = \sum_{m=0}^{k-1} \left( \int_0^1 \beta^m d\nu_\alpha(\beta) \right) X_{k-1-m}$$

By Fubini's theorem (the sum and integral are both over non-negative indices and the integrand is non-negative), we may exchange the summation and integration:

$$U_k^{f\text{-IF}} = \int_0^1 \left( \sum_{m=0}^{k-1} \beta^m X_{k-1-m} \right) d\nu_\alpha(\beta) = \int_0^1 U_k^{\text{LIF}(\beta)} d\nu_\alpha(\beta)$$

where $U_k^{\text{LIF}(\beta)}$ is the state of a LIF neuron with leak factor $\beta$.

This establishes that the fractional neuron state $U_k^{f-\text{IF}}$ is the aggregate output of a population of LIF neurons $U_k^{\text{LIF}(\beta_i)}$, distributed over the leak factor $\beta_i \in (0, 1]$ according to the density $\nu_\alpha$. In other words, the fractional neuron state is a continuous mixture of integer-order LIF neurons. $\qquad\square$

## C.4 COROLLARY: SPIKE TRAIN IRREDUCIBILITY

Theorem 2 establishes irreducibility at the membrane potential level, but the functional output of a spiking neuron is the spike train. We now clarify the connection between membrane potential dynamics and spike output expressivity.

**Clarification: From Membrane Potential to Spike Output**

The spike generation mechanism directly couples membrane potential to output: a spike is emitted when $U_k \geq \theta$ (threshold). Therefore, the spike train $S_k \in \{0, 1\}$ is a deterministic function of the membrane trajectory:

$$S_k = H\left(U_k - \theta\right)$$

where $H(\cdot)$ is the Heaviside step function. **This coupling implies that differences in membrane potential dynamics propagate to differences in spike train patterns.**

We formalize this connection with the following corollary:

**Corollary 1** (Spike Train Irreducibility). *Let $0 < \alpha < 1$. For any finite integer $W$, weights $\{\phi_i\}_{i=1}^{W} \subset \mathbb{R}$, leak factors $\{\beta_i\}_{i=1}^{W} \subset (0, 1)$, and any Boolean function $f : \{0, 1\}^W \to \{0, 1\}$, there exists an input sequence $\{X_k\}_{k \geq 0}$ and threshold $\theta > 0$ such that the spike train of the $f$-IF neuron cannot be reproduced by any Boolean combination of spike trains from the $W$ LIF neurons. That is,*

$$S_k^{f\text{-IF}} \neq f\left(S_k^{\text{LIF}(\beta_1)}, \ldots, S_k^{\text{LIF}(\beta_W)}\right) \quad \text{for some } k. \tag{42}$$

**Remark 7.** *This corollary establishes that the expressivity advantage of $f$-IF neurons extends to the spike train level. While Theorem 2 demonstrates irreducibility in membrane potential dynamics, one might wonder whether this advantage could be "washed out" by the thresholding nonlinearity. The corollary confirms it is not: the spike patterns generated by a single $f$-IF neuron encode temporal information that no finite ensemble of LIF neurons can reproduce, even when their outputs are combined through arbitrary Boolean logic. This implies that $f$-SNNs possess fundamentally richer spike-based representations, enabling them to communicate long-range temporal dependencies through their spike trains in ways that conventional SNNs cannot.*

*Proof.* We prove this by construction using an *impulse-silent-trigger* sequence. This construction explicitly demonstrates the role of long-range temporal memory.

**Step 1: Input design.** Define the input sequence as

$$X_k = \begin{cases} A & \text{if } k = 0, \\ 0 & \text{if } 1 \leq k \leq T - 1, \\ \delta & \text{if } k = T, \end{cases} \tag{43}$$

where $A, \delta > 0$ are chosen appropriately, and $T$ is a large delay parameter.

**Step 2: Membrane potential at detection time $T$.** At time $T$, the membrane potential comprises two components: (i) the memory of the initial impulse $A$, decayed over $T$ time steps, and (ii) the immediate response to the test pulse $\delta$.

For $f$-IF:
$$U_T^{f\text{-IF}} = A \cdot h_T^{f\text{-IF}} + \delta \cdot h_0^{f\text{-IF}} = A \cdot h_T^{f\text{-IF}} + \delta, \tag{44}$$

where $h_T^{f\text{-IF}} \sim T^{\alpha-1}/\Gamma(\alpha)$ exhibits power-law decay.

For the finite LIF ensemble:
$$\hat{U}_T = A \cdot \hat{h}_T + \delta, \tag{45}$$

where $\hat{h}_T = \sum_{i=1}^{W} \phi_i \beta_i^T$.

**Step 3: Asymptotic separation.** Define $M = \sum_{i=1}^{W} |\phi_i|$ and $\beta_{\max} = \max_i |\beta_i| < 1$. Then

$$|\hat{h}_T| \leq M \beta_{\max}^T. \tag{46}$$

The ratio of memory contributions satisfies

$$\frac{|\hat{h}_T|}{h_T^{f\text{-IF}}} \leq \frac{M\beta_{\max}^T \cdot \Gamma(\alpha)}{T^{\alpha-1}} \xrightarrow{T\to\infty} 0, \tag{47}$$

since exponential decay dominates power-law decay.

**Step 4: Parameter selection.** For any fixed finite ensemble $(W, \{\phi_i\}, \{\beta_i\})$, choose $T$ sufficiently large such that

$$|\hat{h}_T| < \tfrac{1}{4} h_T^{f\text{-IF}}. \tag{48}$$

Set the remaining parameters as follows:

- $A = 1$,
- $\delta = \tfrac{1}{2} h_T^{f\text{-IF}}$,
- $\theta = \tfrac{3}{4} h_T^{f\text{-IF}} + \delta = \tfrac{5}{4} h_T^{f\text{-IF}}$.

**Step 5: Spike analysis.** We analyze the spike behavior at two critical times.

*At time $T$:* For $f$-IF,

$$U_T^{f\text{-IF}} = h_T^{f\text{-IF}} + \delta = \tfrac{3}{2} h_T^{f\text{-IF}} > \theta, \tag{49}$$

so the $f$-IF neuron spikes: $S_T^{f\text{-IF}} = 1$.

For each LIF neuron in the ensemble,

$$|\hat{U}_T| \leq |\hat{h}_T| + \delta < \tfrac{1}{4} h_T^{f\text{-IF}} + \tfrac{1}{2} h_T^{f\text{-IF}} = \tfrac{3}{4} h_T^{f\text{-IF}} < \theta, \tag{50}$$

so all LIF neurons are silent: $S_T^{\text{LIF}(\beta_i)} = 0$ for all $i \in \{1, \dots, W\}$.

*At times $0 < k < T$:* The $f$-IF membrane potential satisfies

$$U_k^{f\text{-IF}} = A \cdot h_k^{f\text{-IF}} \leq A \cdot h_0^{f\text{-IF}} = 1 < \theta, \tag{51}$$

so the $f$-IF neuron is silent: $S_k^{f\text{-IF}} = 0$. Similarly, all LIF neurons remain silent during this interval.

**Step 6: Establishing a mismatch for any Boolean function.** At time $T$, all LIF neurons are silent, so

$$f\big(S_T^{\text{LIF}(\beta_1)}, \dots, S_T^{\text{LIF}(\beta_W)}\big) = f(0, \dots, 0). \tag{52}$$

We consider two cases based on the value of $f(0, \dots, 0)$:

*Case 1:* If $f(0, \dots, 0) = 0$, then at time $T$,

$$S_T^{f\text{-IF}} = 1 \neq 0 = f(0, \dots, 0). \tag{53}$$

*Case 2:* If $f(0, \dots, 0) = 1$, then at any time $0 < k < T$, all neurons are silent, so

$$S_k^{f\text{-IF}} = 0 \neq 1 = f(0, \dots, 0). \tag{54}$$

In both cases, the spike trains differ for some $k$, completing the proof. $\qquad\square$

## C.5 COMPLEX ANALYSIS

In the iteration of (13) using the ABM predictor (12), at each time-step $t_j$ we must evaluate the fractional derivative $f(t_j, y_j)$ which is $R\,I_{\text{in}}(t)$ in $f$-IF and $-U(t) + R\,I_{\text{in}}(t)$ in $f$-LIF neuron. If there are $N = T/h$ steps in total, then summing the base cost $C$ per evaluation at each time step for all layers with the growing cost $O(k)$ of accumulating $k$-term histories yields $\sum_{k=0}^{N}(C + O(k)) = O(NC + N^2)$. By leveraging a fast convolution routine (e.g. the FFT-based method of (Mathieu et al., 2013)), the quadratic term can be reduced to $O(N \log N)$, giving an overall forward-pass cost of $O(NC + N \log N)$. Since each step also stores its hidden state vector of dimension $d$, and computing $f$ at one timestamp incurs a peak memory $P$, the forward memory requirement grows as $O(P + Nd)$, where the $Nd$ term accounts for saving all $N$ evaluations $\{f(t_j, y_j)\}_{j=1}^{N}$. Finally, note that one can trim the $O(Nd)$ storage of all past $f(t_j, y_j)$ values down to $O(Md)$ by invoking the "short-memory" approximation, keeping only the most recent $M$ terms in the iterations. The experimental computational complexity is presented in Section D.2.3.

# D  IMPLEMENTATION DETAILS, DATASET SPECIFICS AND MORE EXPERIMENTS

## D.1  EXPERIMENT SETTINGS

### D.1.1  GRAPH LEARNING TASKS.

**Datasets & Baselines.** Our experiments evaluate model performance on a diverse collection of graph-structured datasets spanning multiple domains, following standard preprocessing protocols in geometric deep learning, with their detailed statistics summarized in Table 3. Specifically, node classification is performed with SGCN and DRSGNN on Cora, Citeseer, Pubmed, Photo, Computers, and ogbn-arxiv. Additionally, we conduct link prediction experiments with MSG-based methods (Sun et al., 2024) on Computers, Photo, CS, and Physics, with results presented in the following section Section D.2.1.

Table 3: Dataset Statistics of Node Classification Task

| Name | # of Nodes | # of Classes | # of Features | # of Edges |
|---|---|---|---|---|
| Cora | 2,708 | 7 | 1,433 | 10,556 |
| Pubmed | 19,717 | 3 | 500 | 88,648 |
| Citeseer | 3,327 | 6 | 3,703 | 9,104 |
| Photo | 7,650 | 8 | 745 | 238,162 |
| Computers | 13,752 | 10 | 767 | 491,722 |
| OGBN-Arxiv | 169,343 | 40 | 128 | 1,166,243 |
| CS | 18,333 | 15 | 6,805 | 163,788 |
| Physics | 34,493 | 5 | 8,415 | 495,924 |

**Training& Inference settings.** For node classification tasks based on SGCN and DRSGNN, we use Poisson encoding to generate spike data following a Poisson distribution. The number of timesteps $N$ is set to 100, and the batch size is set to 32. The dataset is divided into training, validation, and test sets in the ratio of $0.7, 0.2$ and $0.1$. Additionally, for experiments based on DRSGNN, we set the dimension of the positional encoding to 32 and adopt the Laplacian eigenvectors (LSPE) (Dwivedi et al., 2023) or random walk (RWPE) (Dwivedi et al., 2021) method. For link prediction tasks on MSG, we follow the experimental settings described in the original MSG paper. Specifically, we use IF neurons and the Lorentz model, set the dimension of the representation space to 32, and configure the timesteps to $[5, 15]$. The optimizer used is Adam, with an initial learning rate of 0.001. All experiments are independently run 20 times, with the mean and standard deviation reported.

Table 4: The selection of the parameter $\alpha$, we report the values that yield the best results on each dataset.

| Methods | Cora | Citeseer | Pubmed | Photo | Computers | ogbn-arxiv |
|---|---|---|---|---|---|---|
| SGNN ($f$-SNN) | 0.3 | 0.3 | 0.9 | 1.0 | 0.8 | 1.0 |
| DRSGNN ($f$-SNN) | 0.3 | 0.3 | 0.8 | 1.0 | 0.8 | 1.0 |

**Selection of Different $\alpha$ Values.** Table 4 shows the parameter $\alpha$ we chose in the graph learning tasks. The corresponding results are shown in Table 2.

### D.1.2  NEUROMORPHIC DATA CLASSIFICATION TASKS.

**Datasets & Baselines.** We conduct experiments on five visual classification tasks, including N-MNIST (Orchard et al., 2015), DVS128Gesture (Amir et al., 2017), N-Caltech101 (Orchard et al., 2015), DVS-Lip (Tan et al., 2022), and the large-scale dataset HarDVS (Wang et al., 2024). N-MNIST is a dataset that converts the classic MNIST handwritten digit dataset into neuromorphic event data. It generates event streams by observing moving MNIST digit images on a screen through a Dynamic Vision Sensor (DVS), covering 10 digit categories (0-9). DVS128Gesture is a neuromorphic

dataset collected using a Dynamic Vision Sensor (DVS) event camera, designed for dynamic gesture recognition tasks. The event camera captures pixel-level brightness changes with microsecond temporal resolution. The dataset includes 1,342 samples across 11 gesture categories, such as clockwise rotation, counterclockwise rotation, and left-hand waving. N-Caltech101 is an event camera-based dataset derived from the traditional static image dataset Caltech101. The original Caltech101 dataset contains 101 object categories (e.g., animals, vehicles, household items), with each category containing 40 to 800 static images. By simulating translational, rotational, and other movements, event cameras dynamically capture these static images to generate the corresponding neuromorphic data. Similar to DVS128Gesture, the data from N-Caltech101 is represented as event streams, which encode pixel-wise brightness changes over time. The classification task focuses on recognizing object categories in dynamic scenes. DVS-Lip is an event camera dataset specifically designed for lip reading, utilizing the high temporal resolution characteristics of event cameras to record fine-grained changes in lip movements. HarDVS is a large-scale human action recognition event dataset containing over 100,000 temporal event stream samples, covering 300 different human activity categories.

To ensure fairness and comparability, we employ two backbone network architectures, CNN and Transformer, on the same datasets, differing only in the choice of spiking neuron modules. The baseline methods use the `neuron.LIFNode` module from the SpikingJelly framework (Fang et al., 2023a) and the `snn.leaky` module from the snnTorch framework (Eshraghian et al., 2023a), respectively, while our method adopts the $f$-LIF neuron module (13) defined in $f$-SNN. The CNN-based-SNN network architecture follows the design of DVSNet in SpikingJelly, while the Transformer-based-SNN uses the structure of Spikformer (Zhou et al., 2022) as the backbone network.

**Training& Inference settings.** For the N-MNIST dataset, we set the batch size to 512, the number of timesteps $T$ to 16, and use the Adam optimizer to train for 100 epochs.

For other neuromorphic datasets, we follow the standard preprocessing pipeline of the Spiking-Jelly framework, converting event data into frame representations. For timesteps configuration, DVS128Gesture, N-Caltech101, and DVS-Lip are set to 16 timesteps, while HarDVS is set to 8 timesteps. N-Caltech101 is split into training and test sets with an 8:2 ratio. All neuromorphic datasets use a batch size of 16, with the input size uniformly adjusted to 128×128 pixels. For CNN-based models, training is conducted using the Adam optimizer for 200 epochs, while Transformer-based models are trained for 500 epochs.

### D.1.3 ROBUSTNESS ANALYSIS SETTING

We validate the robustness advantages of $f$-SNN in Section 4.1. Specifically, we comprehensively test the model's stability from five dimensions: noise injection, occlude block, temporal truncate, temporal jitter, and discard frame.

**Noise Injection:** Real-world applications often encounter sensor noise and environmental interference, which challenges the model's stability under noisy conditions. To evaluate this robustness, we randomly add Gaussian noise of different intensities to the input spike sequences.

**Occlude Block:** Real-world scenarios frequently involve occlusion and partial field-of-view loss, testing the model's ability to handle incomplete visual information. Accordingly, we place square blocks of different sizes at the center of input frames to assess the model's robustness to local information loss.

**Temporal Truncate:** Practical data collection often results in incomplete temporal information due to various constraints, challenging the model's performance with partial temporal data. Thus, we randomly truncate a portion of temporal data by proportion to evaluate the model's adaptability to incomplete sequences.

**Temporal Jitter:** Real systems often suffer from temporal synchronization errors and clock drift, affecting the precise timing of spike events. To simulate these timing uncertainties, we add random temporal offsets to spike events in the time dimension.

**Discard Frame:** Data transmission and processing systems frequently experience packet loss and intermittent data missing, testing the model's tolerance to discontinuous input. Consequently, we randomly discard partial frames in the temporal sequence to evaluate the model's robustness to data loss.

## D.2 Extended Experimental Results

### D.2.1 Link Prediction.

We have demonstrated the effectiveness of the $f$-SNN framework on graph node classification. Here, we present additional results for the graph *link-prediction* task using the graph SNN MSG (Sun et al., 2024). Our fractional adaptation, MSG ($f$-SNN), consistently outperforms MSG (SJ) across all datasets in terms of area under the ROC curve (AUC). Specifically, MSG ($f$-SNN) achieves the best AUC on the Computers, Photo, CS, and Physics datasets, with scores of 94.91%, 96.80%, 96.53%, and 96.57%, respectively.

Table 5: Link prediction results in terms of Area Under Curve (AUC) (%) on multiple datasets. The best results are **boldfaced**.

| Methods | Computers | Photos | CS | Physics |
|---|---|---|---|---|
| MSG (SJ) | $94.65_{\pm 0.73}$ | $96.75_{\pm 0.18}$ | $95.19_{\pm 0.15}$ | $93.43_{\pm 0.16}$ |
| MSG ($f$-SNN) | $\mathbf{94.91_{\pm 0.12}}$ | $\mathbf{96.80_{\pm 0.16}}$ | $\mathbf{96.53_{\pm 0.15}}$ | $\mathbf{96.57_{\pm 0.08}}$ |

As shown in Table 5, compared to the MSG method implemented with the integrator-based approach (SpikingJelly), our $f$-SNN achieves significant improvements in link prediction tasks. These results underscore the effectiveness of $f$-SNN across diverse datasets.

### D.2.2 Ablation Studies, Choice of Numerical Schemes and Parameters

In this section, we will conduct ablation experiments on various hyperparameters in $f$-SNN, including the selection of different order $\alpha$ values, whether to set learnable $\alpha$, whether to set learnable neuron thresholds, different timestamps $T$, different network parameters, and different neuron selections. All experiments, unless otherwise specified, are tested on **CNN-based SNN** and the **DVS128Gesture** dataset.

**Selection of Different $\alpha$ Values:** In our $f$-SNN architecture, different $\alpha$ values can impact experimental results. We evaluate the network performance under various $\alpha$ value conditions.

| $\alpha$ | 0.2 | 0.4 | 0.5 | 0.6 | 0.8 | 1.0 | Learnable |
|---|---|---|---|---|---|---|---|
| Acc1 | 0.9336 | 0.9236 | 0.9480 | 0.9193 | 0.9143 | 0.9340 | 0.9362 (Final $\alpha$: 0.5083) |

Table 6: Ablation study on different $\alpha$ values

The experimental results demonstrate that the network does not achieve optimal performance when $\alpha = 1$, which further validates the superior performance of $f$-SNN compared to conventional SNN. Notably, setting $\alpha$ as a learnable parameter also yields promising results. Although the accuracy is slightly lower than manual hyperparameter tuning, it still outperforms the case when $\alpha = 1$. Moreover, the converged final $\alpha$ value closely approximates our manually fine-tuned result, indicating the effectiveness of the learnable parameter approach.

**Different Network Parameters** To verify the experimental effectiveness of our $f$-SNN model on larger parameter models under different timesteps, we tested the performance of the $f$-SNN model under different parameters and different $T$ values. Since the performance on DVS128Gesture has already approached saturation, we conducted ablation experiments on **N-Caltech101**.

| | Channel 128 (1.7M) | Channel 256 (4.5M) | Channel 512 (13.7M) |
|---|---|---|---|
| LIF(SpikingJelly) | 0.6682 | 0.7053 | 0.7108 |
| LIF(snnTorch) | 0.6521 | 0.6765 | 0.7423 |
| $f$-LIF($f$-SNN) | 0.7026 | 0.7416 | 0.7684 |

Table 7: Performance comparison under different network parameters on **N-Caltech101**

It can be seen that with increased parameter count, our $f$-SNN consistently maintains leading performance, validating the excellent performance of $f$-SNN.

### D.2.3 STATIC DATASET TESTING

In addition to testing on neuromorphic datasets, we also evaluated our method on traditional static datasets, including **CIFAR-10**, **CIFAR-100** (Krizhevsky & Hinton, 2009), and **ImageNet** (Krizhevsky et al., 2012). For the relatively smaller **CIFAR-10** and **CIFAR-100** datasets, we adopt Spiking-ResNet-18 (Fang et al., 2021) as the baseline. For ImageNet, we follow the SpikFormer (Zhou et al., 2022) configuration with 29.7M parameters and set both the training and validation image sizes to $160 \times 160$ for a fair comparison. Additionally, we include experiments with ImageNet generated using the Beornil spike encoder. The results are shown in Table 8. Our $f$-LIF achieves clear gains over both LIF **SpikingJelly** and LIF **snnTorch** on all datasets.

Table 8: Comparison of LIF variants across datasets and architectures.

| Datasets | Architecture | Timesteps | LIF (SJ) | LIF (snnTorch) | $f$-LIF ($f$-SNN) |
|---|---|---|---|---|---|
| CIFAR-10 | Spiking-ResNet-18 | 4 | 0.9134 | 0.9026 | **0.9215** |
| CIFAR-100 | Spiking-ResNet-18 | 4 | 0.6813 | 0.6445 | **0.6874** |
| ImageNet | SpikFormer | 4 | 0.6637 | 0.6584 | **0.6791** |
| ImageNet (spike encoder) | SpikFormer | 4 | 0.5549 | 0.5432 | **0.5738** |

As shown in the results, our proposed $f$-SNN demonstrates superior performance on static datasets as well.

**Different Neuron Type.** To validate the effectiveness of $f$-LIF neurons, we further compare the performance of different neuron types. In addition to $f$-LIF neurons, we also test $f$-IF neurons and traditional IF neurons (SJ) under the same network architecture. The experimental results are shown in Table 9. On the DVS128Gesture dataset, the CNN-based-SNN with $f$-IF neurons achieves an accuracy of 93.83%, while the version using traditional IF neurons achieves 92.70%, representing a 1.13 percentage point improvement for $f$-IF neurons over traditional IF neurons. On the N-Caltech101 dataset, $f$-IF neurons also demonstrate significant advantages, achieving an accuracy of 69.23% compared to 66.59% for traditional IF neurons, representing an improvement of 2.64 percentage points. These results indicate that our proposed functionalized neuron design (whether $f$-LIF or $f$-IF) can bring significant performance improvements compared to traditional spiking neurons. The functionalized design effectively enhances the learning capability and expressive power of spiking neural networks through more flexible dynamic characteristics.

Table 9: Performance comparison of different neuron types

| Dataset | $f$-IF($f$-SNN) | IF (SJ) | Improvement |
|---|---|---|---|
| DVS128Gesture | 0.9383 | 0.9270 | +1.13% |
| N-Caltech101 | 0.6923 | 0.6659 | +2.64% |

**Memory Parameter Analysis**

To evaluate the effectiveness of the memory parameter in the $f$-SNN framework for accelerating training and saving memory, we conduct experiments with different memory settings. The experiments are performed with batch size 16 and timesteps $T$=16. The results are shown in Table 10. These results demonstrate that the memory parameter in our $f$-SNN framework provides an effective mechanism for balancing memory consumption and training speed. Users can adjust this parameter according to their hardware constraints and training requirements to achieve optimal performance.

### D.2.4 EXTENDED ROBUSTNESS EXPERIMENTS.

To further verify the robustness advantages of $f$-SNN, we design and conduct a series of experiments for investigation.

**Graph Learning Tasks.**

**Robustness to Feature Masking Ratios in Graph Learning.** To evaluate the robustness of the network in graph learning tasks, we conduct feature ablation experiments on two benchmark datasets:

Table 10: Performance comparison of different memory parameters (Batch=16, $T$=16)

| Short Memory Size $M$ | Memory (GB) | test-speed(imgs/s) | train-speed(imgs/s)) | Acc |
|---|---|---|---|---|
| 2 | 14.8 | 190 | 32.0 | 0.9175 |
| 4 | 16.9 | 183 | 29.5 | 0.9164 |
| 6 | 17.1 | 175 | 29.1 | 0.9158 |
| 8 | 17.5 | 163 | 28.3 | 0.9130 |
| 10 | 17.5 | 156 | 27.5 | 0.9158 |
| 12 | 17.8 | 153 | 27.3 | 0.9137 |
| 14 | 18.3 | 151 | 27.0 | 0.9270 |
| 16 | 19.8 | 151 | 26.8 | 0.9362 |

Cora and Citeseer. Specifically, we implement a random feature masking strategy where a proportion of features in the graph feature matrix are zeroed out according to predefined ratios. This tests the irregular sampling scenarios with partially observed multivariate event series. The modified feature matrices are then fed into the models for prediction accuracy evaluation. As shown in Fig. 9, our proposed $f$-SNN method demonstrates significantly improved performance compared to the integrator-methods (SpikingJelly or snnTorch) baseline method across all tested missing rates. This consistent superiority under varying feature dropout scenarios substantiates the enhanced robustness of our approach against input perturbations.

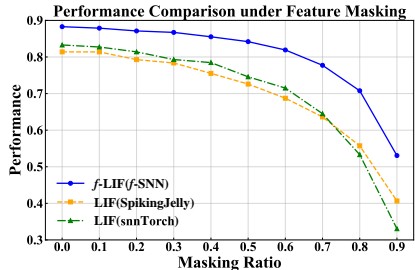
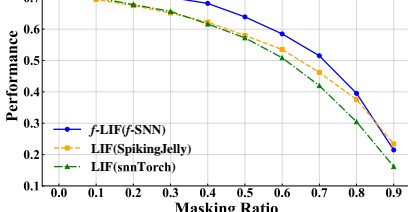

(a) Accuracy vs. Feature Masking on Cora.    (b) Accuracy vs. Feature Masking on Citeseer.

Figure 9: Robustness Comparison: Baseline vs. $f$-SNN in Graph Learning under Feature Dropout Perturbations.

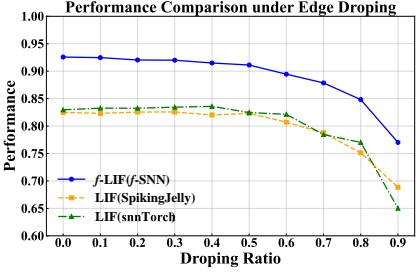
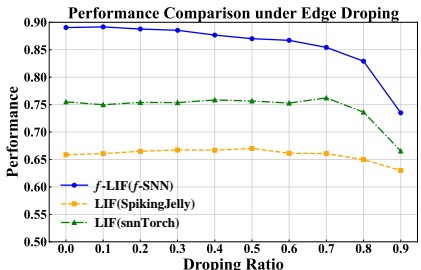

(a) Accuracy vs. Edge Dropping on Photo.    (b) Accuracy vs. Edge Dropping on Computers.

Figure 10: Structural Robustness Evaluation: Baseline vs. $f$-SNN in Graph Learning under Edge Dropping Perturbations.

**Structural Robustness Under Edge Dropping Scenarios.** To comprehensively evaluate the robustness of the network in graph-structured data learning, we conduct edge perturbation experiments on two datasets: Photo and Computers. Specifically, we implement a random edge dropping strategy where a predefined proportion of edges in the graph adjacency matrix are randomly zeroed out according to controlled corruption ratios. The modified adjacency matrices are then utilized for model

evaluation through standard prediction accuracy metrics. As illustrated in Fig. 10, our proposed $f$-SNN framework demonstrates superior performance compared to the SpikingJelly or snnTorch baseline across varying edge dropping rates. This consistent advantage under structural perturbations validates the enhanced robustness of our method, particularly in maintaining predictive stability when encountering incomplete graph information.

**Neuromorphic Data Classification Tasks.** We supplement all robustness test data here and provide a specific list of the tests. As shown in Fig. 11 and Tables 11 to 15. To evaluate the robustness of models under **corrupted frame conditions**, we propose a weighted scoring method. This method is suitable for various frame corruption scenarios, such as frame discarding, noise perturbations, and occlusions. By quantifying model performance under different corruption levels, this method provides a comprehensive assessment of robustness. Specifically, for several corruption levels (e.g., 10%, 20%, 30%, etc.), the model performance is recorded and normalized using its original performance (i.e., performance under no corruption). The normalized performance values are then weighted and summed, and the final weighted score is normalized to a range of $0 - 100$ as the robustness score. The calculation formula is as follows:

$$\text{Final Score}_j = \sum_{i=1}^{N} w_i \cdot \frac{\text{Performance}_{i,j}}{\text{Original Performance}_j} \times 100,$$

where:

- $i$ represents the corruption condition (e.g., frame discard ratio, noise intensity, etc.);
- $j$ represents the model;
- $w_i$ is the weight assigned to the $i$-th corruption condition;
- Performance$_{i,j}$ and Original Performance$_j$ are the model's performance under the $i$-th corruption condition and the original condition, respectively.

In this study, to simplify the experimental design and ensure fair comparisons across models, we assign **equal weights** to all corruption conditions, i.e., $w_i = \frac{1}{N}$, where $N$ is the total number of corruption conditions. This choice avoids introducing any bias and ensures that the robustness score calculation remains objective.

This method provides a unified and intuitive metric to quantify the performance degradation of models across various corruption scenarios, offering a standardized basis for robustness evaluation.

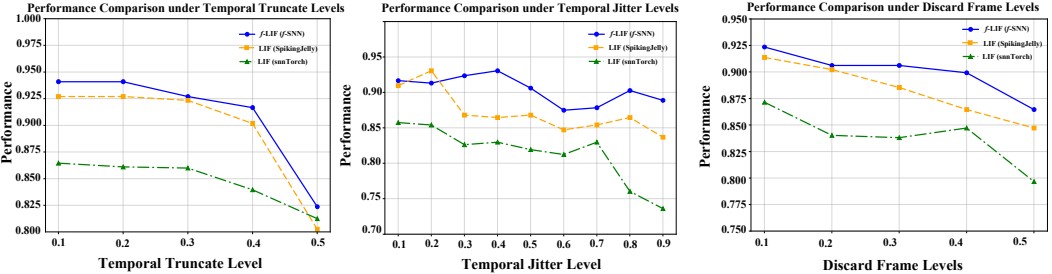

Figure 11: Robustness comparison between traditional SNN and $f$-SNN framework.

| Noise | $f$-SNN | SJ | snnTorch |
|---|---|---|---|
| 0.1 | 0.9479 | 0.9236 | 0.8438 |
| 0.2 | 0.9379 | 0.9201 | 0.8299 |
| 0.3 | 0.9236 | 0.9097 | 0.7674 |
| 0.4 | 0.9132 | 0.7326 | 0.7153 |
| 0.5 | 0.9028 | 0.5139 | 0.6667 |
| Score | 95.96 | 78.81 | 79.12 |
| Original | 0.9480 | 0.9340 | 0.8899 |

Table 11: Performance comparison under noise conditions

| Discard Frame | $f$-SNN | SJ | snnTorch |
|---|---|---|---|
| 0.1 | 0.9236 | 0.9137 | 0.8715 |
| 0.2 | 0.9062 | 0.9023 | 0.8403 |
| 0.3 | 0.9062 | 0.8854 | 0.8381 |
| 0.4 | 0.8993 | 0.8646 | 0.8472 |
| 0.5 | 0.8646 | 0.8472 | 0.7967 |
| Score | 94.93 | 94.50 | 94.25 |
| Original | 0.9480 | 0.9340 | 0.8899 |

Table 12: Performance comparison under frame discard conditions

| Temporal Jitter | $f$-SNN | SJ | snnTorch |
|---|---|---|---|
| 0.1 | 0.9167 | 0.9097 | 0.8576 |
| 0.2 | 0.9132 | 0.9306 | 0.8542 |
| 0.3 | 0.9236 | 0.8681 | 0.8264 |
| 0.4 | 0.9306 | 0.8646 | 0.8299 |
| 0.5 | 0.9062 | 0.8681 | 0.8194 |
| 0.6 | 0.8750 | 0.8472 | 0.8125 |
| 0.7 | 0.8785 | 0.8542 | 0.8299 |
| 0.8 | 0.9028 | 0.8646 | 0.7604 |
| 0.9 | 0.8889 | 0.8368 | 0.7361 |
| Score | 95.35 | 93.31 | 91.47 |
| Original | 0.9480 | 0.9340 | 0.8899 |

Table 13: Performance comparison under temporal jitter conditions

| Temporal Truncate | $f$-SNN | SJ | snnTorch |
|---|---|---|---|
| 0.1 | 0.9410 | 0.9271 | 0.8646 |
| 0.2 | 0.9410 | 0.9271 | 0.8611 |
| 0.3 | 0.9271 | 0.9235 | 0.8600 |
| 0.4 | 0.9167 | 0.9018 | 0.8396 |
| 0.5 | 0.8237 | 0.8026 | 0.8125 |
| Score | 95.98 | 95.97 | 95.24 |
| Original | 0.9480 | 0.9340 | 0.8899 |

Table 14: Performance comparison under temporal truncate conditions

| Occlude Block | $f$-SNN | SJ | snnTorch |
|---|---|---|---|
| 0.1 | 0.9340 | 0.8993 | 0.8681 |
| 0.2 | 0.9340 | 0.8576 | 0.8368 |
| 0.3 | 0.8785 | 0.7917 | 0.8264 |
| 0.4 | 0.8368 | 0.7083 | 0.7396 |
| 0.5 | 0.6840 | 0.5764 | 0.6493 |
| Score | 90.02 | 82.08 | 88.10 |
| Original | 0.9480 | 0.9340 | 0.8899 |

Table 15: Performance comparison under occlude block conditions.

### D.3 ENERGY CONSUMPTION ANALYSIS

Most existing SNN energy analyses primarily account for synaptic operations, while there is no widely adopted methodology for the intrinsic energy of neurons themselves. Therefore, we follow the commonly used practice to estimate the overall SNN energy. In the highlighted energy-analysis part, we use the same methodology as prior work(Yao et al., 2023a; 2024). For neuron-intrinsic costs, we provide the following notation and derivations. The specific algorithm is as follows:

• **Notation**

- $T$: Number of timesteps.
- $\kappa := E_{MAC}/E_{AC}$: Conversion factor from one multiply–accumulate (MAC) operation to equivalent additions ($E_{AC}$).
- $E_{AC}$: Energy of one equivalent addition; $E_{MAC}$: Energy of one multiply–accumulate operation.
- Relation: $E_{MAC} = (\kappa + 1)\,E_{AC}$.
- $N$: Number of SNN convolutional stages.
- $M$: Number of SNN fully connected layers.
- $L$: Number of self-attention (SSA) blocks.
- $\text{FLOPs}(l)$: Floating-point operations of layer $l$ in its dense counterpart.
- $E_{MAC}$: Energy per multiply–accumulate operation.
- $E_{AC}$: Energy per equivalent addition.
- $\rho$: Average firing rate.

Following (Yao et al., 2023a; 2024), we assume that the data for various operations are implemented using 32-bit floating-point arithmetic in 45nm technology, where $E_{MAC} = 4.6pJ$ and $E_{AC} = 0.9pJ$.

**I. LIF (discrete time, hard reset; single neuron, $T$ steps)** Per-term energy (in units of $E_{AC}$):

$$E_{\text{update,in}} = T \cdot \kappa \cdot E_{AC} \quad \text{(update: input multiplication)}, \tag{55}$$

$$E_{\text{leak}} = T \cdot \kappa \cdot E_{AC} \quad \text{(leak: multiplication)}, \tag{56}$$

$$E_{\text{update,sum}} = T \cdot 1 \cdot E_{AC} \quad \text{(update summation: add the two terms)}, \tag{57}$$

$$E_{\text{cmp}} = T \cdot 1 \cdot E_{AC} \quad \text{(threshold comparison)}, \tag{58}$$

$$E_{\text{spike}} = T \cdot \rho \cdot (2\kappa + 1) \cdot E_{AC} \quad \text{(spike: hard reset, arithmetic gating)}. \tag{59}$$

Total energy of a single LIF neuron over $T$ steps:

$$E_{\text{neuron}} = E_{\text{total}}^{\text{LIF,hard}} = T \cdot \left[ (2 + 2\rho)\,\kappa + (2 + \rho) \right] \cdot E_{AC}. \tag{60}$$

**II. $f$-LIF (discrete time, hard reset; single neuron, $T$ steps)** Per-term energy (in units of $E_{AC}$):

$$E_{\text{update+leak}} = T \cdot \log_2 T \cdot (\kappa + 1) \cdot E_{AC} \quad \text{(update + leak: merged)}, \tag{61}$$

$$E_{\text{cmp}} = T \cdot 1 \cdot E_{AC} \quad \text{(threshold comparison)}, \tag{62}$$

$$E_{\text{spike}} = T \cdot \rho \cdot (2\kappa + 1) \cdot E_{AC} \quad \text{(spike: hard reset, arithmetic gating)}. \tag{63}$$

Total energy of a single $f$-LIF neuron over $T$ steps:

$$E_{\text{neuron}} = E_{\text{total}}^{f-\text{LIF,hard}} = T \cdot \left[ (\kappa + 1)\log_2 T + 1 + \rho(2\kappa + 1) \right] \cdot E_{AC}. \tag{64}$$

**Energy accounting for a Spiking-Transformer (SpikFormer).** Based on the updated formula, we further consider a more complex Spiking-Transformer setting and compare energy consumption accordingly. The energy of SpikFormer can be written as

$$E_{\text{SpikFormer}} = E_{MAC} \times \text{FLOPs}_{\text{SNN Conv}}^1 + E_{AC}$$
$$\times \left( \sum_{n=2}^{N} \text{SOP}_{\text{SNN Conv}}^n + \sum_{m=1}^{M} \text{SOP}_{\text{SNN FC}}^m + \sum_{l=1}^{L} \text{SOP}_{\text{SSA}}^l \right), \tag{65}$$

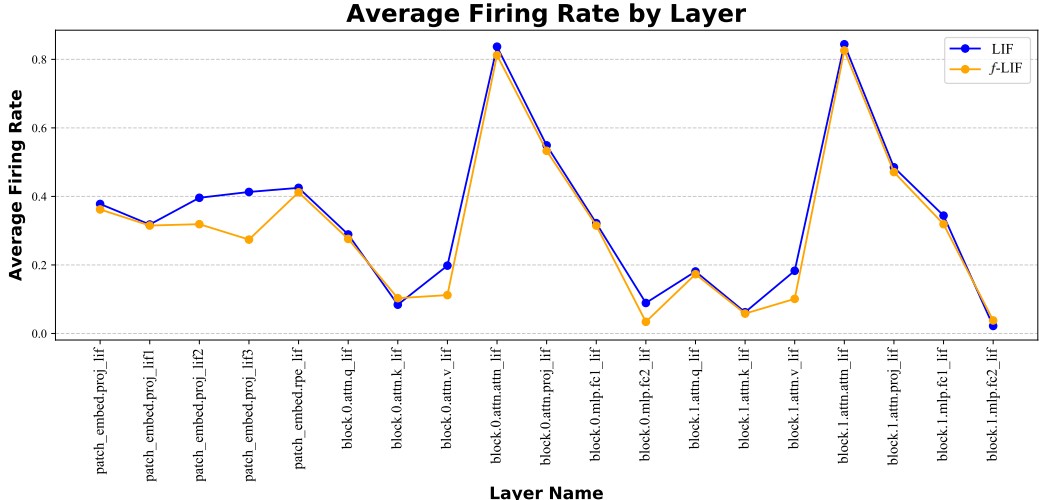

Figure 12: Comparison of average firing rates across layers for LIF and $f$-LIF models. The blue line corresponds to the LIF model, and the orange line corresponds to the f-LIF model.

where SOP denotes the number of spike-based accumulate operations.

For each layer $l$, the spike-based operation count is

$$\text{SOPs}(l) = \rho \times T \times \text{FLOPs}(l). \tag{66}$$

Combining the above equations, the overall energy consumption can be expressed as

$$E_{\text{total}} = E_{\text{SpikFormer}} + E_{\text{neuron}} \times N_{\text{neurons}}. \tag{67}$$

Here, $N_{\text{neurons}}$ denotes the total number of neurons in the network.

As shown in Fig. 12, Table 16, and Table 17, our $f$-LIF neurons enable the network to achieve a lower average firing rate compared to the standard LIF node, although the fractional dynamics introduce additional computational overhead. This results in overall energy usage that remains at a comparable level, balancing the trade-off between energy efficiency and enhanced temporal modeling capabilities.

**Layer-wise energy accounting (SpikFormer).** We report per-layer energy with sparsity-aware synaptic costs and neuron-intrinsic costs. "Energy Type" indicates whether the layer is counted with $E_{MAC}$ or $E_{AC}$; entries of the form "$E_{AC} \times r$" denote $E_{AC}$ cost scaled by the measured average firing rate $r$.

| Layer Name | Type | $T \times$FLOPs (M) | Average Firing Rate | Energy Type | Energy (mJ) |
|---|---|---|---|---|---|
| patch_embed.proj_conv | Conv2d | 150.99 | 1.000 | $E_{MAC}$ | 0.69 |
| patch_embed.proj_lif | LIFNode | 0.00 | 0.378 | Neuron | 0.12 |
| patch_embed.proj_conv1 | Conv2d | 1207.96 | 0.378 | $E_{AC} \times 0.378$ | 0.41 |
| patch_embed.proj_lif1 | LIFNode | 0.00 | 0.318 | Neuron | 0.06 |
| patch_embed.proj_conv2 | Conv2d | 1207.96 | 0.318 | $E_{AC} \times 0.318$ | 0.35 |
| patch_embed.proj_lif2 | LIFNode | 0.00 | 0.396 | Neuron | 0.03 |
| patch_embed.proj_conv3 | Conv2d | 1207.96 | 0.396 | $E_{AC} \times 0.396$ | 0.43 |
| patch_embed.proj_lif3 | LIFNode | 0.00 | 0.413 | Neuron | 0.02 |
| patch_embed.rpe_conv | Conv2d | 603.98 | 0.413 | $E_{AC} \times 0.413$ | 0.22 |
| patch_embed.rpe_lif | LIFNode | 0.00 | 0.425 | Neuron | 0.00 |
| block.0.attn.q_conv | Conv1d | 67.11 | 0.425 | $E_{AC} \times 0.425$ | 0.03 |
| block.0.attn.q_lif | LIFNode | 0.00 | 0.289 | Neuron | 0.00 |
| block.0.attn.k_conv | Conv1d | 67.11 | 0.289 | $E_{AC} \times 0.289$ | 0.02 |
| block.0.attn.k_lif | LIFNode | 0.00 | 0.084 | Neuron | 0.00 |
| block.0.attn.v_conv | Conv1d | 67.11 | 0.084 | $E_{AC} \times 0.084$ | 0.01 |
| block.0.attn.v_lif | LIFNode | 0.00 | 0.198 | Neuron | 0.00 |
| block.0.attn | SSA | 33.55 | 0.198 | $E_{AC} \times 0.198$ | 0.01 |
| block.0.attn.attn_lif | LIFNode | 0.00 | 0.837 | Neuron | 0.00 |
| block.0.attn.proj_conv | Conv1d | 67.11 | 0.837 | $E_{AC} \times 0.837$ | 0.05 |
| block.0.attn.proj_lif | LIFNode | 0.00 | 0.549 | Neuron | 0.00 |
| block.0.mlp.fc1_conv | Conv1d | 268.44 | 0.549 | $E_{AC} \times 0.549$ | 0.13 |
| block.0.mlp.fc1_lif | LIFNode | 0.00 | 0.322 | Neuron | 0.00 |
| block.0.mlp.fc2_conv | Conv1d | 268.44 | 0.322 | $E_{AC} \times 0.322$ | 0.08 |
| block.0.mlp.fc2_lif | LIFNode | 0.00 | 0.089 | Neuron | 0.00 |
| block.1.attn.q_conv | Conv1d | 67.11 | 0.089 | $E_{AC} \times 0.089$ | 0.01 |
| block.1.attn.q_lif | LIFNode | 0.00 | 0.181 | Neuron | 0.00 |
| block.1.attn.k_conv | Conv1d | 67.11 | 0.181 | $E_{AC} \times 0.181$ | 0.01 |
| block.1.attn.k_lif | LIFNode | 0.00 | 0.062 | Neuron | 0.00 |
| block.1.attn.v_conv | Conv1d | 67.11 | 0.062 | $E_{AC} \times 0.062$ | 0.00 |
| block.1.attn.v_lif | LIFNode | 0.00 | 0.183 | Neuron | 0.00 |
| block.1.attn | SSA | 33.55 | 0.183 | $E_{AC} \times 0.183$ | 0.01 |
| block.1.attn.attn_lif | LIFNode | 0.00 | 0.844 | Neuron | 0.00 |
| block.1.attn.proj_conv | Conv1d | 67.11 | 0.844 | $E_{AC} \times 0.844$ | 0.05 |
| block.1.attn.proj_lif | LIFNode | 0.00 | 0.485 | Neuron | 0.00 |
| block.1.mlp.fc1_conv | Conv1d | 268.44 | 0.485 | $E_{AC} \times 0.485$ | 0.12 |
| block.1.mlp.fc1_lif | LIFNode | 0.00 | 0.344 | Neuron | 0.00 |
| block.1.mlp.fc2_conv | Conv1d | 268.44 | 0.344 | $E_{AC} \times 0.344$ | 0.08 |
| block.1.mlp.fc2_lif | LIFNode | 0.00 | 0.022 | Neuron | 0.00 |
| head | Linear | 0.03 | 0.022 | $E_{AC} \times 0.022$ | 0.00 |

Table 16: Layer-wise energy breakdown with sparsity. "Energy Type" uses $E_{MAC}$ for multiply–accumulate energy and $E_{AC}$ scaled by the output average firing rate.

$$E_{\text{synaptic total}} = 2{,}698{,}236.13 \text{ nJ}, \tag{68}$$

$$E_{\text{neuron total}} = 235{,}486.99 \text{ nJ}, \tag{69}$$

$$E_{\text{overall}} = 2{,}933{,}723.12 \text{ nJ} \; = \; 2.933723 \text{ mJ}. \tag{70}$$

Energy composition:

- Synaptic energy share: 92.0%.
- Neuron energy share: 8.0%.

| Layer Name | Type | $T \times$FLOPs (M) | Average Firing Rate | Energy Type | Energy (mJ) |
|---|---|---|---|---|---|
| patch_embed.proj_conv | Conv2d | 150.99 | 1.000 | $E_{MAC}$ | 0.69 |
| patch_embed.proj_lif | $f$-LIFNeuron | 0.00 | 0.362 | Neuron | 0.22 |
| patch_embed.proj_conv1 | Conv2d | 1207.96 | 0.362 | $E_{AC} \times 0.362$ | 0.39 |
| patch_embed.proj_lif1 | $f$-LIFNeuron | 0.00 | 0.315 | Neuron | 0.11 |
| patch_embed.proj_conv2 | Conv2d | 1207.96 | 0.315 | $E_{AC} \times 0.315$ | 0.34 |
| patch_embed.proj_lif2 | $f$-LIFNeuron | 0.00 | 0.319 | Neuron | 0.05 |
| patch_embed.proj_conv3 | Conv2d | 1207.96 | 0.319 | $E_{AC} \times 0.319$ | 0.35 |
| patch_embed.proj_lif3 | $f$-LIFNeuron | 0.00 | 0.274 | Neuron | 0.03 |
| patch_embed.rpe_conv | Conv2d | 603.98 | 0.274 | $E_{AC} \times 0.274$ | 0.15 |
| patch_embed.rpe_lif | $f$-LIFNeuron | 0.00 | 0.412 | Neuron | 0.01 |
| block.0.attn.q_conv | Conv1d | 67.11 | 0.412 | $E_{AC} \times 0.412$ | 0.02 |
| block.0.attn.q_lif | $f$-LIFNeuron | 0.00 | 0.276 | Neuron | 0.00 |
| block.0.attn.k_conv | Conv1d | 67.11 | 0.276 | $E_{AC} \times 0.276$ | 0.02 |
| block.0.attn.k_lif | $f$-LIFNeuron | 0.00 | 0.091 | Neuron | 0.00 |
| block.0.attn.v_conv | Conv1d | 67.11 | 0.091 | $E_{AC} \times 0.091$ | 0.01 |
| block.0.attn.v_lif | $f$-LIFNeuron | 0.00 | 0.112 | Neuron | 0.00 |
| block.0.attn | SSA | 33.55 | 0.112 | $E_{AC} \times 0.112$ | 0.00 |
| block.0.attn.attn_lif | $f$-LIFNeuron | 0.00 | 0.812 | Neuron | 0.00 |
| block.0.attn.proj_conv | Conv1d | 67.11 | 0.812 | $E_{AC} \times 0.812$ | 0.05 |
| block.0.attn.proj_lif | $f$-LIFNeuron | 0.00 | 0.533 | Neuron | 0.00 |
| block.0.mlp.fc1_conv | Conv1d | 268.44 | 0.533 | $E_{AC} \times 0.533$ | 0.13 |
| block.0.mlp.fc1_lif | $f$-LIFNeuron | 0.00 | 0.315 | Neuron | 0.00 |
| block.0.mlp.fc2_conv | Conv1d | 268.44 | 0.315 | $E_{AC} \times 0.315$ | 0.08 |
| block.0.mlp.fc2_lif | $f$-LIFNeuron | 0.00 | 0.034 | Neuron | 0.00 |
| block.1.attn.q_conv | Conv1d | 67.11 | 0.034 | $E_{AC} \times 0.034$ | 0.00 |
| block.1.attn.q_lif | $f$-LIFNeuron | 0.00 | 0.173 | Neuron | 0.00 |
| block.1.attn.k_conv | Conv1d | 67.11 | 0.173 | $E_{AC} \times 0.173$ | 0.01 |
| block.1.attn.k_lif | $f$-LIFNeuron | 0.00 | 0.058 | Neuron | 0.00 |
| block.1.attn.v_conv | Conv1d | 67.11 | 0.058 | $E_{AC} \times 0.058$ | 0.00 |
| block.1.attn.v_lif | $f$-LIFNeuron | 0.00 | 0.101 | Neuron | 0.00 |
| block.1.attn | SSA | 33.55 | 0.101 | $E_{AC} \times 0.101$ | 0.00 |
| block.1.attn.attn_lif | $f$-LIFNeuron | 0.00 | 0.826 | Neuron | 0.00 |
| block.1.attn.proj_conv | Conv1d | 67.11 | 0.826 | $E_{AC} \times 0.826$ | 0.05 |
| block.1.attn.proj_lif | $f$-LIFNeuron | 0.00 | 0.471 | Neuron | 0.00 |
| block.1.mlp.fc1_conv | Conv1d | 268.44 | 0.471 | $E_{AC} \times 0.471$ | 0.11 |
| block.1.mlp.fc1_lif | $f$-LIFNeuron | 0.00 | 0.319 | Neuron | 0.00 |
| block.1.mlp.fc2_conv | Conv1d | 268.44 | 0.319 | $E_{AC} \times 0.319$ | 0.08 |
| block.1.mlp.fc2_lif | $f$-LIFNeuron | 0.00 | 0.038 | Neuron | 0.00 |
| head | Linear | 0.03 | 0.038 | $E_{AC} \times 0.038$ | 0.00 |

Table 17: Layer-wise energy breakdown with sparsity. "Energy Type" uses $E_{MAC}$ for multiply–accumulate energy and $E_{AC}$ scaled by the output firing rate for addition-only equivalents.

$$E_{\text{synaptic total}} = 2{,}490{,}467.47 \text{ nJ}, \tag{71}$$

$$E_{\text{neuron total}} = 421{,}324.24 \text{ nJ}, \tag{72}$$

$$E_{\text{overall}} = 2{,}911{,}791.71 \text{ nJ} = 2.911792 \text{ mJ}. \tag{73}$$

Energy composition:

- Synaptic energy share: 85.5%.
- Neuron energy share: 14.5%.

# E  $f$-SNN TOOLBOX "SPIKEDE" PRESENTATION

We propose an open-source, out-of-the-box toolbox, named **spikeDE**, to support our $f$-SNN framework, built on the `PyTorch` platform. The toolbox enables the construction of SNNs through interfaces closely aligned with `PyTorch`. It supports various neural network architectures like convolutional neural networks (CNN), Transformer, ResNet, and multilayer perceptron (MLP) (Vaswani et al., 2017; LeCun et al., 1989; He et al., 2016; Zhou et al., 2022). We believe it will serve the SNN community well, encouraging the advancement of a broader class of SNNs that capture richer temporal patterns. The directory structure of the toolbox is as follows:

```
├── __init__.py        # Package initialization
├── layer.py           # Base layer definitions
├── neuron.py          # Base neuron definitions
├── odefunc.py         # f-ODE function definitions for SNNs
├── snn.py             # High-level SNN wrapper
├── solver.py          # f-ODE solvers
└── surrogate.py       # Surrogate gradient implementations
```

The `neuron` module provides a variety of classic spiking neurons, including `LIFNeuron` (Leaky Integrate-and-Fire) and `IFNeuron` (Integrate-and-Fire), along with a unified interface for custom neuron dynamics. The `snn` module offers a high-level SNN interface that supports either direct conversion of artificial neural networks (ANNs) into SNNs or wrapping SNNs into trainable network objects.

**SNN Neurons. spikeDE** supports spiking neurons like `LIFNeuron` and `IFNeuron`. Besides, **spikeDE** supports custom spiking neurons. Users can define their own neuron types by inheriting from the provided `BaseNeuron` class:

```
class LIFNeuron(BaseNeuron):
    def forward(self, v_mem, current_input=None):
        if current_input is None:
            return v_mem
        tau = self.get_tau()
        dt = 1.0
        dv_no_reset = (-v_mem + current_input) / tau
        v_post_charge = v_mem + dt * dv_no_reset
        spike = self.surrogate_f(v_post_charge - self.threshold,
                                 self.surrogate_grad_scale)
        dv_dt = dv_no_reset - (spike.detach() * self.threshold) / tau
        return dv_dt, spike

class IFNeuron(BaseNeuron):
    def forward(self, v_mem, current_input=None):
        if current_input is None:
            return v_mem
        tau = self.get_tau()
        v_scaled = v_mem - self.threshold
        spike = self.surrogate_f(v_scaled, self.surrogate_grad_scale)
        dv_dt = (-spike * self.threshold + current_input) / tau
        return dv_dt, spike
```

Our `LIFNeuron` and `IFNeuron` produce two outputs: the first is the derivative of the membrane potential `dv_dt`, and the second is a binary spike signal `spike`.

The `surrogate_opt` parameter specifies the surrogate gradient function used to enable backpropagation through the non-differentiable spiking operation. Multiple options (e.g., `"arctan_surrogate"`) are provided, allowing users to choose based on their needs.

**How to Build and Train a Custom SNN Network.**

**1. Building a CNN-based Network** The **spikeDE** framework is highly flexible and supports various network backbones, including CNNs, ResNets, GNNs, and Transformers. By replacing the activation

functions in traditional artificial neural networks (ANNs) with our spiking neurons, spiking neural dynamics can be seamlessly incorporated into the model. Below is an example using a simple CNN backbone:

```python
import torch.nn as nn

class CustomCNN(nn.Module):

    def __init__(self, args):
        super(CustomCNN, self).__init__()

        # Conv Blocks
        self.conv1 = nn.Conv2d(2, 128, 3, 1, bias=False)
        self.bn1 = nn.BatchNorm2d(128)
        self.lif1 = LIFNeuron(
            args.tau, args.threshold, args.surrogate_grad_scale
        )
        self.pool1 = nn.MaxPool2d(2)

        self.conv2 = nn.Conv2d(128, 128, 3, 1, bias=False)
        self.bn2 = nn.BatchNorm2d(128)
        self.lif2 = LIFNeuron(
            args.tau, args.threshold, args.surrogate_grad_scale
        )
        self.pool2 = nn.MaxPool2d(2)

        self.conv3 = nn.Conv2d(128, 128, 3, 1, bias=False)
        self.bn3 = nn.BatchNorm2d(128)
        self.lif3 = LIFNeuron(
            args.tau, args.threshold, args.surrogate_grad_scale
        )
        self.pool3 = nn.MaxPool2d(2)

        self.conv4 = nn.Conv2d(128, 128, 3, 1, bias=False)
        self.bn4 = nn.BatchNorm2d(128)
        self.lif4 = LIFNeuron(
            args.tau, args.threshold, args.surrogate_grad_scale
        )
        self.pool4 = nn.MaxPool2d(2)

        self.conv5 = nn.Conv2d(128, 128, 3, 1, bias=False)
        self.bn5 = nn.BatchNorm2d(128)
        self.lif5 = LIFNeuron(
            args.tau, args.threshold, args.surrogate_grad_scale
        )
        self.pool5 = nn.MaxPool2d(2)

        # Fully Connected Layers
        self.flatten = nn.Flatten()
        self.dropout1 = nn.Dropout(0.5)
        self.fc1 = nn.Linear(128 * 4 * 4, 512)
        self.lif6 = LIFNeuron(
            args.tau, args.threshold, args.surrogate_grad_scale
        )

        self.dropout2 = nn.Dropout(0.5)
        self.fc2 = nn.Linear(512, 110)
        self.lif7 = LIFNeuron(
            args.tau, args.threshold, args.surrogate_grad_scale
        )

        # Output Layer
        self.output_layer = (
            VotingLayer(10) if args.voting else nn.Linear(110, 11)
```

```
        )

    def forward(self, x):
        # First convolutional block
        x = self.conv1(x)
        x = self.bn1(x)
        x = self.lif1(x)
        x = self.pool1(x)

        # Second convolutional block
        x = self.conv2(x)
        x = self.bn2(x)
        x = self.lif2(x)
        x = self.pool2(x)

        # Third convolutional block
        x = self.conv3(x)
        x = self.bn3(x)
        x = self.lif3(x)
        x = self.pool3(x)

        # Fourth convolutional block
        x = self.conv4(x)
        x = self.bn4(x)
        x = self.lif4(x)
        x = self.pool4(x)

        # Fifth convolutional block
        x = self.conv5(x)
        x = self.bn5(x)
        x = self.lif5(x)
        x = self.pool5(x)

        # Fully connected layers
        x = self.flatten(x)
        x = self.dropout1(x)
        x = self.fc1(x)
        x = self.lif6(x)

        x = self.dropout2(x)
        x = self.fc2(x)
        x = self.lif7(x)

        # Output layer
        x = self.output_layer(x)
        return x
```

Users are free to design other backbone architectures tailored to their specific tasks.

**2. Warpping Your Network.** Once a suitable network backbone is defined, it can be wrapped into a trainable SNN object using our `SNNWrapper`:

```
from spikeDE.snn import SNNWrapper

model = SNNWrapper(CNNBackbone,
    integrator="fdeint",
    interpolation_method="linear"
)

# Set the initial input shape of the network (Needed!!!)
model._set_neuron_shapes(input_shape=(1, 3, h, w))
# h, w denotes the input image's height and width
```

The resulting `model` is a fully trainable SNN that can be trained using standard PyTorch workflows, including automatic differentiation and backpropagation.

The input to this object is a tensor $\boldsymbol{X} \in \mathbb{R}^{T \times N \times *}$, where $T$ denotes the number of time steps and $N$ is the batch size. The output $O$ is a tuple containing the membrane potentials of each layer at every time step, as well as the accumulated spikes from the final layer.

The `integrator` argument supports two modes: `"odeint"` (for integer-order ODE integration) and `"fdeint"` (for fractional-order differential equation integration). Each integrator supports multiple numerical methods, allowing users to balance accuracy and computational efficiency.

**3. Training your Network.** Since our implementation is based on the Neural ODE framework, the training procedure differs slightly from standard PyTorch networks: time integration parameters can be specified or learnable. Below is a basic training loop.

```
def train(args, model, device, train_loader, optimizer, criterion):
    """Train for one epoch."""
    model.train()
    for batch_idx, (data, target) in enumerate(train_loader):
        data, target = data.to(device), target.to(device)
        # data shape: [T, N, *]

        # Define input time points
        data_time = torch.linspace(
            0,
            args.time_interval * (args.time_steps - 1),
            args.time_steps,
            device=device
        ).float()

        method = args.method  # Integration method, e.g., 'euler'
        options = {'step_size': args.step_size}
        optimizer.zero_grad()

        # Define output time points (can be adjusted to save memory,
        #    especially with odeint_adjoint)
        output_time = torch.linspace(
            0,
            args.time_interval * (args.T - 1),
            args.T,
            device=device
        ).float()

        output = model(data, data_time, output_time, method, options)
        # output.mean(0) computes the mean of the output spikes
        # across the time steps in the output layer.
        loss = criterion(output.mean(0), target)
        loss.backward()
        optimizer.step()
        # ... additional logging or evaluation logic
```

`output.mean(0)` corresponds to the network's final prediction across the time steps in the output layer, which can be used flexibly depending on the task (e.g., classification, regression).

## F  LIMITATIONS AND BROADER IMPACTS

### F.1  LIMITATIONS

First, the hyperparameter tuning process for fractional dynamics, such as selecting the fractional order, can be non-trivial and requires domain expertise or extensive experimentation. This may pose challenges for practitioners aiming to deploy the framework in new applications. Second, although our experiments show robustness to noise and strong performance on several datasets, the method has not been extensively compared with a broader set of recent graph learning methods in graph-structured settings. Further study is needed to evaluate its practical constraints and comparative advantages

against newer graph modeling approaches, including recent graph Transformer methods and graph optimization-oriented frameworks (Tao et al., 2025; Zhao et al., 2025). The current $f$-SNN toolbox also lacks mature distributed-training support and remains under active optimization. In addition, modern toolboxes such as SpikingJelly implement CuPy-accelerated computation; integrating similar acceleration is a promising direction for our framework.

## F.2 BROADER IMPACTS

The proposed $f$-SNN framework introduces a biologically inspired approach to spiking neural networks, with the potential for significant positive impact on both research and application domains. By incorporating fractional-order dynamics, $f$-SNN advances the modeling of complex temporal dependencies, offering new insights into brain-like computation and contributing to the understanding of non-Markovian behavior in biological neurons. This could inspire further interdisciplinary research in neuroscience and machine learning.

From an application perspective, the ability of $f$-SNN to process temporal information with enhanced accuracy and energy efficiency makes it well-suited for tasks such as edge computing, Internet of Things (IoT) devices, and neuromorphic hardware. The open-sourced toolbox provides a practical resource for researchers and practitioners, potentially accelerating innovation in fields like computational neuroscience, robotics, and bio-inspired artificial intelligence.

However, as with any machine learning framework, ethical considerations must be addressed. The deployment of $f$-SNN in sensitive applications, such as autonomous systems or decision-making tasks, should be carefully evaluated to avoid unintended consequences. Additionally, the increased computational requirements of fractional dynamics raise concerns about energy consumption during training, which may counteract the energy efficiency benefits of SNNs in some scenarios. Responsible use, combined with efforts to improve computational efficiency, will be essential to maximize the positive societal impact of this technology.

