# OpenReview forum: "Fractional-Order Spiking Neural Network"
_ICLR.cc/2026/Conference — ICLR 2026 Poster_

### Official Review · Reviewer_XVNF · 2025-10-31

**Soundness:** 3
**Presentation:** 3
**Contribution:** 2
**Rating:** 4
**Confidence:** 3

**Summary:**

This paper proposes a fractional-order spiking neural network (f-SNN) framework, which replaces traditional first-order membrane potential dynamics in SNNs with fractional-order ordinary differential equations (f-ODEs). The key idea is that fractional derivatives introduce non-Markovian memory effects, allowing neurons to integrate long-term dependencies following a power-law decay rather than exponential decay. The paper provides (1) theoretical analysis of fractional memory; (2) efficient discretization; and (3) extensive experiments on neuromorphic vision and graph datasets, showing consistent improvements in accuracy, energy efficiency, and robustness.

**Strengths:**

1. The introduction of fractional dynamics into SNNs is novel and biologically inspired. It directly addresses the Markovian limitation of conventional LIF/IF neurons, linking to real neuronal behaviors and fractal dendritic structures.

2. The derivation from Caputo derivatives to the discrete f-LIF formulation is rigorous. Theorem 1 and the comparison of exponential vs. power-law decay are clearly presented and intuitive.

3. The f-SNN framework is a strict superset of conventional SNNs (α = 1), making it easily integrable into existing architectures and toolkits.

4. Results across multiple domains (neuromorphic datasets and graph benchmarks) show consistent gains. The robustness tests (noise, occlusion, temporal jitter, etc.) are comprehensive and convincing.

**Weaknesses:**

1. The theory mainly discusses long-term dependence qualitatively via the Mittag–Leffler function; deeper analyses such as stability, gradient dynamics, or expressive power would strengthen the theoretical contribution.

2. Although the paper mentions O(NlogN) and truncated-memory approximations, there is no clear empirical benchmark on training/inference speed or memory consumption compared to standard SNNs.

3. Lack of ablation on fractional order $\alpha$. While $\alpha$ is a key hyperparameter, the paper does not analyze its sensitivity or interpretability across tasks.

4. Some sections (e.g., Sec. 3.1 discretization) could benefit from clearer notation and step-by-step explanation. A schematic showing the memory kernel evolution over time would help readers intuitively grasp the effect of $\alpha$.

**Questions:**

1. How scalable is f-SNN to large datasets (e.g., DVS-Gesture full size) in wall-clock time and memory usage?

2. Have the authors tried integrating f-LIF with surrogate gradient training, jointly optimizing α and synaptic weights?

3. Can the same principle apply to recurrent  SNN architectures?

4. Why do SpikingJelly and snnTorch produce such different results under the same dataset and network architecture?

---

> ### Author Response · Authors · 2025-11-21
> **Weakness 1: More theoretic analysis**
>
> > The theory mainly discusses long-term dependence qualitatively via the Mittag–Leffler function; deeper analyses such as stability, gradient dynamics, or expressive power would strengthen the theoretical contribution.
>
> **Response:**
>
> We thank the reviewer for this insightful suggestion. We agree that establishing rigorous guarantees regarding stability and expressivity significantly strengthens the contribution. In response, we have derived two new theorems in Section 3.2 (with full proofs in Appendix C.2 and C.3) regarding the robustness and theoretical expressivity of *f*-SNNs compared to integer-order counterparts.
>
> The following have been added to Section 3.2 of the paper. The full proofs are provided in Sections C.2 and C.3.
>
> The *f*-SNN framework demonstrates superior robustness compared to traditional SNNs. Empirical studies show that *f*-LIF neurons maintain reliable spike patterns under noisy inputs [C1]. Here, we provide theoretical robustness guarantees.
>
> **Theorem 1 (Robustness of *f*-SNN).**
> Consider a fractional *f*-IF neuron governed by the dynamics $\tau D^\alpha U(t)=R I_{\mathrm{in}}(t)$ with fractional order $0<\alpha<1$ and initial condition $U_0=0$. Under a constant input current $I_{\mathrm{c}}$ subject to an additive perturbation $\epsilon$ (where $|\epsilon| \ll I_{\mathrm{c}}$), the system exhibits the following robustness properties relative to the classical integer-order model ($\alpha=1$):
>
> - **Membrane Potential Robustness:** The membrane potential deviation due to perturbation evolves as:
> $$\begin{aligned}
> \Delta U^{f-\mathrm{IF}}(t) & =\frac{R \epsilon}{\tau \Gamma(\alpha+1)} t^\alpha \quad(\text{sub-linear growth}), \\\\
> \Delta U^{\mathrm{IF}}(t) & =\frac{R \epsilon}{\tau} t \quad(\text{linear growth})
> \end{aligned}$$
> For $0<\alpha<1$, the fractional-order dynamics suppress long-term perturbation accumulation through sub-linear temporal scaling.
>
> - **Spike Timing Sensitivity:** For small perturbations $\epsilon \ll I_{\mathrm{c}}$, the spike time shift magnitude scales as:
> $$\begin{aligned}
> \left|\Delta t_s^{f-\mathrm{IF}}\right| & \propto \epsilon \cdot I_{\mathrm{c}}^{-(1+1/\alpha)}, \\\\
> \left|\Delta t_s^{\mathrm{IF}}\right| & \propto \epsilon \cdot I_{\mathrm{c}}^{-2}
> \end{aligned}$$
> Since $(1+1/\alpha)>2$ for $0<\alpha<1$, the fractional-order model exhibits enhanced spike timing robustness for high input currents.
>
> **Remark 4.** The fractional-order dynamics yield distinct robustness advantages. The sub-linear perturbation growth $t^\alpha(\alpha<1)$ significantly suppresses long-term accumulation compared to linear growth in classical models. Additionally, the enhanced spike timing stability becomes crucial for precise temporal coding applications [C2,C3]. These properties make *f*-SNNs particularly suited for tasks requiring sustained accuracy and temporal precision under varying input conditions, as confirmed by our experiments in Section 4.
>
> We now establish that *f*-SNNs possess computational capabilities that fundamentally exceed those of finite integer-order systems:
>
> **Theorem 2 (Irreducibility of *f*-IF Dynamics to Finite Classic LIF Ensembles).** Let $U^{f\text{-IF}}$ denote the trajectory of a *f*-IF neuron with order $\alpha \in(0,1)$. There exist no finite integer $W$, weights $\\{\phi_i\\}\_{i=1}^W$, and leak factors $\\{\beta_i\\}\_{i=1}^W$ such that the following holds:
>
> $$\hat{U}\_k=\sum_{i=1}^W \phi_i U_k^{\mathrm{LIF}(\beta_i)} \equiv U_k^{f-\mathrm{IF}} \quad \forall k$$
>
> for general input $X_k$. The impulse response error of the approximation is $O(k^{\alpha-1})$, decaying algebraically slowly. The f-IF neuron is mathematically equivalent to an aggregate of integer-order LIF neurons if and only if $W \rightarrow \infty$, specifically as an integral over a continuum of leak factors.
>
> **Remark 5 (Implications for Neural Computation and Expressive Power).** Theorem 2 demonstrates that *f*-SNNs possess a temporal expressivity that surpasses that of finite integer-order SNN ensembles. A single *f*-IF neuron can represent a continuum of timescales, whereas an infinite bank of integer-order units would be needed for exact representation. The slow $O(k^{\alpha-1})$ decay of the impulse response approximation error highlights that such long-range dependencies are inaccessible to finite-order models without considerable architectural complexity.
>
>
> [C1] Teka, Wondimu, et al. "Neuronal spike timing adaptation described with a fractional leaky integrate-and-fire model." *PLoS Computational Biology*, 10(3):e1003526, 2014.
>
> [C2] Sander M. Bohte, et al. "Error-backpropagation in temporally encoded networks of spiking neurons." *Neurocomputing*, 48(1-4):17–37, 2002.
>
> [C3] Nitin Rathi, et al. "Enabling deep spiking neural networks with hybrid conversion and spike timing dependent backpropagation." In *International Conference on Learning Representations*, pp. 1–14, 2019.

---

> ### Author Response · Authors · 2025-11-21
> **Weakness 2 & Question 1: More empirical complexity analysis and more datasets**
>
> >Although the paper mentions O(NlogN) and truncated-memory approximations, there is no clear empirical benchmark on training/inference speed or memory consumption compared to standard SNNs.
>
> >How scalable is f-SNN to large datasets (e.g., DVS-Gesture full size) in wall-clock time and memory usage?
>
> **Response:**
>
> We are not entirely certain which dataset the reviewer refers to by the "DVS-Gesture full-size" dataset. If the reviewer is referring to the uncropped version with the original event resolution of 240×180, the number of samples remains the same as in DVS-Gesture128, with the primary difference being the spatial resolution/cropping. To comprehensively address scalability concerns, we evaluated our method on the `larger HAR-DVS dataset`, with results reported in **Table A3**.
>
> Furthermore, in the rebuttal, we have added results on more datasets, including CIFAR-10, CIFAR-100, and `ImageNet` as shown in **Table B1**.
> For the relatively smaller CIFAR-10 and CIFAR-100 datasets, we adopt Spiking-ResNet-18 as the baseline.
> For ImageNet, we follow the SpikFormer [D1] configuration with 29.7M parameters, and set both the training and validation image sizes to 160 × 160 for a fair comparison.
> Our *f*-LIF achieves clear gains over both LIF (SpikingJelly) and LIF (snnTorch) on all datasets.
>
> ---
>
> | Model | Train speed (FPS) | Test speed (FPS) | Memory (GB) | Acc1 |
> |-------|-------------------|------------------|-------------|------|
> | LIF (snnTorch) | 125.85 | 258.26 | 12.89 | 0.4610 |
> | LIF (SpikingJelly) | 143.58 | 276.20 | 12.36 | 0.4626 |
> | *f*-LIF (*f*-SNN) | 126.77 | 260.24 | 13.84 | __0.4766__ |
> | *f*-LIF ($M=1$) | 167.83 | 344.79 | 13.84 | 0.4518 |
> | *f*-LIF ($M=2$) | 147.67 | 308.72 | 13.84 | 0.4656 |
> | *f*-LIF ($M=4$) | 138.70 | 268.23 | 13.84 | 0.4712 |
> | *f*-LIF (*f*-SNN, adjoint) | 98.74 | 238.23 | 9.35 | 0.4752 |
>
> **Table A3:** Training and inference speed (FPS), peak memory (GB), and top-1 accuracy (Acc1) across implementations. Here, $M$ denotes the truncation window size for history terms; "adjoint" indicates the training with adjoint method. Experiments are conducted on a CNN-based SNN architecture with the large-scale dataset HAR-DVS with a timestamp of 8.
>
>
>
> ---
>
>
>  | Datasets | Architecture | Timesteps | LIF (SJ) | LIF (snnTorch) | *f*-LIF (*f*-SNN) |
> |----------|--------------|-----------|----------|----------------|-------------------|
> | CIFAR-10 | Spiking-ResNet-18 | 4 | 0.9134 | 0.9026 | __0.9215__ |
> | CIFAR-100 | Spiking-ResNet-18 | 4 | 0.6813 | 0.6445 | __0.6874__ |
> | ImageNet | SpikFormer | 4 | 0.6637 | 0.6584 | __0.6791__ |
> | ImageNet (spike encoder) | SpikFormer | 4 | 0.5549 | 0.5432 | __0.5738__ |
>
> **Table B1:** Top-1 accuracy across datasets, architectures, and timesteps for LIF baselines (SpikingJelly and snnTorch) and our fractional LIF (*f*-SNN).
>
> ---
>
> **Empirical Computation Efficiency:** Conventional SNNs predominantly rely on Backpropagation Through Time (BPTT) with surrogate gradients; we adhere to this standard for training our *f*-SNN. As detailed in Table R3, our model achieves training speeds comparable to the LIF implementation in snnTorch and is only marginally slower than the highly optimized LIF in SpikingJelly. Notably, utilizing a truncated history window allows *f*-SNN to surpass baseline speeds while preserving (or even improving) accuracy. Furthermore, during the rebuttal, we implemented the fractional adjoint sensitivity method [D2]. As shown in Table R3, this approach substantially reduces peak memory usage while maintaining competitive training speeds. Overall, the introduction of fractional dynamics does not impose a significant computational burden compared to standard SNNs. While alternative learning rules (e.g., STDP) are promising, they remain orthogonal to the contributions of this work.
>
> [D1] Zhou, Zhaokun, et al. "Spikformer: When spiking neural network meets Transformer", ICLR 2023.
>
> [D2] SM, Sivalingam. "Neural Fractional Order Differential Equations with Adjoint Based Training."

---

> ### Author Response · Authors · 2025-11-21
> **Weakness 3 & Question 2: Discussion of $\alpha$  and Clarification of f-SNN training**
>
> >Lack of ablation on fractional order $\alpha$. While $\alpha$ is a key hyperparameter, the paper does not analyze its sensitivity or interpretability across tasks.
>
> >Have the authors tried integrating f-LIF with surrogate gradient training, jointly optimizing  $\alpha$ and synaptic weights?
>
> We conducted dedicated experiments on $\alpha$ in Sec. D.2.2 and further added a new ablation study on Transformer-based SNNs for the N-Caltech101 and DVS-Lips datasets, as shown in **Table C1** and **Table A1**. The optimal $\alpha$ exhibits notable dataset dependence, as summarized in **Table A2**.
>
> __Explanation:__ Intuitively, a smaller $\alpha$ indicates that the *f*-SNN relies more heavily on historical information. On sparser neuromorphic datasets, the learned $\alpha$ tends to be smaller, reflecting a greater dependence on temporal history that aligns with the data's intrinsic characteristics. For instance, after denoising, DVS-Lips and HAR-DVS exhibit much lower event densities than earlier datasets such as DVS-Gesture128 and N-Caltech101; correspondingly, their optimal $\alpha$ values are generally smaller.
>
> ---
>
> | $\alpha$ | 0.1    | 0.2    | 0.4    | 0.6    | 0.8    | 1.0    | LIF (SJ) |
> |----------|--------|--------|--------|--------|--------|--------|----------|
> | Acc1     | 0.4723 | 0.4695 | 0.4677 | 0.4683 | 0.4626 | 0.4621 | 0.4614   |
>
> **Table C1:** Accuracy (Acc1) across different fractional orders $\alpha$ and the LIF (SJ) baseline on the Har-DVS dataset using a Transformer-based SNN.
>
> ---
>
> | $\alpha$ | 0.2    | 0.4    | 0.6    | 0.8    | 0.9    | 1.0    | Learnable                        | LIF (SJ) |
> |----------|--------|--------|--------|--------|--------|--------|----------------------------------|----------|
> | Acc1     | 0.7426 | 0.7325 | 0.7548 | 0.7595 | 0.7626 | 0.7421 | 0.7543 (Final $\alpha$: 0.9106) | 0.7263   |
>
> **Table A1:** Accuracy (Acc1) across different fractional orders $\alpha$, a learnable-$\alpha$ setting (final learned $\alpha = 0.9106$), and the LIF (SJ) baseline on the N-Caltech101 dataset using a Transformer-based SNN. The learnable-$\alpha$ setting achieved results comparable to manually tuned $\alpha$.
>
> ---
>
>
> | Dataset       | DVS-Gesture128 | N-Caltech101 | HAR-DVS | DVS-Lips |
> |---------------|----------------|--------------|---------|----------|
> | Best $\alpha$ | 0.5            | 0.9          | 0.1     | 0.1      |
>
> **Table A2:** Best-performing $\alpha$ values for the datasets used in our experiments.
>
> ---
>
>
> **Jointly Learnable vs. Fixed $\alpha$.** **Our framework supports treating $\alpha$ as a learnable parameter.**
> As demonstrated in Table 6 (Sec. D.2.2), a learnable $\alpha$ achieves performance on par with manual tuning and consistently outperforms the integer-order LIF baseline.
> We report manually tuned results primarily to **ensure numerical stability in extreme regimes**.
>
> The fractional derivative definition involves the Gamma function $\Gamma(\alpha)$, which contains poles (singularities) at $\alpha=0$.
>
> - **Regime $\alpha \gg 0$:** For datasets like N-Caltech101, where optimal dynamics require larger $\alpha$, joint optimization is stable and effective. (The best-performing $\alpha$ values are given in **Table A2**)
> - **Regime $\alpha \rightarrow 0$:** For datasets requiring long-term memory (e.g., HARDVS), the optimal $\alpha$ is small. Learning $\alpha$ in this region risks gradient explosion due to the digamma function's behavior near the pole.
>
> Therefore, *manual tuning was chosen to enforce a rigorous, consistent evaluation standard across all datasets*, regardless of their temporal characteristics.
>
> **Marginal Cost of Tuning.** It is important to note that $\alpha$ is a single scalar hyperparameter. Unlike architecture search, tuning $\alpha$ requires only a coarse grid search (e.g., interval of 0.1 or 0.2). The empirical improvements (accuracy boost and robustness improvement) outweigh the overhead required to find the optima $\alpha$.
>
> **Path to More Efficiency: Distributed Fractional Operators.** To solve the stability issue at $\alpha \rightarrow 0$ and fully automate training in future work, we propose adopting *Distributed Fractional-Order Differential Equations* [E1, E2]. Rather than optimizing a single parameter $\alpha$, we can model the system as a weighted superposition of derivatives:
> $$\mathcal{D}[U(t)] \approx \sum_{j=0}^n w_j D^{\alpha_j} U(t)$$
> Here, $\alpha_j$ are fixed discretization points (avoiding singularities) and $w_j$ are learnable weights. This formulation converts finding $\alpha$ into a convex optimization of weights $w_j$, ensuring both training stability and efficiency.
>
> [E1] Ding, Wei, et al. "Applications of distributed-order fractional operators: A review." Entropy 23.1 (2021): 110.
>
> [E2] Diethelm, Kai, and Neville J. Ford. "Numerical analysis for distributed-order differential equations." J. Computational Applied Math. (2009): 96-104.

---

> ### Author Response · Authors · 2025-11-21
> **Weakness 4: More explanation above discretization**
>
> > Some sections (e.g., Sec. 3.1 discretization) could benefit from clearer notation and step-by-step explanation. A schematic showing the memory kernel evolution over time would help readers intuitively grasp the effect of $\alpha$.
>
> Thank you for your suggestion. We have added further clarification in the revised manuscript. In Section B.3, we now explain the following:
>
> The coefficients $c_m^{(\alpha)} = \frac{1}{\tau^\alpha \alpha \Gamma(\alpha)} \left[ (m+1)^\alpha - m^\alpha \right]$ define a causal memory kernel that decays according to a power-law, capturing the historical influence of the system. As shown in Fig.8, this decay is "heavy-tailed," meaning that while recent states dominate the influence, the contributions from past events remain significant over time. This behavior contrasts with the rapid decay observed in integer-order systems.
>
> For large $m$, the decay of input influence follows the relationship:
> $$c_m^{(\alpha)} \propto (m+1)^\alpha - m^\alpha$$
> Expanding this expression, for large $m$, we get:
> $$c_m^{(\alpha)} \approx m^\alpha \left( \left( 1 + \frac{1}{m} \right)^\alpha - 1 \right) \propto m^{\alpha-1}.$$
> This confirms that the kernel exhibits algebraic decay.

---

> ### Author Response · Authors · 2025-11-21
> **Question 2: Recurrent SNN architectures**
>
> > Can the same principle apply to recurrent SNN architectures?
>
> **Response:** Yes
>
> **Recurrent SNN experiments.**
> Although an SNN can be viewed as a recurrent computation over time, by "recurrent SNN" here we refer to architectures such as spiking GRU or spiking LSTM. In these designs, the spiking component is often used merely as a gating (activation) mechanism without explicit neural dynamical states; thus, they fall outside the primary scope of our paper.
> Nevertheless, to broaden our evaluation of RNN-style models, we replaced the gating units in spiking LSTM/GRU with integrate-and-fire (IF) neurons that possess neural dynamics and conducted experiments using an open-source project. Specifically, we performed 18-language classification with spiking GRU and spiking LSTM. The results in **Table C2** below validate the effectiveness of our *f*-SNN in recurrent settings:
>
>
> ---
>
> | Split ratio (8:2) | spikingGRU (IF) | spikingLSTM (IF) | spikingGRU (*f*-IF) | spikingLSTM (*f*-IF) |
> |-------------------|-----------------|------------------|---------------------|----------------------|
> | Accuracy          | 0.827           | 0.808            | __0.829__               | 0.813                |
>
> **Table C2:** Performance comparison of spiking recurrent models on 18-language classification task. *f*-IF denotes fractional integrate-and-fire neurons.

---

> ### Author Response · Authors · 2025-11-21
> **Question 3: SpikingJelly vs. snnTorch**
>
> The discrepancies between SpikingJelly and snnTorch may stem from the implementation differences. For example, they have different discretization/iteration settings:
>
> **SpikingJelly membrane dynamics:** It uses proper Euler discretization with temporal scaling. The input current $I[t]$ is scaled by $(\Delta t/\tau)$ before accumulation (with $V_{\mathrm{reset}}=0$, a simplified form is $V[t+1]=(1-\Delta t/\tau)V[t]+(\Delta t/\tau) I[t]$).
>
> **snnTorch membrane dynamics:** It use simplified recursion without scaling. The input $I[t+1]$ is added directly to the decayed potential, $V[t+1]=\beta V[t]+I[t+1]$, without additional scaling by $(1-\beta)$.
>
> This leads to different effective time constants even with matched parameters.

---

> ### Author Response · Authors · 2025-11-26
>
> Dear Reviewer XVNF,
>
> We sincerely appreciate the time and effort you dedicated to reviewing our paper. In our rebuttal and revised draft, we have conducted additional experiments and expanded the theoretical analysis to address your comments as thoroughly as possible.
>
> Could you kindly let us know whether these revisions sufficiently address your main concerns?
>
> Thank you again for your valuable time and feedback.
>
> Best regards,
>
> The Authors

---

> > ### Comment · Reviewer_XVNF · 2025-11-28
> >
> > Thank you for your detailed response. Most of my concerns have been addressed, so I have decided to raise the score to 6. I still have a few minor suggestions for further improvement.
> >
> > (1) The way SpikingJelly and snnTorch are described in the manuscript may cause some confusion. You may consider distinguishing them using “with input decay” and “without input decay,” since SpikingJelly also provides a "input decay=False" API that computes in the same form as the version you labeled as snnTorch.
> >
> > (2) You may also consider discussing additional spiking neuron models in the related work, such as CLIF, ALIF, GLIF, PSN and others.
> >
> > (3) The firing rates in Tables 16 and 17 can be plotted as line charts to allow for clearer comparison.

---

> > > ### Author Response · Authors · 2025-12-03
> > > **Thanks for the Revised Score and New Suggestions**
> > >
> > > We thank Reviewer XVNF for the positive reassessment and the helpful suggestions!
> > >
> > >
> > > ### **Suggestion 1: More settings for SpikingJelly and snnTorch**
> > >
> > > > (1) The way SpikingJelly and snnTorch are described in the manuscript may cause some confusion. You may consider distinguishing them using “with input decay” and “without input decay,” since SpikingJelly also provides a "input decay=False" API that computes in the same form as the version you labeled as snnTorch.
> > >
> > >
> > > **Response:**
> > >
> > >
> > > We thank the reviewers for recommending a deeper alignment between SpikingJelly and snnTorch. To address this, we aligned these factors and conducted an experiment on the most divergent dataset (DVS-Lips). The results are presented in **Table A1**.
> > >
> > > | Dataset   | SJ decay (True) | SJ decay (False) | snnTorch | *f*-LIF (Ours) |
> > > |-----------|-----------------|------------------|----------|-------------|
> > > | DVS-Lips  | 0.4241          | 0.3645           | 0.3271   | 0.4342      |
> > >
> > > **Table A1:** Comparison of SpikingJelly with `input_decay=False` (SJ decay: False), snnTorch LIF, and our proposed *f*-IF model.
> > >
> > > With `input_decay=False`, SpikingJelly and snnTorch exhibit closer behavior but still differ. Beyond `input_decay`, we identified a key difference in reset semantics: snnTorch’s reset detaches hidden states (breaking prior computation graphs), whereas SpikingJelly’s reset only zeroes membrane potentials without detaching by default. This alters gradient boundaries in segmented BPTT/long-sequence training and can lead to performance differences. Notably, achieving the same detach behavior in SpikingJelly is not feasible without modifying its source code. We do not claim that one reset mechanism is universally superior; performance depends on the dataset and task (e.g., snnTorch outperforms SpikingJelly on HarDVS).
> > >
> > > In the revision, we will label them in our comparison table as `input_decay=True` and `input_decay=False`.

---

> ### Author Response · Authors · 2025-12-03
> **Suggestion 2: Related works including CLIF, ALIF, GLIF, PSN and others**
>
> > (2) You may also consider discussing additional spiking neuron models in the related work, such as CLIF, ALIF, GLIF, PSN and others.
>
>
>
> **Response:**
> Thank you for your suggestion. We have added additional related works in Section A.1. in terms of additional SNN models:
>
> _Updated parts of Section A.1:_ （marked in blue in the paper)
>
> ......Beyond the basic IF/LIF models, a variety of extensions have been proposed to address different modeling challenges. Adaptive Leaky Integrate-and-Fire (ALIF) SNN neuron incorporates neural adaptation mechanisms, such as adaptive thresholds, enhancing the temporal dependency modeling and working memory capacity of SNNs [A1,A2]. The Generalized LIF (GLIF) introduces a more physiologically motivated framework, enabling accurate spike detection and unsupervised differentiation of cortical cell types [A3]. The Complementary LIF (CLIF) model further enhances temporal gradient propagation and long-term dependency learning by incorporating a complementary membrane potential state [A4]. The Parallel Spiking Neuron (PSN) model eliminates the need for reset mechanisms, facilitating fully connected temporal modeling that allows for time-step parallelism, which accelerates both training and inference [A5]. Additional neuron models, such as ternary spikes [A6] and adaptive membrane time constants [A7,A8], further extend the capabilities of SNNs.......
>
>
> **Distinction from Prior SNN Families.**
> The models discussed above all belong to the *integer-order* family, where the subthreshold dynamics can be expressed as a first-order ordinary differential equation (ODE) of the form
>
> >First-order SNN neuron dynamics: $\frac{\mathrm{d} U(t)}{\mathrm{d} t} = \mathrm{Dynamic}\bigl(U(t), I_{\mathrm{in}}(t)\bigr)$
>
> where $\mathrm{Dynamic}(\cdot)$ denotes the specific membrane-potential update rule used by models such as IF, LIF (cf. (10) and (11)), ALIF, GLIF, CLIF, and other related variants. These approaches mainly explore different choices for the function $\mathrm{Dynamic}(\cdot)$, all within the same first-order, Markovian framework.
>
> In contrast, our work introduces a *fractional-order* SNN framework, which is based on
>
> >Fractional-order SNN neuron dynamics: $D_t^{\alpha} U(t) = \mathrm{Dynamic}\bigl(U(t), I_{\mathrm{in}}(t)\bigr), \qquad 0 < \alpha \le 1$
>
> where $D_t^{\alpha}$ represents a fractional (nonlocal-in-time) derivative. This formulation generalizes the integer-order case, which is recovered when $\alpha = 1$, and incorporates long-term memory in the membrane potential and spike trains through fractional dynamics. In the main text, we instantiate this framework with fractional IF and LIF neurons, but the same approach can naturally extend to more complex neuron models, such as ALIF, GLIF, and CLIF. Our `spikeDE` toolbox offers modular implementations that facilitate the realization of these fractional variants. We believe this contribution significantly advances the field by introducing nonlocal-in-time discrete dynamics to SNN modeling.
>
> [A1] Bellec, Guillaume, et al. "Long short-term memory and learning-to-learn in networks of spiking neurons." NeurIPS 2018.
>
> [A2] Benda, Jan. "Neural adaptation." Current Biology 31 (2021): R110-R116.
>
> [A3] Teeter, Corinne, et al. "Generalized leaky integrate-and-fire models classify multiple neuron types." Nature Communications 9 (2018): 709.
>
> [A4] Huang, Yulong, et al. "Clif: Complementary leaky integrate-and-fire neuron for spiking neural networks." ICML 2025.
>
> [A5] Fang, Wei, et al. "Parallel spiking neurons with high efficiency and ability to learn long-term dependencies." NeurIPS 2023.
>
> [A6] Guo, Yufei, et al. "Ternary spike: Learning ternary spikes for spiking neural networks." AAAI 2024.
>
> [A7] Koch, Christof, et al. "A brief history of time (constants)." Cerebral Cortex 6 (1996): 93–101.
>
> [A8] Zhang, Jiqing, et al. "Spiking neural networks with adaptive membrane time constant for event-based tracking." IEEE Transactions on Image Processing 2025.
>
>
> ### **Suggestion 3: Visualization of firing rates**
>
> >(3) The firing rates in Tables 16 and 17 can be plotted as line charts to allow for clearer comparison.
>
>
>
> **Response:**
>
> We thank the reviewers for the suggestions to improve readability. We have redrawn the figure comparing firing rates between Spikformer using LIF and *f*-LIF (see Figure 12).

---

### Official Review · Reviewer_xtnY · 2025-11-03

**Soundness:** 3
**Presentation:** 3
**Contribution:** 3
**Rating:** 6
**Confidence:** 4

**Summary:**

This paper proposes a fractional-order spiking neural network (f-SNN) framework that replaces traditional integer-order ODEs with fractional-order ODEs to capture long-term dependencies in biological neurons, outperforming conventional SNNs in accuracy, energy efficiency, and noise robustness across neuromorphic vision and graph tasks while providing an open-source toolbox spikeDE for easy adoption.

**Strengths:**

1. The proposed fractional-order spiking neural network (f-SNN) framework effectively captures long-term dependencies in membrane potential and spike trains via fractional dynamics, addressing the limitation of traditional integer-order SNNs (based on IF/LIF models) that only rely on immediate past states.

2. The f-SNN demonstrates superior performance in multiple aspects: it achieves higher accuracy than conventional SNNs on neuromorphic vision

**Weaknesses:**

1. The concept of fractional-order has already been proposed by others; the authors seem to have only put forward a framework and tested it on different datasets, which is more like an engineering problem.

2. The datasets used are limited, and results on widely tested datasets such as CIFAR-10, CIFAR-100, and ImageNet are missing.

**Questions:**

The energy consumption analysis of SNNs only includes synaptic analysis, while the energy consumption of the neurons themselves should also be compared.

---

> ### Author Response · Authors · 2025-11-21
> **Weakness 1: Novelty Beyond Prior Fractional-Order Work**
>
> > The concept of fractional-order has already been proposed by others; the authors seem to have only put forward a framework and tested it on different datasets, which is more like an engineering problem.
>
> We thank the reviewer for raising this point, which gives us the opportunity to clarify the novelty and contributions of our work beyond engineering implementation.
>
> We first review the prior application of fractional calculus into SNNs/ANNs and then position our contribution accordingly.\
> **Fractional biological neuron modelling and shallow fractional Hopfield spiking network.** At the neuron level, the *f*-LIF modelling [A1,A2] of biological neurons explains spike-frequency adaptation in pyramidal neurons [A3] and yields more reliable spike patterns under noise [A1].
> The work [A4] proposes that a neuron's spike‑train can be interpreted as a fractional derivative of its input signal. They show encoding/decoding efficiency and link fractional dynamics to predictive coding.
> At the network level, related efforts investigate shallow fractional Hopfield-type spiking networks [A5]. These studies primarily focus on dynamical system properties, proving the coexistence of multiple equilibrium points, solution boundedness, and global attractivity—rather than learning representations for complex tasks.
>
> ***Distinction: Crucially, prior work is restricted to biological modeling, signal‑approximation, or dynamical analysis of fixed-weight, shallow networks, neglecting the learning problem. We bridge this gap by formulating the first generalizable *f*-SNNs framework for end-to-end training. This advances *f*-SNNs from theoretical constructs to a trainable computational paradigm compatible with modern deep architectures (e.g., Transformers), strictly generalizing integer-order SNNs.***
>
>
> **Fractional deep learning and fractional differential equation neural solvers.**
> In the continuous ANN domain, fractional calculus has been integrated into deep learning frameworks to enhance expressivity. For instance [A6, A7] leverage fractional calculus to improve graph neural network performance and robustness, while [A8] utilizes fractional diffusion processes to improve diversity in generative modeling.
> Separately, in the domain of scientific computing, Physics-Informed Neural Networks (PINNs) have been extended to solve fractional partial differential equations (*f*-PINNs) [A9]. Subsequent developments have focused on scalability, such as gradient-enhanced variants for convergence [A10], and optimized training via operator-matrix methods for high-dimensional problems [A11, A12].
>
> ***Distinction: Our *f*-SNN model fundamentally differs from these approaches. First, unlike f-PINNs, which serve as function approximators to solve a given fractional equation, we embed fractional dynamics inside the neuron model as a computational engine. Second, unlike fractional ANNs that operate on continuous signals, *f*-SNNs function in the discrete, event-driven domain.***
>
>
> **Fractional-order gradients for training NNs.** A complementary line of research applies fractional derivatives to define gradient operators and learning dynamics for training SNNs/ANNs. For example, fractional gradient descent algorithms [A13, A14] replace the standard integer-order gradient update with a fractional counterpart. These methods smooth the optimization landscape, enabling faster convergence and better escape from local minima compared to standard stochastic gradient descent (SGD). In the spiking domain, Gyongyossy et al. [A15, A16, A17] have applied fractional gradients for training SNNs.
>
> ***Distinction: These approaches use fractional calculus as an optimization tool to adjust the weight update trajectory, whereas our work embeds fractional dynamics within the neurons themselves. This is analogous to the difference between designing a network optimizer (Adam vs. SGD) versus changing the network architecture (CNN vs. Transformer).***

---

> > ### Author Response · Authors · 2025-11-21
> > **Reference List**
> >
> > [A1] Teka, Wondimu, et al. "Neuronal spike timing adaptation described with a fractional leaky integrate-and-fire model." *PLoS Computational Biology*, 10(3):e1003526, 2014.
> >
> > [A2] Deng, Yabin, et al. "Fractional spiking neuron: Fractional leaky integrate-and-fire circuit described with dendritic fractal model." *IEEE Transactions on Biomedical Circuits and Systems*, 16(6):1375–1386, 2022.
> >
> > [A3] Ha, Go Eun, et al. "Spike frequency adaptation in neurons of the central nervous system." *Experimental Neurobiology*, 26(4):179, 2017.
> >
> > [A4] Rombouts, Jaldert, et al. "Fractionally predictive spiking neurons." *Advances in Neural Information Processing Systems*, 23, 2010.
> >
> > [A5] Zhang, Shuo, et al. "Multistability and global attractivity for fractional-order spiking neural networks." *Applied Mathematics and Computation*, 508:129617, 2026.
> >
> > [A6] Kang, Qiyu, et al. "Unleashing the potential of fractional calculus in graph neural networks with FROND." In *Proceedings of the International Conference on Learning Representations*, 2024.
> >
> > [A7] Kang, Qiyu, et al. "Coupling graph neural networks with fractional order continuous dynamics: A robustness study." In *Proceedings of the AAAI Conference on Artificial Intelligence*, Vancouver, Canada, February 2024.
> >
> > [A8] Nobis, Gabriel, et al. "Generative fractional diffusion models." *Advances in Neural Information Processing Systems*, 37: 25469–25509, 2024.
> >
> > [A9] Pang, Guofei, et al. "fpinns: Fractional physics-informed neural networks." *SIAM Journal on Scientific Computing*, 41(4): A2603–A2626, 2019.
> >
> > [A10] Yu, Jeremy, et al. "Gradient-enhanced physics-informed neural networks for forward and inverse PDE problems." *Computer Methods in Applied Mechanics and Engineering*, 393: 114823, 2022.
> >
> > [A11] Ma, Lei, et al. "Bi-orthogonal fpinn: A physics-informed neural network method for solving time-dependent stochastic fractional PDEs." *arXiv preprint*, arXiv:2303.10913, 2023.
> >
> > [A12] Taheri, Tayebeh, et al. "Accelerating fractional PINNs using operational matrices of derivative." *arXiv preprint*, arXiv:2401.14081, 2024.
> >
> > [A13] Khan, Shujaat, et al. "A novel fractional gradient-based learning algorithm for recurrent neural networks." *Circuits, Systems, and Signal Processing*, 37(2):593–612, 2018.
> >
> > [A14] Shin, Yeonjong, et al. "Accelerating gradient descent and Adam via fractional gradients." *Neural Networks*, 161:185–201, 2023.
> >
> > [A15] Gyöngyössy, Natabara Máté, et al. "Exploring the effects of Caputo fractional derivative in spiking neural network training." *Electronics*, 11(14):2114, 2022.
> >
> > [A16] Yang, Yi, et al. "Fractional-order spike-timing-dependent gradient descent for multi-layer spiking neural networks." *Neurocomputing*, 611:128662, 2025.
> >
> > [A17] Yang, Honggang, et al. "Fractional gradient descent method for spiking neural networks." In *2023 2nd Conference on Fully Actuated System Theory and Applications (CFASTA)*, pp. 636–641. IEEE, 2023.

---

> ### Author Response · Authors · 2025-11-21
> **Weakness 2: Performance over More Datasets**
>
> > The datasets used are limited, and results on widely tested datasets such as CIFAR-10, CIFAR-100, and ImageNet are missing.
>
> **Response:**
>
> We have added results on more datasets, including CIFAR-10, CIFAR-100, and ImageNet.
> For the relatively smaller CIFAR-10 and CIFAR-100 datasets, we adopt Spiking-ResNet-18 as the baseline.
> For ImageNet, we follow the SpikFormer [C1] configuration with 29.7M parameters, and set both the training and validation image sizes to $160 \times 160$ for a fair comparison. The results are shown in **Table B1**.
>  Our *f*-LIF achieves clear gains over both LIF (SpikingJelly) and LIF (snnTorch) on all datasets.
>
>  | Datasets | Architecture | Timesteps | LIF (SJ) | LIF (snnTorch) | *f*-LIF (*f*-SNN) |
> |----------|--------------|-----------|----------|----------------|-------------------|
> | CIFAR-10 | Spiking-ResNet-18 | 4 | 0.9134 | 0.9026 | __0.9215__ |
> | CIFAR-100 | Spiking-ResNet-18 | 4 | 0.6813 | 0.6445 | __0.6874__ |
> | ImageNet | SpikFormer | 4 | 0.6637 | 0.6584 | __0.6791__ |
> | ImageNet (spike encoder) | SpikFormer | 4 | 0.5549 | 0.5432 | __0.5738__ |
>
> **Table B1:** Top-1 accuracy across datasets, architectures, and timesteps for LIF baselines (SpikingJelly and snnTorch) and our fractional LIF (*f*-SNN).

---

> ### Author Response · Authors · 2025-11-21
> **Question: More detailed energy consumption analysis**
>
> > The energy consumption analysis of SNNs only includes synaptic analysis, while the energy consumption of the neurons themselves should also be compared.
>
> **Response:**
>
> **Energy analysis.**
> We thank the reviewers for the suggestion. Most existing SNN energy analyses primarily account for synaptic operations, while there is no widely adopted methodology for the intrinsic energy of neurons themselves. Therefore, we follow the commonly used practice to estimate the overall SNN energy. In the highlighted energy-analysis part, we use the same methodology as prior work. For neuron-intrinsic costs, we provide the following notation and derivations. The full energy analysis has been included in Section D.3. of the paper revision.
>
> **Notation**
> - $T$: Number of timesteps.
> - $\kappa := E_{MAC}/E_{AC}$: Conversion factor from one multiply–accumulate (MAC) operation to equivalent additions ($E_{AC}$).
> - $E_{AC}$: Energy of one equivalent addition;
> - $E_{MAC}$: Energy of one multiply–accumulate operation.
> - Relation: $E_{MAC} = (\kappa+1)\,E_{AC}$.
> - $N$: Number of SNN convolutional stages.
> - $M$: Number of SNN fully connected layers.
> - $L$: Number of self-attention (SSA) blocks.
> - $\mathrm{FLOPs}(l)$: Floating-point operations of layer $l$ in its dense counterpart.
> - $E_{MAC}$: Energy per multiply–accumulate operation.
> - $E_{AC}$: Energy per equivalent addition.
> - $\rho$: Average firing rate.
>
>
>
>
> ## I. LIF (discrete time, hard reset; single neuron, $T$ steps)
>
> Per-term energy (in units of $E_{AC}$):
> $$E_{\text{update,in}} = T \cdot \kappa \cdot E_{AC} \quad \text{(update: input multiplication)}$$
>
> $$E_{\text{leak}} = T \cdot \kappa \cdot E_{AC} \quad \text{(leak: multiplication)}$$
>
> $$E_{\text{update,sum}} = T \cdot 1 \cdot E_{AC} \quad \text{(update summation: add the two terms)}$$
>
> $$E_{\text{cmp}} = T \cdot 1 \cdot E_{AC} \quad \text{(threshold comparison)}$$
>
> $$E_{\text{spike}} = T \cdot \rho \cdot (2\kappa + 1) \cdot E_{AC} \quad \text{(spike: hard reset, arithmetic gating)}$$
>
> Total energy of a single LIF neuron over $T$ steps:
> $$E_{\text{neuron}} = E_{\text{total}}^{\mathrm{LIF},\mathrm{hard}}
> = T \cdot \big[ (2 + 2\rho)\,\kappa + (2 + \rho) \big] \cdot E_{AC}.$$
>
> ## II. *f*-LIF (discrete time, hard reset; single neuron, $T$ steps)
>
> Per-term energy (in units of $E_{AC}$):
> $$E_{\text{update+leak}} = T \cdot \log_2 T \cdot (\kappa+1) \cdot E_{AC} \quad \text{(update + leak: merged)}$$
>
> $$E_{\text{cmp}} = T \cdot 1 \cdot E_{AC} \quad \text{(threshold comparison)}$$
>
> $$E_{\text{spike}} = T \cdot \rho \cdot (2\kappa + 1) \cdot E_{AC} \quad \text{(spike: hard reset, arithmetic gating)}$$
>
> Total energy of a single *f*-LIF neuron over $T$ steps:
> $$E_{\text{neuron}} = E_{\text{total}}^{f\text{-LIF},\text{hard}}
> = T \cdot \Big[ (\kappa+1)\log_2 T + 1 + \rho(2\kappa + 1) \Big] \cdot E_{AC}.$$
>
> **Energy accounting for a Spiking-Transformer (SpikFormer).**
> Based on the updated formula, we further consider a more complex Spiking-Transformer setting and compare energy consumption accordingly. The energy of SpikFormer can be written as
> $$E_{\text{SpikFormer}} = E_{MAC} \times \text{FLOPs}^{1} Conv + E_{AC} \times \left( \sum_{n=2}^{N} \text{SOP}^{n} Conv + \sum_{m=1}^{M}  \text{SOP}^{m} FC+ \sum_{l=1}^{L} \text{SOP}^{l} SSA \right)$$
> where $\mathrm{SOP}$ denotes the number of spike-based accumulate operations.
>
> For each layer $l$, the spike-based operation count is
> $$\mathrm{SOPs}(l) = \rho \times T \times \mathrm{FLOPs}(l),$$
>
> Combining the above equations, the overall energy consumption can be expressed as
> $$\begin{align}
> E_{\text{total}}
> &= E_{\text{SpikFormer}} + E_{\text{neuron}} \times N_{\text{neurons}},
> \end{align}$$
> Here, $N_{\text{neurons}}$ denotes the total number of neurons in the network.
>
> We report per-layer energy with sparsity-aware synaptic costs and neuron-intrinsic costs. "Energy Type" indicates whether the layer is counted with $E_{MAC}$ or $E_{AC}$; entries of the form "$E_{AC} \times \rho$" denote $E_{AC}$ cost scaled by the measured average firing rate $\rho$.
>
> The detailed results are presented in **Table 16 and Table 17** of the revision. The Spikformer using LIF and the Spikformer using *f*-LIF theoretically consume **2.93mJ** and **2.91mJ**, respectively. Our *f*-LIF  neurons enable the network to achieve a **lower average firing rate** compared with the standard LIF node, although the fractional dynamics introduce additional computational overhead. This results in overall energy usage that remains at a comparable level, balancing the trade-off between energy efficiency and enhanced temporal modeling capabilities.

---

> ### Author Response · Authors · 2025-11-25
> **Thank you for your invaluable support!**
>
> Dear Reviewer xtnY,
>
> We sincerely appreciate the time and expertise you have dedicated to our paper. Your constructive feedback has been very helpful in refining our work, and we value your support.
>
> We have carefully addressed your comments in our revision and response. We would be grateful if you could confirm whether these updates have adequately resolved your concerns. Thank you again for your guidance.
>
> Best regards,
>
> Authors

---

> ### Comment · Reviewer_xtnY · 2025-11-28
>
> The author conducted experiments on more datasets and explained the innovation, so I decided to raise my score.

---

### Official Review · Reviewer_3jw6 · 2025-11-11

**Soundness:** 3
**Presentation:** 3
**Contribution:** 2
**Rating:** 4
**Confidence:** 2

**Summary:**

This paper proposes fractional-order spiking neural networks (s-SNN). The idea is to replace the linear ODE that describes the pre-spike potential dynamics of a (L)IF model with a fractional ODE. This allows for better capturing of long-term dependencies in the membrane potential evolution, enhancing the expressivity of the corresponding models. In particular, this paper shows how f-SNNs can be seen as a generalization of the (L)-IF SNNs and how to effectively discretize f-SNNs to get an implementable model using the Adams-Bashforth-Moulton discretization scheme. Finally, the author empirically demonstrates that f-SNNs consistently outperform traditional SNNs in both performance and robustness to input perturbations across multiple tasks and architectures. The authors further developed an open-source toolbox compatible with PyTorch, simplifying the design and implementation of custom SNN neurons.

**Strengths:**

The paper is well-written, precise, and enjoyable to read. The definition of the continuous-time f-SNN and the derivation of the discrete-time counterpart are well-explained. Also, the experimental setup is clear and well-documented, and the results are consistent with the claims.

**Weaknesses:**

The paper contains some minor imprecision that can be easily fixed. For instance, in equation (1) line 126, there is an a which I suppose is meant to be a 0. Moreover, in line 172, the authors mentioned "learnable synaptic weights" that are not introduced since in line 170, only the input current is mentioned $I_{in, k}$. The caption of Figure 4, page. 7, could be a little bit extended by explaining how to interpret Figure 4 (at least to me, it is not straightforward).

I would appreciate seeing a related work section in the main part and a more detailed comparison with the existing models that exploit fractional derivatives. The authors mention them Appendix A2, but it is not clear how it compares with your model. This would be essential for the novelty assessment and the main reason for the overall score.

The theoretical result (Theorem 1 in Section 3.2) seems somewhat modest to be presented as a theorem, as it addresses only the constant input current case and relies on known solutions of the corresponding ODE and fractional differential equation. It might be more appropriate to present it as a remark.

**Questions:**

I ask the following questions to the authors:

1) In the article "Time to Spike? Understanding the Representational Power of Spiking Neural Networks in Discrete Time" the authors study the expressivity of the LIF SNN model both in terms of approximation properties and in terms of linear regions in the static case. Would you expect the f-SNNs neuron to produce more regions in the input space? Would you expect that f-SNNs neuron regions are in more general positions?

2) Combining the fact that $\alpha$ is selected through a hyperparameter tuning and that training SNNs in general is hard, what can you say about the overall efficiency of the training of an f-SNN model?

---

> ### Author Response · Authors · 2025-11-21
> **Weakness 1: Notation Clarification and Figure Interpretation**
>
> >The paper contains some minor imprecision that can be easily fixed. For instance, in equation (1) line 126 , there is an $a$ which I suppose is meant to be a 0 . Moreover, in line 172 , the authors mentioned "learnable synaptic weights" that are not introduced since in line 170 , only the input current is mentioned $I_{i n, k}$. The caption of Figure 4 , page. 7 , could be a little bit extended by explaining how to interpret Figure 4 (at least to me, it is not straightforward).
>
>
> **Response:**
>
> We sincerely thank the reviewer for these constructive comments. We have carefully addressed all the mentioned issues as follows:
>
> - **Correction in Equation (1):** The symbol $a$ was indeed a typo and has been corrected to 0.
>
> - **Clarification of learnable synaptic weights notation:** We acknowledge the confusion regarding the notation. To address this:
>
>     - We have revised our notation to be more explicit in response to the reviewer's suggestions. Following the convention in recent SNN literature [R1,R2], we now denote the presynaptic input as $X_k^{(\Phi)}$, where $\Phi$ explicitly represents the learnable synaptic weights of an arbitrary layer (e.g., convolutional, fully connected, or Transformer blocks).
>
>     - This notation choice is deliberate: while some works [R3,R4] embed weight matrices directly in the neuron dynamics (e.g., $\boldsymbol{u}(t)=\beta \boldsymbol{u}(t-1)+\boldsymbol{W} \boldsymbol{x}+\boldsymbol{b}-\theta \boldsymbol{s}(t)$), we opted for a more general formulation to emphasize that our framework is architecture-agnostic and supports diverse layer types beyond simple FC/CNN architectures.
>
> - **Enhanced Figure 4 caption:** Robustness comparison between the proposed *f*-SNN and two integer-order baselines (LIF in SpikingJelly and LIF in snnTorch). Left: Radar chart aggregating five corruption types (larger is better): Gaussian noise injection, center occlude block, temporal truncate, temporal jitter, and discard frame. Middle: Performance vs. noise level (x-axis: Gaussian noise std). Right: Performance vs. occlusion ratio (x-axis: area ratio of the center block). The *f*-LIF (*f*-SNN) shows consistently higher performance and slower degradation under all corruption types.
>
> References:
> [R1] Yao, Man, et al. “Spike-driven transformer v2: Meta spiking neural network architecture inspiring the design of next-generation neuromorphic chips.” ICLR 2024.
> [R2] Yao, Man, et al. “Attention spiking neural networks.” IEEE TPAMI 45.8 (2023): 9393–9410.
> [R3] Eshraghian, Jason K., et al. “Training spiking neural networks using lessons from deep learning.” Proceedings of the IEEE 111(9): 1016–1054, 2023.
> [R4] Nguyen, Duc Anh, et al. “Time to Spike? Understanding the Representational Power of Spiking Neural Networks in Discrete Time.” ICML 2025.

---

> ### Author Response · Authors · 2025-11-21
> **Weakness 2: Bring the related work section into the paper's main context and clarify novelty**
>
> > I would appreciate seeing a related work section in the main part and a more detailed comparison with the existing models that exploit fractional derivatives. The authors mention them Appendix A2, but it is not clear how it compares with your model. This would be essential for the novelty assessment and the main reason for the overall score.
>
> **Response:**
>
> Thank you for the suggestion. We have expanded the related work section and integrated it into the main paper (see Page 3), providing clear distinctions between our work and prior studies.
>
> We first review the prior application of fractional calculus into SNNs/ANNs and then position our contribution accordingly.\
> **Fractional biological neuron modelling and shallow fractional Hopfield spiking network.** At the neuron level, the *f*-LIF modelling [B1,B2] of biological neurons explains spike-frequency adaptation in pyramidal neurons [B3] and yields more reliable spike patterns under noise [B1].
> The work [B4] proposes that a neuron's spike‑train can be interpreted as a fractional derivative of its input signal. They show encoding/decoding efficiency and link fractional dynamics to predictive coding.
> At the network level, related efforts investigate shallow fractional Hopfield-type spiking networks [B5]. These studies primarily focus on dynamical system properties, proving the coexistence of multiple equilibrium points, solution boundedness, and global attractivity—rather than learning representations for complex tasks.
>
> ***Distinction: Crucially, prior work is restricted to biological modeling, signal‑approximation, or dynamical analysis of fixed-weight, shallow networks, neglecting the learning problem. We bridge this gap by formulating the first generalizable *f*-SNNs framework for end-to-end training. This advances *f*-SNNs from theoretical constructs to a trainable computational paradigm compatible with modern deep architectures (e.g., Transformers), strictly generalizing integer-order SNNs.***
>
>
> **Fractional deep learning and fractional differential equation neural solvers.**
> In the continuous ANN domain, fractional calculus has been integrated into deep learning frameworks to enhance expressivity. For instance [B6, B7] leverage fractional calculus to improve graph neural network performance and robustness, while [B8] utilizes fractional diffusion processes to improve diversity in generative modeling.
> Separately, in the domain of scientific computing, Physics-Informed Neural Networks (PINNs) have been extended to solve fractional partial differential equations (*f*-PINNs) [B9]. Subsequent developments have focused on scalability, such as gradient-enhanced variants for convergence [B10], and optimized training via operator-matrix methods for high-dimensional problems [B11, B12].
>
> ***Distinction: Our *f*-SNN model fundamentally differs from these approaches. First, unlike f-PINNs, which serve as function approximators to solve a given fractional equation, we embed fractional dynamics inside the neuron model as a computational engine. Second, unlike fractional ANNs that operate on continuous signals, *f*-SNNs function in the discrete, event-driven domain.***
>
>
> **Fractional-order gradients for training NNs.** A complementary line of research applies fractional derivatives to define gradient operators and learning dynamics for training SNNs/ANNs. For example, fractional gradient descent algorithms [B13, B14] replace the standard integer-order gradient update with a fractional counterpart. These methods smooth the optimization landscape, enabling faster convergence and better escape from local minima compared to standard stochastic gradient descent (SGD). In the spiking domain, Gyongyossy et al. [B15, B16, B17] have applied fractional gradients for training SNNs.
>
> ***Distinction: These approaches use fractional calculus as an optimization tool to adjust the weight update trajectory, whereas our work embeds fractional dynamics within the neurons themselves. This is analogous to the difference between designing a network optimizer (Adam vs. SGD) versus changing the network architecture (CNN vs. Transformer).***

---

> > ### Author Response · Authors · 2025-11-21
> > **Reference List**
> >
> > [B1] Teka, Wondimu, et al. "Neuronal spike timing adaptation described with a fractional leaky integrate-and-fire model." *PLoS Computational Biology*, 10(3):e1003526, 2014.
> >
> > [B2] Deng, Yabin, et al. "Fractional spiking neuron: Fractional leaky integrate-and-fire circuit described with dendritic fractal model." *IEEE Transactions on Biomedical Circuits and Systems*, 16(6):1375–1386, 2022.
> >
> > [B3] Ha, Go Eun, et al. "Spike frequency adaptation in neurons of the central nervous system." *Experimental Neurobiology*, 26(4):179, 2017.
> >
> > [B4] Rombouts, Jaldert, et al. "Fractionally predictive spiking neurons." *Advances in Neural Information Processing Systems*, 23, 2010.
> >
> > [B5] Zhang, Shuo, et al. "Multistability and global attractivity for fractional-order spiking neural networks." *Applied Mathematics and Computation*, 508:129617, 2026.
> >
> > [B6] Kang, Qiyu, et al. "Unleashing the potential of fractional calculus in graph neural networks with FROND." In *Proceedings of the International Conference on Learning Representations*, 2024.
> >
> > [B7] Kang, Qiyu, et al. "Coupling graph neural networks with fractional order continuous dynamics: A robustness study." In *Proceedings of the AAAI Conference on Artificial Intelligence*, Vancouver, Canada, February 2024.
> >
> > [B8] Nobis, Gabriel, et al. "Generative fractional diffusion models." *Advances in Neural Information Processing Systems*, 37: 25469–25509, 2024.
> >
> > [B9] Pang, Guofei, et al. "fpinns: Fractional physics-informed neural networks." *SIAM Journal on Scientific Computing*, 41(4): A2603–A2626, 2019.
> >
> > [B10] Yu, Jeremy, et al. "Gradient-enhanced physics-informed neural networks for forward and inverse PDE problems." *Computer Methods in Applied Mechanics and Engineering*, 393: 114823, 2022.
> >
> > [B11] Ma, Lei, et al. "Bi-orthogonal fpinn: A physics-informed neural network method for solving time-dependent stochastic fractional PDEs." *arXiv preprint*, arXiv:2303.10913, 2023.
> >
> > [B12] Taheri, Tayebeh, et al. "Accelerating fractional PINNs using operational matrices of derivative." *arXiv preprint*, arXiv:2401.14081, 2024.
> >
> > [B13] Khan, Shujaat, et al. "A novel fractional gradient-based learning algorithm for recurrent neural networks." *Circuits, Systems, and Signal Processing*, 37(2):593–612, 2018.
> >
> > [B14] Shin, Yeonjong, et al. "Accelerating gradient descent and Adam via fractional gradients." *Neural Networks*, 161:185–201, 2023.
> >
> > [B15] Gyöngyössy, Natabara Máté, et al. "Exploring the effects of Caputo fractional derivative in spiking neural network training." *Electronics*, 11(14):2114, 2022.
> >
> > [B16] Yang, Yi, et al. "Fractional-order spike-timing-dependent gradient descent for multi-layer spiking neural networks." *Neurocomputing*, 611:128662, 2025.
> >
> > [B17] Yang, Honggang, et al. "Fractional gradient descent method for spiking neural networks." In *2023 2nd Conference on Fully Actuated System Theory and Applications (CFASTA)*, pp. 636–641. IEEE, 2023.

---

> ### Author Response · Authors · 2025-11-21
> **Weakness 3 & Question 1: More theoretical understanding of f-SNN**
>
> >  The theoretical result (Theorem 1 in Section 3.2) ....It might be more appropriate to present it as a remark.
>
> > In the article "Time to Spike? Understanding the Representational Power of Spike.... Would you expect that \emph{f}-SNNs neuron regions are in more general positions?
>
> **Response:**
>
> Thank you for your valuable comments. We have renamed Theorem 1 as Proposition 1 to clarify its illustrative role in demonstrating the strong memory influence of *f*-SNNs. Furthermore, to deepen the theoretical understanding of the robustness and expressivity of *f*-SNNs over time, we have added two new theorems. We show that fractional-order systems strictly exceed the expressive capacity of integer-order models. We also demonstrate that *f*-SNNs show superior robustness to input perturbations.
>
> The following have been added to Section 3.2 of the paper. The full proofs are provided in Sections C.2 and C.3.
>
> The *f*-SNN framework demonstrates superior robustness compared to traditional SNNs. Empirical studies show that *f*-LIF neurons maintain reliable spike patterns under noisy inputs [C1]. Here, we provide theoretical robustness guarantees.
>
> **Theorem 1 (Robustness of *f*-SNN).**
> Consider a fractional *f*-IF neuron governed by the dynamics $\tau D^\alpha U(t)=R I_{\mathrm{in}}(t)$ with fractional order $0<\alpha<1$ and initial condition $U_0=0$. Under a constant input current $I_{\mathrm{c}}$ subject to an additive perturbation $\epsilon$ (where $|\epsilon| \ll I_{\mathrm{c}}$), the system exhibits the following robustness properties relative to the classical integer-order model ($\alpha=1$):
>
> - **Membrane Potential Robustness:** The membrane potential deviation due to perturbation evolves as:
> $$\begin{aligned}
> \Delta U^{f-\mathrm{IF}}(t) & =\frac{R \epsilon}{\tau \Gamma(\alpha+1)} t^\alpha \quad(\text{sub-linear growth}), \\\\
> \Delta U^{\mathrm{IF}}(t) & =\frac{R \epsilon}{\tau} t \quad(\text{linear growth})
> \end{aligned}$$
> For $0<\alpha<1$, the fractional-order dynamics suppress long-term perturbation accumulation through sub-linear temporal scaling.
>
> - **Spike Timing Sensitivity:** For small perturbations $\epsilon \ll I_{\mathrm{c}}$, the spike time shift magnitude scales as:
> $$\begin{aligned}
> \left|\Delta t_s^{f-\mathrm{IF}}\right| & \propto \epsilon \cdot I_{\mathrm{c}}^{-(1+1/\alpha)}, \\\\
> \left|\Delta t_s^{\mathrm{IF}}\right| & \propto \epsilon \cdot I_{\mathrm{c}}^{-2}
> \end{aligned}$$
> Since $(1+1/\alpha)>2$ for $0<\alpha<1$, the fractional-order model exhibits enhanced spike timing robustness for high input currents.
>
> **Remark 4.** The fractional-order dynamics yield distinct robustness advantages. The sub-linear perturbation growth $t^\alpha(\alpha<1)$ significantly suppresses long-term accumulation compared to linear growth in classical models. Additionally, the enhanced spike timing stability becomes crucial for precise temporal coding applications [C2,C3]. These properties make *f*-SNNs particularly suited for tasks requiring sustained accuracy and temporal precision under varying input conditions, as confirmed by our experiments in Section 4.
>
> We now establish that *f*-SNNs possess computational capabilities that fundamentally exceed those of finite integer-order systems:
>
> **Theorem 2 (Irreducibility of *f*-IF Dynamics to Finite Classic LIF Ensembles).** Let $U^{f\text{-IF}}$ denote the trajectory of a *f*-IF neuron with order $\alpha \in(0,1)$. There exist no finite integer $W$, weights $\\{\phi_i\\}\_{i=1}^W$, and leak factors $\\{\beta_i\\}\_{i=1}^W$ such that the following holds:
>
> $$\hat{U}\_k=\sum_{i=1}^W \phi_i U_k^{\mathrm{LIF}(\beta_i)} \equiv U_k^{f-\mathrm{IF}} \quad \forall k$$
>
> for general input $X_k$. The _impulse response_ error of the approximation is $O(k^{\alpha-1})$, decaying algebraically slowly. The f-IF neuron is mathematically equivalent to an aggregate of integer-order LIF neurons if and only if $W \rightarrow \infty$, specifically as an integral over a continuum of leak factors.
>
> **Remark 5 (Implications for Neural Computation and Expressive Power).** Theorem 2 demonstrates that *f*-SNNs possess a temporal expressivity that surpasses that of finite integer-order SNN ensembles. A single *f*-IF neuron can represent a continuum of timescales, whereas an infinite bank of integer-order units would be needed for exact representation. The slow $O(k^{\alpha-1})$ decay of the impulse response approximation error highlights that such long-range dependencies are inaccessible to finite-order models without considerable architectural complexity.
>
>
> [C1] Teka, Wondimu, et al. "Neuronal spike timing adaptation described with a fractional leaky integrate-and-fire model." *PLoS Computational Biology*, 2014.
>
> [C2] Sander M. Bohte, et al. "Error-backpropagation in temporally encoded networks of spiking neurons." *Neurocomputing*, 2002.
>
> [C3] Nitin Rathi, et al. "Enabling deep spiking neural networks with hybrid conversion and spike timing dependent backpropagation." ICLR, 2019.

---

> ### Author Response · Authors · 2025-11-21
> **Question 2: Discussion of f-SNN training efficiency (including ways to improve it)**
>
> > Combining the fact that $\alpha$ is selected through a hyperparameter tuning and that training SNNs in general is hard, what can you say about the overall efficiency of the training of an f-SNN model?
>
> **Response:**
>
> We appreciate the reviewer’s comments regarding training efficiency and the potential overhead of hyperparameter tuning.
>
> **Learnable vs. Fixed $\alpha$.** **Our framework supports treating $\alpha$ as a learnable parameter.**
> As demonstrated in Table 6 (Sec. D.2.2) and **Table A1**, a learnable $\alpha$ achieves performance on par with manual tuning and consistently outperforms the integer-order LIF baseline.
> We report manually tuned results primarily to **ensure numerical stability in extreme regimes**.
>
>
> The fractional derivative definition involves the Gamma function $\Gamma(\alpha)$, which contains poles (singularities) at $\alpha=0$.
>
> - **Regime $\alpha \gg 0$:** For datasets like N-Caltech101, where optimal dynamics require larger $\alpha$, joint optimization is stable and effective. (The best-performing $\alpha$ values are given in **Table A2**)
> - **Regime $\alpha \rightarrow 0$:** For datasets requiring long-term memory (e.g., HARDVS), the optimal $\alpha$ is small. Learning $\alpha$ in this region risks gradient explosion due to the digamma function's behavior near the pole.
>
> Therefore, *manual tuning was chosen to enforce a rigorous, consistent evaluation standard across all datasets*, regardless of their temporal characteristics.
>
> ---
>
> | $\alpha$ | 0.2 | 0.4 | 0.6 | 0.8 | 0.9 | 1.0 | Learnable | LIF (SJ) |
> |--------|-----|-----|-----|-----|-----|-----|-----------|----------|
> | Acc1   | 0.7426 | 0.7325 | 0.7548 | 0.7595 | 0.7626 | 0.7421 | 0.7543 (Final $\alpha$: 0.9106) | 0.7263 |
>
> **Table A1:** Accuracy (Acc1) across different fractional orders $\alpha$, a learnable-$\alpha$ setting (final learned $\alpha$ = 0.9106), and the LIF (SJ) baseline on the N-Caltech101 dataset using a Transformer-based SNN. The learnable-$\alpha$ setting achieved results comparable to manually tuned $\alpha$.
>
> ---
>
> **Marginal Cost of Tuning.** It is important to note that $\alpha$ is a single scalar hyperparameter. Unlike architecture search, tuning $\alpha$ requires only a coarse grid search (e.g., interval of 0.1 or 0.2). The empirical improvements (accuracy boost and robustness improvement) outweigh the overhead required to find the optimal $\alpha$.
>
> **Path to More Efficiency: Distributed Fractional Operators.** To solve the stability issue at $\alpha \rightarrow 0$ and fully automate training in future work, we propose adopting *Distributed Fractional-Order Differential Equations* [D1, D2]. Rather than optimizing a single parameter $\alpha$, we can model the system as a weighted superposition of derivatives:
> $$\mathcal{D}[U(t)] \approx \sum_{j=0}^n w_j D^{\alpha_j} U(t)$$
> Here, $\alpha_j$ are fixed discretization points (avoiding singularities) and $w_j$ are learnable weights. This formulation converts finding $\alpha$ into a convex optimization of weights $w_j$, ensuring both training stability and efficiency.
>
> ---
>
> | Dataset | DVS-Gesture128 | N-Caltech101 | HAR-DVS | DVS-Lips |
> |---------|----------------|--------------|---------|----------|
> | Best $\alpha$ | 0.5 | 0.9 | 0.1 | 0.1 |
>
> **Table A2:** Best-performing $\alpha$ values for different datasets used in our experiments.
>
>
> ---
>
>
> **SNN and *f*-SNN Training.** Conventional SNNs predominantly rely on Backpropagation Through Time (BPTT) with surrogate gradients; we adhere to this standard for training our *f*-SNN. As detailed in **Table A3**, our model achieves training speeds comparable to the LIF implementation in snnTorch and is only marginally slower than the highly optimized LIF in SpikingJelly. Notably, utilizing a truncated history window allows *f*-SNN to surpass baseline speeds while preserving (or even improving) accuracy. Furthermore, during the rebuttal, we implemented the fractional adjoint sensitivity method [D3]. As shown in **Table A3**, this approach substantially reduces peak memory usage while maintaining competitive training speeds. Overall, the introduction of fractional dynamics does not impose a significant computational burden compared to standard SNNs. While alternative learning rules (e.g., STDP) are promising, they remain orthogonal to the contributions of this work.
>
> [D1] Ding, Wei, et al. "Applications of distributed-order fractional operators: A review." Entropy 23.1 (2021): 110.
>
> [D2] Diethelm, Kai, and Neville J. Ford. "Numerical analysis for distributed-order differential equations." J. Computational Applied Math. (2009): 96-104.
>
> [D3] SM, Sivalingam. "Neural Fractional Order Differential Equations with Adjoint Based Training."

---

> > ### Author Response · Authors · 2025-11-21
> > **Table**
> >
> > | Model | Train speed (FPS) | Test speed (FPS) | Memory (GB) | Acc1 |
> > |-------|-------------------|------------------|-------------|------|
> > | LIF (snnTorch) | 125.85 | 258.26 | 12.89 | 0.4610 |
> > | LIF (SpikingJelly) | 143.58 | 276.20 | 12.36 | 0.4626 |
> > | *f*-LIF (*f*-SNN) | 126.77 | 260.24 | 13.84 | 0.4766 |
> > | *f*-LIF ($M=1$) | 167.83 | 344.79 | 13.84 | 0.4518 |
> > | *f*-LIF ($M=2$) | 147.67 | 308.72 | 13.84 | 0.4656 |
> > | *f*-LIF ($M=4$) | 138.70 | 268.23 | 13.84 | 0.4712 |
> > | *f*-LIF (*f*-SNN, adjoint) | 98.74 | 238.23 | 9.35 | 0.4752 |
> >
> > **Table A3:** Training and inference speed (FPS), peak memory (GB), and top-1 accuracy (Acc1) across implementations. Here, $M$ denotes the truncation window size for history terms; "adjoint" indicates the training with adjoint method. Experiments are conducted on a CNN-based SNN architecture with the large-scale dataset HAR-DVS with a timestamp of 8.

---

> ### Author Response · Authors · 2025-11-25
> **Looking forward to receiving your feedback**
>
> Dear Reviewer 3jw6,
>
> We sincerely appreciate the time and effort you dedicated to reviewing our paper! Could you kindly let us know if our responses have sufficiently addressed your concerns? We thank you again for your valuable time.
>
> Best Regards,
>
> Authors

---

> > ### Author Response · Authors · 2025-11-27
> > **Looking forward to receiving your feedback**
> >
> > Dear Reviewer 3jw6,
> >
> > We hope this message finds you well. We truly appreciate the time and effort you have dedicated to reviewing our work. We would be very grateful for your feedback on our rebuttal. If you require further clarification or have any additional concerns, please let us know. We are happy to engage in further discussion.
> >
> > Best wishes,
> >
> > The Authors

---

> ### Comment · Reviewer_3jw6 · 2025-11-27
>
> I want to thank the authors for the detailed clarification. I think that this work mainly contributes to show that fractional dynamics-based neurons actually enhance the capability and the stability of traditional LIF neuron-based models on different tasks. However, it looks like this model has already been proposed in the literature, and this work actually provides a complete framework to train and test f-SNNs (please correct me if I have misunderstood something).
>
> I conclude this comment with two additional questions:
>
> 1) The first question regards Theorem 2 (Irreducibility of f-IF Dynamics to Finite Classic LIF Ensembles). Since the output of a spiking neuron is given by the sequence of produced spikes, and Theorem 2 points out that it is not possible to reproduce the same f-, how do the properties that you have shown in f-IF neuron membrane potential not reproducible with an ensemble of IF neurons, but it is not clear how this relates to the output of the nerons and hence the model expressivity.
> 2) From the theoretical perspective, is it possible to extend the analysis for dynamic inputs? I think that this would strengthen the theoretical analysis of the model.
>
> Finally, I really appreciate the experimental part of this work. However, I feel that this model was already introduced in the literature, and the theoretical analysis is not strong enough to be considered a strong novel contribution. Having said so, I increase my score to 6, since I find the paper interesting, well-written, and the experimental section complete and very nice.

---

> ### Author Response · Authors · 2025-12-03
> **Thanks for the Revised Score; Responses to New Questions**
>
> We thank Reviewer 3jw6 for the positive assessment and for increasing the score in light of our response. We address the new questions as follows:
>
>
>
> ### **New Response 1: Further Clarification over Our Contribution**
>
> > However, it looks like this model has already been proposed in the literature, and this work actually provides a complete framework to train and test f-SNNs (please correct me if I have misunderstood something).*
>
> **Response:**
>
> We respectfully clarify the distinction between prior work and our contributions.
>
> The IF neuron was introduced by Stein (1965) [M1] as a biological model, not for computation. It was not until Maass (1997) [M2] that SNNs were formalized as a machine learning framework. Similarly, prior work on fractional LIF (*f*-LIF) neurons (e.g., [B1-B3]) focuses on biological modeling and signal approximation, not on training neural networks.
>
> Our work presents the **first complete framework for training fractional-order SNNs (*f*-SNNs)**, extending fractional dynamics into **trainable computational models**. Beyond the framework itself, we establish key **theoretical advantages** of *f*-SNNs over traditional SNNs: persistent memory via power-law relaxation, irreducibility to finite integer-order ensembles, and enhanced robustness to perturbations. We conduct **extensive experiments** demonstrating the effectiveness of incorporating fractional-order dynamics into deep SNNs across diverse benchmarks. Finally, our PyTorch-compatible toolbox `spikeDE` enables fractional variants of popular neuron models (ALIF, GLIF, CLIF, etc.), facilitating future research in this direction.
>
> [M1] Richard B. Stein. "A theoretical analysis of neuronal variability." Biophysical Journal, 5(2):173–194, 1965.
>
> [M2] Wolfgang Maass. Networks of spiking neurons: the third generation of neural network models. Neural networks, 10(9):1659–1671, 1997.
>
>
> ### **New Response 2: More Explanation for Theorem 2**
>
> > The first question regards Theorem 2 (Irreducibility of f-IF Dynamics to Finite Classic LIF Ensembles). Since the output of a spiking neuron is given by the sequence of produced spikes, and Theorem 2 points out that it is not possible to reproduce the same f-, how do the properties that you have shown in f-IF neuron membrane potential not reproducible with an ensemble of IF neurons, but it is not clear how this relates to the output of the nerons and hence the model expressivity.*
>
> **Response:**
>
> We thank the reviewer for this insightful question and apologize if our presentation was unclear. The reviewer raises an important point: Theorem 2 establishes irreducibility at the membrane potential level, but the functional output of a spiking neuron is the spike train. We now clarify the connection between membrane potential dynamics and spike output expressivity.
>
> **Clarification: From Membrane Potential to Spike Output:** The spike generation mechanism directly couples membrane potential to output: a spike is emitted when $U_k \geq \theta$ (threshold). Therefore, the spike train $S_k \in\{0,1\}$ is a deterministic function of the membrane trajectory:
>
> $$S_k=H\left(U_k-\theta\right)$$
>
> where $H(\cdot)$ is the Heaviside step function. **This coupling implies that differences in membrane potential dynamics propagate to differences in spike train patterns.**
>
> We formalize this connection with the following corollary:
>
> **Corollary 1 (Spike Train Irreducibility)**
> Let $0 < \alpha < 1$. For any finite integer $W$, weights $\\{\phi_i\\}_{i=1}^W \subset \mathbb{R}$, leak factors $\\{\beta_i\\}_{i=1}^W \subset (0,1)$, and any Boolean function $f \colon \\{0,1\\}^W \to \\{0,1\\}$, there exists an input sequence $\\{X_k\\}_{k \geq 0}$ and threshold $\theta > 0$ such that the spike train of the *f*-IF neuron cannot be reproduced by any Boolean combination of spike trains from the $W$ LIF neurons. That is,
> $$S_k^{f\text{-IF}} \neq f\bigl(S_k^{\mathrm{LIF}(\beta_1)}, \ldots, S_k^{\mathrm{LIF}(\beta_W)}\bigr) \quad \text{for some } k.$$
>
> **Remark**
> This corollary establishes that the expressivity advantage of *f*-IF neurons extends to the spike train level. While Theorem 2 demonstrates irreducibility in membrane potential dynamics, one might wonder whether this advantage could be "washed out" by the thresholding nonlinearity. The corollary confirms it is not: the spike patterns generated by a single *f*-IF neuron encode temporal information that no finite ensemble of LIF neurons can reproduce, even when their outputs are combined through arbitrary Boolean logic. This implies that *f*-SNNs possess fundamentally richer spike-based representations, enabling them to communicate long-range temporal dependencies through their spike trains in ways that conventional SNNs cannot.
>
> The full proof for this Corollary is provided in Section C.4.

---

> ### Author Response · Authors · 2025-12-03
> **New Response 3: Analysis for Dynamic Inputs**
>
> > From the theoretical perspective, is it possible to extend the analysis for dynamic inputs? I think that this would strengthen the theoretical analysis of the model.
>
> We thank the reviewer for this suggestion. We would like to clarify that **Theorem 2 and Corollary 1 already address dynamic (time-varying) inputs**. The partition-based expressivity analysis in [G1], while insightful, considers only static inputs. Extending this framework to sequential data presents substantial technical challenges that we believe merit dedicated future work. The present paper establishes the foundational framework and provides initial expressivity results (Theorem 2, Corollary 1) that motivate such investigations.
>
> [G1] "Time to Spike? Understanding the Representational Power of Spiking Neural Networks in Discrete Time" ICML 2025.

---

### Author Response · Authors · 2025-11-24
**Summary of Rebuttal:**

**Update:**

Dear PCs, SACs, and ACs,

We sincerely appreciate the time and effort invested by each reviewer in evaluating our submission.

Following the rebuttal phase, we are pleased to report that we have finally received ***positive ratings from all reviewers***.

- Reviewer 3jw6 has increased their rating from **`4 to 6`**. The official comments can be found [here](https://openreview.net/forum?id=NJhBSLJ0nL&filter=invitations%3AICLR.cc%2F2026%2FConference%2FSubmission10943%2F-%2FOfficial_Comment%20signatures%3AICLR.cc%2F2026%2FConference%2FSubmission10943%2FReviewer_3jw6&nesting=1&sort=date-desc)

- Reviewer xtnY has updated their rating from **`6 to a score >=8`**. The official comments can be found [here](https://openreview.net/forum?id=NJhBSLJ0nL&filter=invitations%3AICLR.cc%2F2026%2FConference%2FSubmission10943%2F-%2FOfficial_Comment%20signatures%3AICLR.cc%2F2026%2FConference%2FSubmission10943%2FReviewer_xtnY&nesting=1&sort=date-desc)
.

- Reviewer XVNF has increased their rating from **`4 to 6`**. The official comments can be found [here](https://openreview.net/forum?id=NJhBSLJ0nL&filter=invitations%3AICLR.cc%2F2026%2FConference%2FSubmission10943%2F-%2FOfficial_Comment%20signatures%3AICLR.cc%2F2026%2FConference%2FSubmission10943%2FReviewer_XVNF&nesting=1&sort=date-desc)
.

We believe these updates effectively address the reviewers’ concerns. We are grateful for their valuable insights, which have further improved the manuscript.

Best regards,

The Authors

----

We sincerely thank all reviewers for their valuable feedback! We summarize our rebuttal as follows:

1. __Novelty & Related Work (Reviewers 3jw6 & xtnY):__ We have expanded the related work section from the original appendix and integrated it into the main paper (Page 3) now, clearly distinguishing our contributions from prior fractional-order work. While previous work focused on biological modeling, signal‑approximation, or dynamical analysis of fixed-weight shallow networks, we present the first generalizable *f*-SNN framework for end-to-end training compatible with modern deep architectures.

2. __Theoretical Analysis (Reviewers 3jw6 & XVNF):__ We added two new theorems in Section 3.2 with full proofs in Appendix C.2-C.3: (i) Theorem 1 proves *f*-SNNs' superior robustness through sub-linear perturbation growth and enhanced spike timing stability; (ii) Theorem 2 establishes that *f*-SNNs possess computational capabilities strictly exceeding finite integer-order systems. We renamed the original Theorem 1 to Proposition 1 as suggested.

3. __Extended Experiments (Reviewer xtnY & XVNF):__ We added comprehensive results on CIFAR-10, CIFAR-100, and ImageNet datasets. The experimental results are presented in Table B1. Our *f*-SNN consistently outperforms both baselines across all datasets, demonstrating the generalizability of our approach beyond neuromorphic & graph datasets.

4. __Computational Efficiency & Energy Analysis (Reviewers xtnY & XVNF):__ We provided detailed empirical benchmarks showing *f*-SNN achieves comparable training/inference speeds to standard SNNs. We also implemented the fractional adjoint method, reducing peak memory by ~32% while maintaining performance. For energy analysis, we included detailed calculations for neuronal and synaptic energy. While *f*-LIF neurons consume slightly more energy than LIF neurons, the network’s lower spiking rate _keeps the total energy consumption comparable_.

5. __$\alpha$ Hyperparameter Clarification (Reviewers 3jw6 & XVNF):__ We demonstrated that $\alpha$ can be made learnable and achieves comparable performance to manual tuning. We chose consistent manual tuning for numerical stability in extreme regimes ($\alpha\rightarrow 0$). The optimal $\alpha$ shows intuitive dataset dependence: sparser datasets benefit from smaller $\alpha$ (stronger memory). We also proposed distributed fractional operators as a future direction for fully automated training.

Finally, we thank all reviewers again for their enormous effort in reviewing our paper. Without their valuable comments, we could not improve our manuscript to its current status. The rebuttal version has been updated with major changes highlighted in blue/brown.

---

### Meta-Review · Area_Chair_wwXg · 2026-01-07

**Summary:**

All reviewers agree the idea is biologically-motivated and empirically sound, but initially questioned (i) limited novelty beyond prior biological f-LIF models, (ii) missing theory on expressivity/robustness, (iii) absence of large-scale datasets and speed/memory benchmarks, (iv) unclear α-sensitivity and energy breakdown, and (v) readability issues in discretization and notation.

**Reviewer Concerns:**

Addressed:
Novelty gap clarified; authors position themselves as first end-to-end trainable f-SNN framework vs earlier fixed-weight/biological studies.
Two new theorems + corollary give formal robustness and irreducibility guarantees; dynamic inputs are covered.
CIFAR-10/100, ImageNet, HAR-DVS added; speed, memory and adjoint results provided; α ablation and learnable-α included; full neuron+synaptic energy model supplied.
Notation cleaned; memory-kernel figure added; SpikingJelly/snnTorch differences explained.
Outstanding (minor):
Theory still restricted to single-neuron perturbation and constant-input robustness; large-scale recurrent or deep-network stability unanswered.
Practical training near α→0 remains unstable (authors propose future distributed-order fix).
Broader baseline comparison with advanced integer-order neurons (ALIF/GLIF/PSN) mentioned but not exhaustively evaluated.

**Reviewer Scores:**

R1 (3jw6): 4 → 6 (raised in rebuttal)
R2 (xtnY): 6 → >7 (raised in rebuttal)
R3 (XVNF): 4 → 6 (raised in rebuttal)

---

### Decision · Program_Chairs · 2026-01-26

Accept (Poster)